# Intracellular magnesium optimizes transmission efficiency and plasticity of hippocampal synapses by reconfiguring their connectivity

Hang Zhou ®[1,2] ✉, Guo-Qiang Bi ®[1,2,3,4] & Guosong Liu ®[5,6] ✉

Synapses at dendritic branches exhibit specific properties for information processing. However, how the synapses are orchestrated to dynamically modify their properties, thus optimizing information processing, remains elusive. Here, we observed at hippocampal dendritic branches diverse configurations of synaptic connectivity, two extremes of which are characterized by low transmission efficiency, high plasticity and coding capacity, or inversely. The former favors information encoding, pertinent to learning, while the latter prefers information storage, relevant to memory. Presynaptic intracellular $Mg^{2+}$ crucially mediates the dynamic transition continuously between the two extreme configurations. Consequently, varying intracellular $Mg^{2+}$ levels endow individual branches with diverse synaptic computations, thus modulating their ability to process information. Notably, elevating brain $Mg^{2+}$ levels in aging animals restores synaptic configuration resembling that of young animals, coincident with improved learning and memory. These findings establish intracellular $Mg^{2+}$ as a crucial factor reconfiguring synaptic connectivity at dendrites, thus optimizing their branch-specific properties in information processing.

Modifications in synaptic connectivity play a pivotal role in information processing during learning and memory[1–6]. Previous research has predominantly focused on connectivity changes in single synapses during learning and memory, proposing classical paradigms such as synaptic potentiation and depression[1,5,7–10] that contribute to strengthening and weakening of synapses, respectively. When considering a group of synapses instead of a single one, learning induces populational changes in their synaptic connectivity, reshaping the distribution pattern of synaptic strengths (or weights)[3]. Such distribution changes are involved in encoding information during learning[3,11]; for instance, in cerebellar Purkinje cells, motor skill learning induces changes in synaptic weight distribution, which help to establish task-specific input-output associations, thus perpetuating the motor skill[3,11–13].

Despite the advancements made at the single-synapse and cellular levels, it remains elusive at the dendritic branch level how the synapses, with various presynaptic origins, are organized to generate diverse computational properties to meet varying demands of information processing. This inquiry is particularly important because dendritic branches are the basic unit for information processing in neural

[1]Faculty of Life and Health Sciences, Shenzhen University of Advanced Technology, Shenzhen 518107, China. [2]Interdisciplinary Center for Brain Information, Brain Cognition and Brain Disease Institute, Shenzhen Institute of Advanced Technology, Chinese Academy of Sciences, Shenzhen 518055, China. [3]Shenzhen-Hong Kong Institute of Brain Science, Shenzhen 518055, China. [4]Hefei National Laboratory for Physical Sciences at the Microscale, and School of Life Sciences, University of Science and Technology of China, Hefei 230031, China. [5]School of Medicine, Tsinghua University, Beijing 100084, China. [6]NeuroCentria Inc., Walnut Creek, CA 94596, USA. ✉e-mail: zhouhang01@outlook.com; liu.guosong@gmail.com

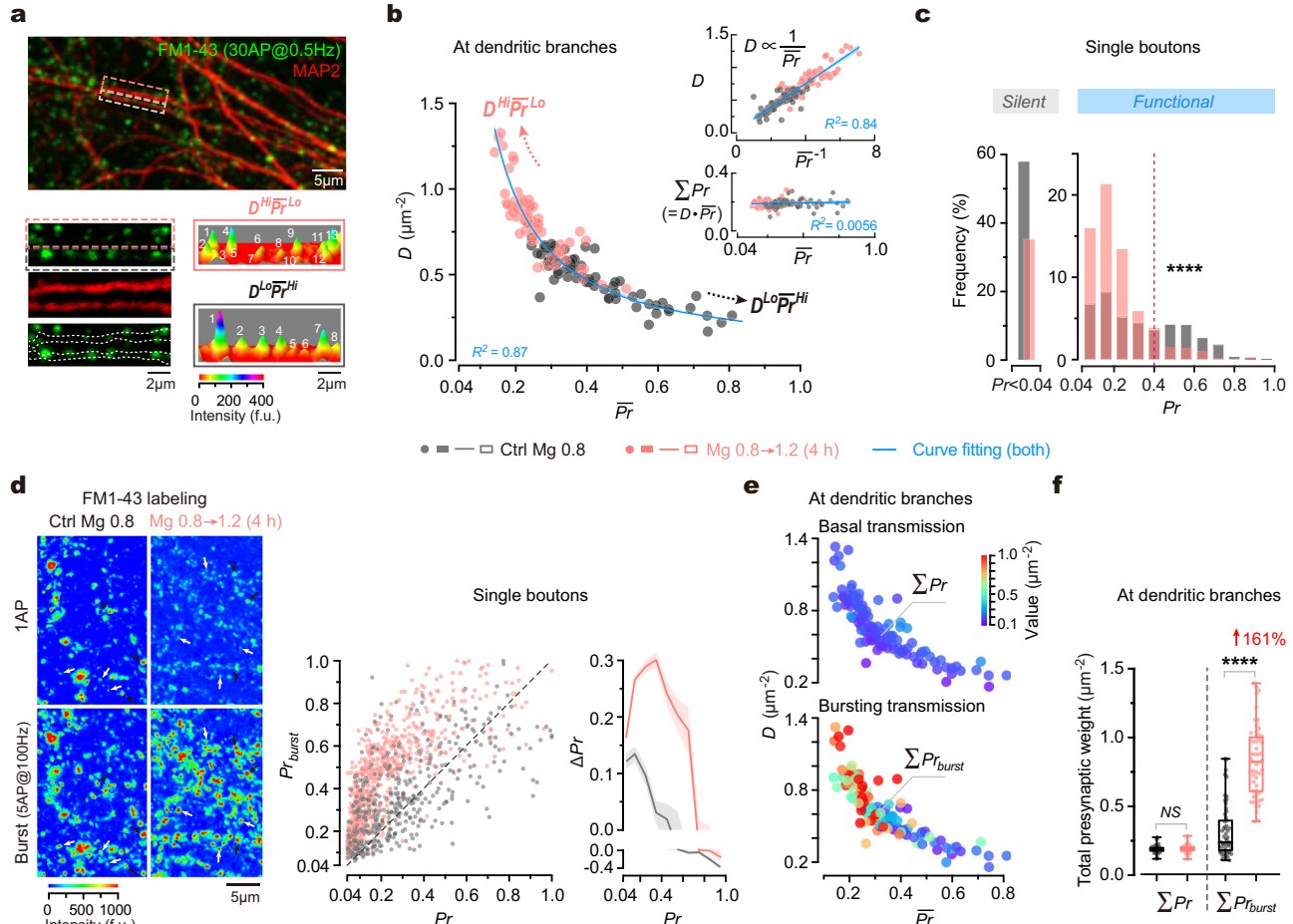

**Fig. 1 | Influence of synaptic configuration on presynaptic release at dendrites.**
**a** Top, an overlaid image of FM1-43$^+$ boutons and MAP2$^+$ dendrites. Bottom, magnification of boxes and 3D pseudo-color images to show FM1-43 fluorescence (indicative of $Pr$) of synapses. Punctum height, peak fluorescent intensity. **b** Plot of functional synapse density ($D$) against mean $Pr$ ($\overline{Pr}$) at distal dendritic branches ($n = 66, 67$ branches from 5, 5 biological repeats). Arrows, the trend towards $D^{Hi}\overline{Pr}^{Lo}$ or $D^{Lo}\overline{Pr}^{Hi}$ configuration. Inset, plots of $D$ against $1/\overline{Pr}$ (top) and total $Pr$ ($\Sigma Pr$) against $\overline{Pr}$ (bottom). Hyperbolic fitting, $R^2 = 0.87$, $P < 0.0001$. Linear regressions, $R^2 = 0.84$, 0.0056; $P < 0.0001$, $= 0.25$. Blue lines/error bands, fitted curves/95% confidence interval (CI). Biological repeats, individual coverslips of culture. **c** Shifted $Pr$ distribution ($n = 863, 885$ boutons from 5, 4 repeats; two-sided Kolmogorov–Smirnov test, ****$P < 0.0001$). Dashed line indicates where $Pr = 0.4$. **d** Left, confocal images to show FM1-43 staining elicited by single AP (1AP) inputs (30AP@0.5 Hz) and burst

inputs (6 repetitive trains of 5AP@100 Hz burst) in the same boutons. Black and white arrows, examples of high-$Pr$ and low-$Pr$ boutons. Middle, plot of the release probability upon bursts ($Pr_{burst}$) against $Pr$ in the same boutons ($n = 458, 763$ boutons from 3, 4 repeats). Dashed line, bisector. Right, $\Delta Pr$ ($= Pr_{burst} - Pr$) versus $Pr$. Curve/error band, mean/SEM. **e** Plots of synaptic configurations (indicated by $D$ - $\overline{Pr}$ combinations) against $\Sigma Pr$ and $\Sigma Pr_{burst}$ (in pseudo color) at individual dendritic branches ($n = 58, 53$ branches from 4, 4 repeats, some branches from **b**). **f** Box-whisker plot of $\Sigma Pr$ and $\Sigma Pr_{burst}$ at dendritic branches under physiological and elevated Mg$^{2+}$ conditions ($n = 58, 53$ branches from 4, 4 repeats; data from **e**). Two-sided Kolmogorov–Smirnov tests, $P = 0.0930$ ($NS$), ****$P < 0.0001$. Box borders/line, quartiles/median; whiskers, min and max. Source data are provided as a Source Data file.

systems[14–28], and branch-specific reorganization of synapses is crucially involved in learning and memory[29–32]. Moreover, as aging brains undergo pronounced alterations to cortical synaptic connectivity, hallmarked by both reduced synapse number and increased synaptic weight (and/or structural size)[33–37], gaining insights into this question would advance our understanding of synaptic correlates behind the progression of brain aging and might inspire new therapeutic strategies.

By examining synaptic connectivity, our study addresses an organization principle that orchestrates the synapses at hippocampal dendritic branches to generate diverse computational properties, including transmission efficiency, plasticity, and coding capacity. We reveal that the configuration of synaptic connectivity is a biomarker for branch-specific synaptic computations, and intracellular Mg$^{2+}$ is a crucial regulator of the configuration. Finally, we show evidence in aging brains and behaving animals supporting the major findings in vitro.

## Results

### Diversity of synaptic configuration across dendritic branches

To address the organization principle of the synapses at individual dendritic branches, we initially investigated the synaptic connectivity at distal dendrites of primary cultured hippocampal neurons. To measure presynaptic weight (indicated by vesicular release probability, $Pr$) of individual synapses, we utilized a fluorescent indicator of vesicle turnover, FM1-43, to visualize and quantify the released vesicles from individual presynaptic boutons in response to 30 single action potentials (APs) at a frequency of 0.5 Hz (Fig. 1a, Supplementary Fig. 1a; Methods). This approach allows for measuring $Pr$ at a minimum level of ~ 0.04, corresponding to one released vesicle per 30 APs. At distal dendritic branches ($0.62 \pm 0.21\,\mu m$ in diameter, mean ± SD), we observed significant variations in presynaptic weight distribution of dendritic branches, as indicated by diverse combinations of $D$ and mean $Pr$ ($\overline{Pr}$) (Fig. 1b). Notably, an inverse correlation between $D$ and $\overline{Pr}$ can be accurately described by a hyperbolic function, resulting in a

linear correlation between $D$ and $1/\overline{Pr}$ (Fig. 1b). We termed the pattern of synaptic connectivity at individual branches as 'synaptic configuration', denoted by the $D \sim \overline{Pr}$ combination. Despite continuous variations in $D \sim \overline{Pr}$ combination among branches, for illustrative purposes, we categorized two extreme patterns as follows: high-$D$ and low-$\overline{Pr}$ ($D^{Hi}\overline{Pr}^{Lo}$), or conversely, low-$D$ and high-$\overline{Pr}$ ($D^{Lo}\overline{Pr}^{Hi}$), facilitating the description of the synaptic configuration.

Under diverse synaptic configurations, the total presynaptic weight per unit area of dendrites (denoted as $\Sigma Pr$) remained constant (Fig. 1b, coefficient of variation of $\Sigma Pr$ was 0.14). To investigate whether $\Sigma Pr$ is always constant, we perturbed the synaptic configuration and examined its impact on $\Sigma Pr$. Previously, we discovered that chronic elevation of extracellular $Mg^{2+}$ concentration ($[Mg^{2+}]_o$) around a physiological range (0.8–1.2 mM) increases functional synapse density[38]. Here, we evaluated the effect of changing $[Mg^{2+}]_o$ on the synaptic configuration. By elevating $[Mg^{2+}]_o$ in the culture medium from 0.8 mM (corresponding to a typical 'physiological $Mg^{2+}$ condition' in the rodent brain) to 1.2 mM (corresponding to an 'elevated $Mg^{2+}$ condition' within physiological range) for 4 h (for rationales for $Mg^{2+}$ conditions, refer to Supplementary Notes), we observed both an increase in $D$ and a decrease in $\overline{Pr}$, resulting in a global shift of synaptic configurations towards the $D^{Hi}\overline{Pr}^{Lo}$ mode (Fig. 1b). Notably, throughout this perturbation, $\Sigma Pr$ remained constant across dendritic branches (Fig. 1b). These findings support the notion that the total presynaptic weight per unit area of dendrites remains constant, regardless of varying synaptic configurations.

To understand the influence of $[Mg^{2+}]_o$ on the synaptic configuration, we analyzed how the $Pr$ distribution of synapses was affected. Analyses revealed three apparent functional states of synapses based on the shape of $Pr$ distribution: silent ($Pr < 0.04$), low-$Pr$ (0.04–0.4), and high-$Pr$ (0.4–1.0) (Fig. 1c). Elevating $[Mg^{2+}]_o$ exerted a dual effect on the strength of vesicle release, reducing the proportion of silent presynaptic boutons (from 57.93% to 35.03%) and converting some high-$Pr$ boutons to low-$Pr$ ones. As a result, the percentage of low-$Pr$ boutons increased from 24.79% to 56.62%, whereas that of high-$Pr$ boutons decreased from 17.29% to 8.35% (Fig. 1c). These analyses suggest that during the configurational shift from $D^{Lo}\overline{Pr}^{Hi}$ to $D^{Hi}\overline{Pr}^{Lo}$ following the elevation of $[Mg^{2+}]_o$, the increase in $D$ was primarily attributed to the conversion of silent boutons into functional ones, while the decrease in $\overline{Pr}$ was due to the reduction of high-$Pr$ synapses (Fig. 1c).

In summary, the configuration of synaptic connectivity, as indicated by presynaptic weight distribution, varies across dendritic branches and can be modified by altering $Mg^{2+}$ levels. Notably, despite the substantial diversity in synaptic configuration, the total presynaptic weight of dendritic branches remains constant.

## Impact of synaptic configuration on presynaptic release at dendrites

We have shown that $\Sigma Pr$ remains constant during basal transmission. Given the critical role of bursting activity in synaptic computations during information processing[39–47], we further investigated how the total presynaptic weight during bursting transmission is influenced by the synaptic configuration. In the hippocampus, naturally-occurred bursts typically consist of 2–7 APs with inter-spike intervals of 1.5–6 ms under physiological conditions[48]. Of particular interest are theta bursts, a specific form of bursts that induce synaptic plasticity and are relevant to multiple cognitive functions[49–51]. Theta bursts typically consist of several trains of high-frequency APs occurring at a theta frequency. Here, we used 5 APs at 100 Hz (referred to as '5AP burst') to represent bursting activity.

To compare basal with bursting transmission, we measured $Pr$ and the release probability in bursting transmission ($Pr_{burst}$) in the same boutons sequentially (Fig. 1d, Methods). We observed a supralinear relationship between $Pr_{burst}$ and $Pr$, with the largest difference

($\Delta Pr$) occurring in the middle range of $Pr$ (~0.1–0.4) (Fig. 1d). When comparing $\Sigma Pr$ with the total presynaptic weight in bursting transmission ($\Sigma Pr_{burst}$) at the same dendritic branches, we found intriguingly that unlike the constant $\Sigma Pr$, the $\Sigma Pr_{burst}$ was significantly higher under $D^{Hi}\overline{Pr}^{Lo}$ configuration than under $D^{Lo}\overline{Pr}^{Hi}$ configuration (Fig. 1e). Furthermore, trials with a 4-h elevation of $[Mg^{2+}]_o$ from 0.8 to 1.2 mM significantly enhanced $\Sigma Pr_{burst}$ by 161.19% (Fig. 1f, from $0.32 \pm 0.19$ to $0.83 \pm 0.26$), showing a strong positive effect of $Mg^{2+}$ on presynaptic weights in bursting transmission. Similar phenomena were observed when examining other patterns of bursting activity (2–15 APs at 100 Hz per burst) (Supplementary Fig. 1b–f).

Together, synaptic configuration differentially impacts presynaptic release during basal and bursting transmission, with no effect in basal transmission but a strong regulatory effect in bursting transmission.

## Impact of synaptic configuration on postsynaptic AMPAR and PSD95 distributions

We next examined postsynaptic properties under varying synaptic configurations. During synaptic transmission, the postsynaptic weight of an excitatory synapse is primarily determined by its quantal size ($q$), which is influenced by three major factors: the amount of α-amino-3-hydroxy-5-methyl-4-isoxazolepropionic acid receptors (AMPARs) anchored to the postsynaptic density (PSD), AMPAR channel conductance, and AMPAR opening probability. Since AMPARs containing the GluA2 subunits (GluA2*AMPARs) at postsynaptic sites are crucial for maintaining and scaling basal synaptic transmission[52–55], we conducted immunostaining to detect the extracellular epitope of GluA2 and used its quantity at a postsynaptic site ([GluA2]) as an index to approximate the $q$. To ensure the GluA2 immunoreactive (GluA2$^+$) sites from real synapses, we first labeled functional presynaptic boutons using FM1-43 (to measure their $Pr$) before conducting post hoc immunostaining (Methods). The GluA2$^+$ sites juxtaposed with FM$^+$ puncta were considered to originate from real synapses (Fig. 2a).

At the single-synapse level, we observed a weak positive correlation ($R^2 = 0.34$) between [GluA2] and $Pr$ (Fig. 2a), demonstrating a coordination between $q$ and its corresponding presynaptic weight. This coordination suggests that the pattern of postsynaptic $q$ distribution would resemble the pattern of $Pr$ distribution at dendritic branches. Indeed, when quantifying the density of juxtaposed FM1-43$^+$GluA2$^+$ puncta ($D$), which reports the density of functional synapses, we found an inverse association between $D$ and the mean [GluA2] ($\overline{[GluA2]}$) of synapses along individual branches (Fig. 2b), similar to the inverse relationship between $D$ and $\overline{Pr}$ at presynaptic sites (Fig. 1b). This relationship can be well fitted by a hyperbolic function, resulting consequently in a linear correlation between $D$ and $1/\overline{[GluA2]}$ (Fig. 2b). Importantly, despite varying $D \sim \overline{[GluA2]}$ combinations, the total amount of postsynaptic GluA2*AMPARs per unit area of dendrites ($\Sigma[GluA2]$) remained constant across different branches (Fig. 2b). These findings suggest that the distribution pattern of $q$ correlates with the synaptic configuration ($D \sim \overline{Pr}$ combination). When using another biomarker postsynaptic density protein 95 (PSD95), a scaffolding protein closely associated with AMPARs[56–58], to estimate $q$ alternatively, we observed similar results as those using GluA2 (Fig. 2c, d). Moreover, when changing synaptic configurations by a 4-h elevation of $[Mg^{2+}]_o$, the distribution pattern of [GluA2] or [PSD95] was remarkably altered (Fig. 2a, c), and the combinations of $D \sim \overline{[GluA2]}$ and $D \sim \overline{[PSD95]}$ at dendritic branches significantly shifted, maintaining constant $\Sigma[GluA2]$ and $\Sigma[PSD95]$ (Fig. 2b, d).

Together, these findings suggest that synaptic configuration impacts the distribution pattern of postsynaptic proteins critical for excitatory synaptic transmission (representing $q$) while maintaining a pre-postsynaptic coordination.

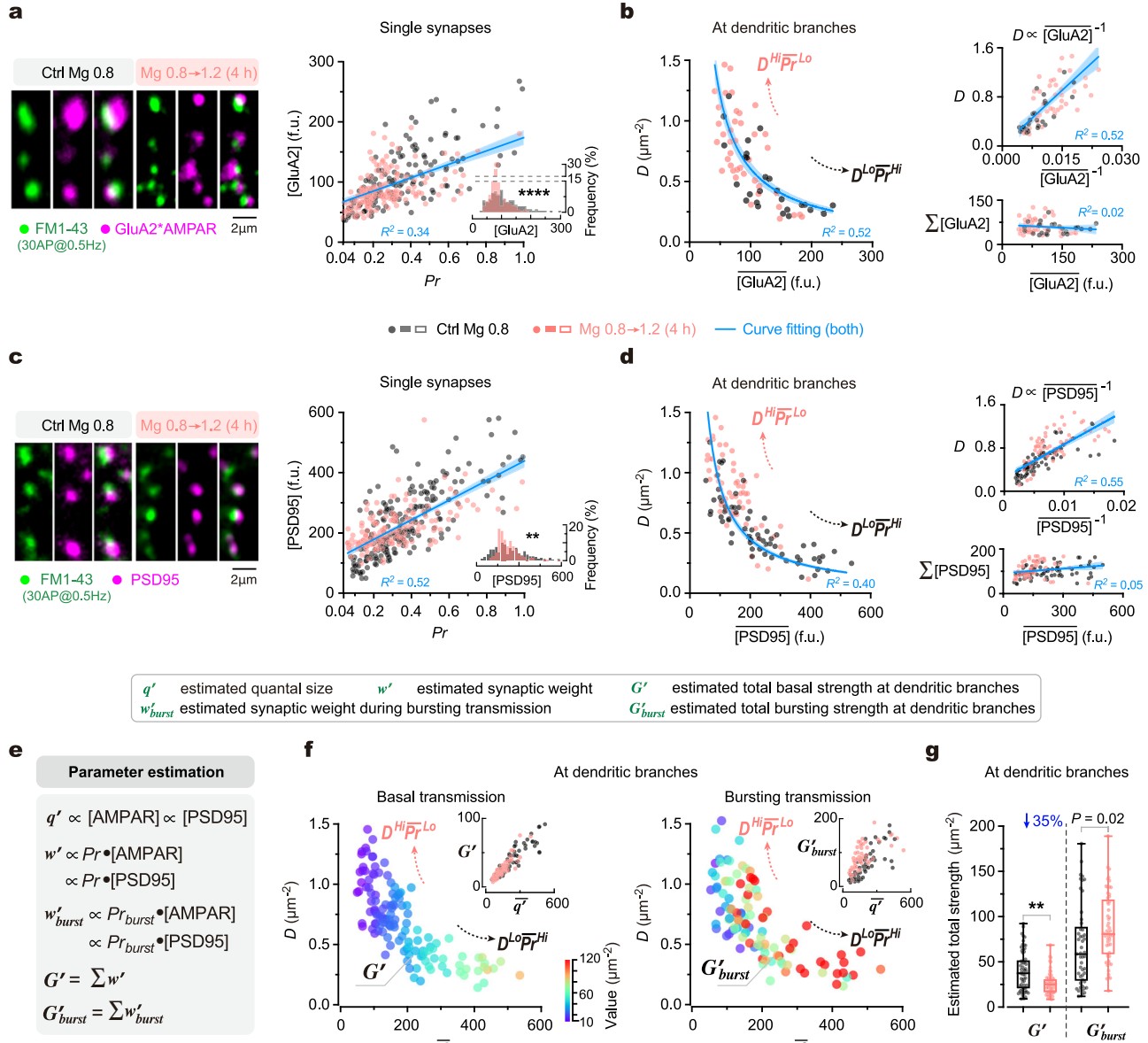

q′ estimated quantal size  w′ estimated synaptic weight  G′ estimated total basal strength at dendritic branches
w′_burst estimated synaptic weight during bursting transmission  G′_burst estimated total bursting strength at dendritic branches

**Fig. 2 | Influence of synaptic configuration on excitatory synaptic transmission in dendrites. a** Left, confocal images to show juxtaposed FM1-43⁺ and GluA2*AMPAR⁺ puncta at example dendritic segments. Right, synaptic [GluA2] against $Pr$ in single synapses ($n = 140$, 167 synapses from 3, 3 repeats; linear regression, $R^2 = 0.34$, $P < 0.0001$). Inset, shifted synaptic [GluA2] distribution (two-sided Kolmogorov–Smirnov test, ****$P < 0.0001$). **b** Left, plot of $D$ against the mean [GluA2] ($\overline{[GluA2]}$) of synapses at dendritic branches ($n = 26$, 42 branches from 3, 3 repeats; hyperbolic fitting, $R^2 = 0.52$). Top right, $D$ against 1/[GluA2] (linear regression, $R^2 = 0.55$, $P < 0.0001$). Bottom right, total [GluA2] ($\Sigma$[GluA2]) against $\overline{[GluA2]}$ (linear regression, $R^2 = 0.018$, $P = 0.28$). **c**, **d** Similar to (**a**, **b**), but immunostaining for PSD95 ($n = 210$, 163 synapses from 4, 3 repeats for **c**; 65, 62 branches from 4, 4 repeats for **d**). Two-sided Kolmogorov test (**c**, inset), **$P = 0.0019$. Linear regressions (**c**, **d**), $R^2 = 0.52$, 0.55, 0.046, $P < 0.0001$, <0.0001, = 0.0150. Hyperbolic fitting

(**d**, left), $R^2 = 0.40$. **e** Formulas for estimating quantal size ($q′$), synaptic weight ($w′$) and bursting synaptic weight ($w′_{burst}$) of single synapses, as well as total basal strength ($G′$) and total bursting strength ($G′_{burst}$) of dendritic branches. **f** The relationship between $D$, $\overline{q′}$ and $G′$ (left; $n = 65$, 62 branches from 4, 4 repeats) or $G′_{burst}$ (right; $n = 49$, 56 branches from 3, 3 repeats) (coded by pseudo color) at individual dendritic branches (data from separate measurements). Insets, $G′$-$\overline{q′}$ and $G′_{burst}$-$\overline{q′}$ associations. The $q′$ of individual synapses was estimated by postsynaptic [PSD95]. **g** Box-whisker plot of $G′$ and $G′_{burst}$ from physiological and elevated Mg²⁺ conditions ($n = 65$, 62, 49, 56 branches from 4, 4, 3, 3 repeats; the same data as in **f**). Box borders/line, quantiles/median; whiskers, min and max. Two-sided Kolmogorov–Smirnov tests, **$P = 0.0016$, *$P = 0.0162$. In (**a**–**d**), blue lines/error bands, fitted curves/95% CIs. Source data are provided as a Source Data file.

## Impact of synaptic configuration on transmission efficiency at dendrites

Next, we examined how the synaptic configuration affects the synaptic transmission efficiency of dendritic branches. In basal transmission, the synaptic transmission efficiency of an individual branch can be reported by the total strength of excitatory synaptic transmission (that is, the total AMPAR-mediated current) upon a single AP (denoted by $G$). To measure $G$, we first measured the synaptic weight of excitatory

transmission ($w$) in an individual synapse, equal to $Pr \cdot q$. To precisely measure $w$ of synapses distributed along dendrites, both $Pr$ and $q$ need to be determined in individual synapses, presently beyond the reach of current techniques. As a compromise, we directly measured $Pr$ while indirectly estimating $q$ by postsynaptic [GluA2] or [PSD95] in dendrites as shown above (Fig. 2a–d), then further estimated both $w$ of each synapse and the $G$ of all synapses within the dendritic branch, which is the sum of $w$ per unit area of dendrites (denoted by $\Sigma w$) (Fig. 2e,

Supplementary Fig. 2a–d). Hereafter, $q'$, $w'$ and $G'$ are used to denote the estimated $q$, $w$ and $G$, respectively. As mentioned above, the constant $\Sigma[GluA2]$ and $\Sigma[PSD95]$ at a dendritic branch suggests a constant $\Sigma q'$ (Fig. 2a–d). However, intriguingly, unlike the constant $\Sigma Pr$ and $\Sigma q'$, the $G'$ ($\Sigma w'$) was lower under the "high-$D$ and low-$\overline{q'}$" pattern than the "low-$D$ and high-$\overline{q'}$" pattern, meaning that $G'$ was lower under $D^{Hi}\overline{Pr}^{Lo}$ than $D^{Lo}\overline{Pr}^{Hi}$ configuration (Fig. 2f, Supplementary Fig. 2b–e). The total strength of synapses in bursting transmission ($G'_{burst}$, Fig. 2e) showed a weak trend being lower under the "high-$D$ and low-$\overline{q'}$" pattern than the "low-$D$ and high-$\overline{q'}$" pattern. However, this trend was not as uniform as $G'$ but instead highly variable in the medium range of $\overline{q'}$ (at 100–300) (Fig. 2f). Intriguingly, when we altered synaptic configurations by elevating $[Mg^{2+}]_o$, the general effect was to substantially suppress the average $G'$ of dendritic branches by 34.73% (from $39.51 \pm 21.54$ to $25.79 \pm 12.08$) (Fig. 2g).

Collectively, these data suggest that the synaptic configuration has a substantial impact on the transmission efficiency in basal transmission at individual dendritic branches. The transition of synaptic configurations from $D^{Lo}\overline{Pr}^{Hi}$ to $D^{Hi}\overline{Pr}^{Lo}$ decreases basal transmission efficiency.

### Impact of synaptic configuration on postsynaptic Ca²⁺ signals in dendrites

While the synaptic configuration affects transmission efficiency, how does it relate to synaptic plasticity at dendritic branches? To address this question, we first examined how synaptic configuration influences the distribution of postsynaptic Ca²⁺ signals that are essential for the generation of synaptic plasticity.

To start, using a genetically encoded Ca²⁺ indicator (GCaMP6f), we measured both $Pr$ and spine Ca²⁺ influx during basal transmission (that is, single AP-evoked Ca²⁺ influx through spines) at the same synapses (Methods) and found that they were positively correlated (Fig. 3a). Thus, spine Ca²⁺ influx of synapses may distribute along dendritic branches in a similar pattern as the $Pr$. To examine this possibility, we quantified functional synapse density $D$ and the Ca²⁺ entry through individual spines (denoted by $w_{Ca}$) (Fig. 3b). As expected, $D$ and the mean $w_{Ca}$ of spines ($\overline{w_{Ca}}$) were inversely correlated, resulting in constant total $w_{Ca}$ at per unit area of dendrites in basal transmission ($\Sigma w_{Ca}$) (Fig. 3c). Thus, the distribution of $w_{Ca}$ showed "high-$D$ and low-$\overline{w_{Ca}}$" and "low-$D$ and high-$\overline{w_{Ca}}$" patterns under the $D^{Hi}\overline{Pr}^{Lo}$ and $D^{Lo}\overline{Pr}^{Hi}$ configurations, respectively (similar to $q'$ distribution patterns, see Fig. 2b, d).

Given the crucial role of bursting activity in synaptic plasticity induction, we examined how synaptic configuration influences spine Ca²⁺ influx in bursting transmission. By measuring both the 1AP- and burst-evoked Ca²⁺ influx ($w_{Ca}$ and $w_{Caburst}$) in the same spines (Fig. 3b), we observed a supralinear-like association between $w_{Caburst}$ and $w_{Ca}$ (Fig. 3e), and consequently, medium-$w_{Ca}$ spines (~ 10–15 in value) showed larger $\Delta w_{Ca}$ (that is, $w_{Caburst}$ - $w_{Ca}$) than others (Fig. 3e, inset), similar to the result of $Pr$ and $Pr_{burst}$ (Fig. 1d). These findings suggest that medium-$w_{Ca}$ spines can better distinguish bursting activity from basal activity. At the branch levels, the total burst-induced Ca²⁺ influx ($\Sigma w_{Caburst}$) was higher under $D^{Hi}\overline{Pr}^{Lo}$ than $D^{Lo}\overline{Pr}^{Hi}$ configuration (Fig. 3f), alike presynaptic $\Sigma Pr_{burst}$ (Fig. 1f).

Moreover, when altering synaptic configurations by a 4-h elevation of $[Mg^{2+}]_o$ from 0.8 to 1.2 mM, we observed both increased $D$ and decreased $\overline{w_{Ca}}$ at dendritic branches, reshaping $w_{Ca}$ distribution towards the "high-$D$ and low-$\overline{w_{Ca}}$" pattern while maintaining constant $\Sigma w_{Ca}$ (Fig. 3c, f, g, from $8.09 \pm 1.23$ to $8.44 \pm 1.41$). During this configurational transition, a significant portion of spines got converted from inactive (no detectable Ca²⁺ influx) to active (Fig. 3d). Notably, like the $Pr$ redistribution (Fig. 1c), $w_{Ca}$ also underwent a similar redistribution, showing a remarkable reduction of both Ca²⁺-inactive spines and large-$w_{Ca}$ ones, as a result, the majority of spines became medium-$w_{Ca}$ ones (within the range of 0–15) (Fig. 3d). By contrast, during bursting

transmission, the $\Sigma w_{Caburst}$ of dendritic branches was significantly increased by 84.10% (Fig. 3f, g, Supplementary Fig. 3a, from $16.29 \pm 6.36$ to $29.99 \pm 9.55$), similar to the results of $\Sigma Pr$ and $\Sigma Pr_{burst}$ after such configurational transition (Fig. 1e, f).

Together, these results indicate that postsynaptic Ca²⁺ signals essential for synaptic plasticity induction are correlated with the synaptic configuration. The transition of synaptic configurations from $D^{Lo}\overline{Pr}^{Hi}$ to $D^{Hi}\overline{Pr}^{Lo}$ increases bursting Ca²⁺ influx in dendritic branches, suggestive of enhanced synaptic plasticity.

### Impact of synaptic configuration on postsynaptic GluN2B distribution

We proceeded to investigate the molecular basis of the observed properties of postsynaptic Ca²⁺ activity. Since vesicle release-induced spine Ca²⁺ influx is primarily mediated by *N*-methyl-D-aspartate receptors (NMDARs)[59,60], the observed spine Ca²⁺ influx (Fig. 3a–g) may be largely induced by the opening of postsynaptic NMDARs. Indeed, utilizing GluN2B subunit-specific pharmacological inhibition with Ro25-6981, we confirmed that GluN2B-containing (GluN2B*) NMDARs accounted for $87.12 \pm 3.59\%$ of 1AP-evoked Ca²⁺ entry (Fig. 3h, Methods). For a single synapse, its postsynaptic Ca²⁺ entry in synaptic transmission (that is, $w_{Ca}$) is determined by the product of $Pr$ and the quantal release-induced Ca²⁺ current through postsynaptic NMDARs (denoted by $q_{NMDAR}$), thus the ratio of $w_{Ca}/Pr$ should quantitatively approximate $q_{NMDAR}$ (denoted by $q'_{NMDAR}$). Notably, $q'_{NMDAR}$ was negatively associated with $Pr$ for individual synapses (Fig. 3i). During the configurational transition from $D^{Lo}\overline{Pr}^{Hi}$ to $D^{Hi}\overline{Pr}^{Lo}$ by elevating $[Mg^{2+}]_o$, $q'_{NMDAR}$ increased while $Pr$ decreased in single synapses, maintaining their inverse association (Fig. 3i). Meanwhile, this transition also caused a decrease in $\overline{Pr}$ and an increase in $\Sigma q'_{NMDAR}$ at individual dendritic branches (Supplementary Fig. 3b), suggesting higher expression levels of postsynaptic GluN2B*NMDARs.

To further verify the expression levels of GluN2B*NMDARs in dendrites, we directly examined the distribution of postsynaptic [GluN2B*NMDAR] of individual synapses along dendritic branches. To visualize postsynaptic GluN2B subunits, we transfected a green fluorescent protein-tagged GluN2B (GluN2B-GFP) vector[61] in conjunction with a red fluorescent indicator (mKate2) to label spine and dendrite morphology (Fig. 3j, Methods). The fluorescence ratio of GFP against mKate2 was used to estimate the concentration of transfected GluN2B subunits in spines ($[GluN2B]_{Sp}$) (Fig. 3j), which also approximates $q_{NMDAR}$. By measuring $Pr$ and $[GluN2B]_{Sp}$ of the same synapses, we found that they were inversely associated (Fig. 3j). Moreover, when shifting synaptic configurations from $D^{Lo}\overline{Pr}^{Hi}$ to $D^{Hi}\overline{Pr}^{Lo}$ by an elevation of $[Mg^{2+}]_o$, such an inverse association remained but $[GluN2B]_{Sp}$ increased (Fig. 3j), indicating that spine GluN2B levels in dendritic branches are higher under $D^{Hi}\overline{Pr}^{Lo}$ than $D^{Lo}\overline{Pr}^{Hi}$ configuration.

These findings (Fig. 3) suggest that the synaptic configuration determines synaptic plasticity of individual dendritic branches, given the crucial role of bursting Ca²⁺ signals through postsynaptic GluN2B*NMDARs for synaptic plasticity[62–65].

### Intracellular Mg²⁺ regulates synaptic configuration by inhibiting release probability

The above findings suggest that the synaptic configuration is a significant biomarker indicating the computational features of synapses at individual dendritic branches, including synaptic transmission efficiency and synaptic plasticity. Notably, chronically altering $[Mg^{2+}]_o$ can effectively modify synaptic configurations. We next explored the molecular mechanisms underlying the Mg²⁺-mediated regulation of synaptic configuration. Previously, we have shown that intracellular Mg²⁺ concentration ($[Mg^{2+}]_i$) determines the functional synapse density[38], inspiring us to hypothesize that the effect of $[Mg^{2+}]_o$ on the synaptic configuration is mediated by changes in $[Mg^{2+}]_i$. To test this

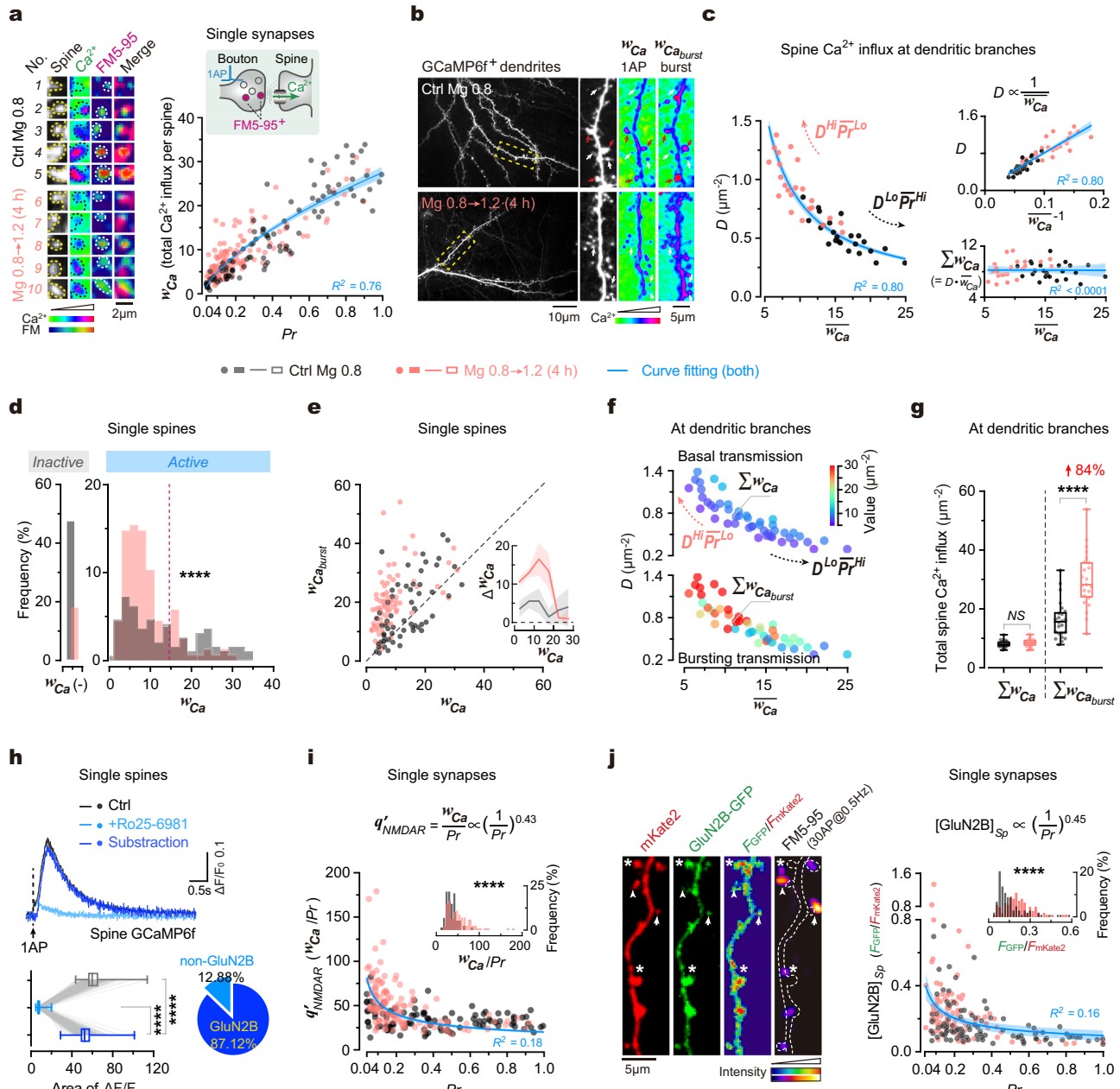

**Fig. 3 | Influence of synaptic configuration on postsynaptic Ca²⁺ signals in dendrites. a** Left, spine Ca²⁺ influx (GCaMP⁺) and $Pr$ (FM5-95⁺) of synapses. Right, integrated spine Ca²⁺ ($w_{Ca}$) against $Pr$ ($n = 101, 127$ synapses from 3, 3 repeats). Curve fitting, $w_{Ca} = 31.10 \cdot Pr^{0.55} - 3.95$ ($R^2 = 0.76$). **b** Left, GCaMP⁺ dendrites. Right, box magnification. Pseudo-color, integrated $\Delta F/F_0$. White/red arrows, low/high-Ca²⁺ spines. **c** Left, synapse density ($D$) against $\overline{w_{Ca}}$ in dendrites ($n = 24, 23$ branches from 3, 3 repeats; hyperbolic fitting, $R^2 = 0.80$). Top-right, $D$ versus $1/\overline{w_{Ca}}$ (linear regression, $R^2 = 0.81, P < 0.0001$). Bottom-right, total $w_{Ca}$ ($\Sigma w_{Ca}$) versus $\overline{w_{Ca}}$ (linear regression, $R^2 < 0.0001, P = 0.97$). **d** Shifted $w_{Ca}$ distribution ($n = 193, 156$ spines from 4, 3 repeats; two-sided Kolmogorov–Smirnov test, ****$P < 0.0001$). $w_{Ca}(-)$, no Ca²⁺ influx. Dashed line, $w_{Ca} = 15$. **e** Integrated burst-evoked Ca²⁺ ($w_{Caburst}$) against $w_{Ca}$ in single spines ($n = 59, 81$ spines from 3, 3 repeats). Dashed line, bisector. Inset, $\Delta w_{Ca}$ (=$w_{Caburst}-w_{Ca}$) against $w_{Ca}$. Curve/error band, mean/SEM. **f** Relationship between $D$, $\overline{w_{Ca}}$, $\Sigma w_{Ca}$ and $\Sigma w_{Caburst}$ at dendrites ($n = 24, 23$ branches from **c**). **g** Comparison of $\Sigma w_{Ca}$ and $\Sigma w_{Caburst}$ of individual branches ($n = 24, 23$ branches

from **f**; two-sided Kolmogorov–Smirnov tests, $P = 0.73$, ****$P < 0.0001$). **h** Top, evoked spine Ca²⁺ before and after GluN2B blockade. Traces, averaged 30 sweeps from 441 spines. Bottom-left, integrated Ca²⁺ ($n = 441$ spines from 7 repeats; two-sided Kolmogorov–Smirnov tests, ****$P < 0.0001$). Bottom-right, percentages. **i** Plot of $q'_{NMDAR}$ (=$w_{Ca}/Pr$) against $Pr$ ($n = 101, 127$ synapses from **a**). Inset, distribution (two-sided Kolmogorov–Smirnov test, ****$P < 0.0001$). Curve fitting, $w_{Ca}/Pr = 20.25 \cdot (1/Pr)^{0.43}$ ($R^2 = 0.18$). **j** Left, juxtaposed FM5-95⁺ boutons and GluN2B-GFP⁺ spines. Arrowheads/asterisks, high-$Pr$–low-[GluN2B] or low-$Pr$–high-[GluN2B] synapses. Right, spine [GluN2B] versus $Pr$ ($n = 102, 110$ synapses from 3, 3 repeats). Inset, [GluN2B]$_{Sp}$ distribution (two-sided Kolmogorov–Smirnov tests, ****$P < 0.0001$). Curve fitting, [GluN2B]$_{Sp} = 0.0987 \cdot (1/Pr)^{0.45}$ ($R^2 = 0.16$). In (**a**, **c**, **i**, **j**), blue lines and error bands, fitted curves and 95% CIs. In (**g**, **h**), box borders/line, quantiles/median; whiskers, min/max. Source data are provided as a Source Data file.

hypothesis, we utilized a chemical fluorescent probe called Magnesium Green acetoxymethyl ester (MgGrn) to visualize and quantify [Mg²⁺]ᵢ[38]. Interestingly, altering [Mg²⁺]ₒ did not immediately impact the synaptic configuration; however, it was followed by significant changes

in [Mg²⁺]ᵢ after 4–6 h (Fig. 4a). The temporal correlation between [Mg²⁺]ᵢ and the synaptic configuration suggests that [Mg²⁺]ᵢ may be responsible for mediating the changes in configuration observed. To further investigate this, we administered imipramine, a compound

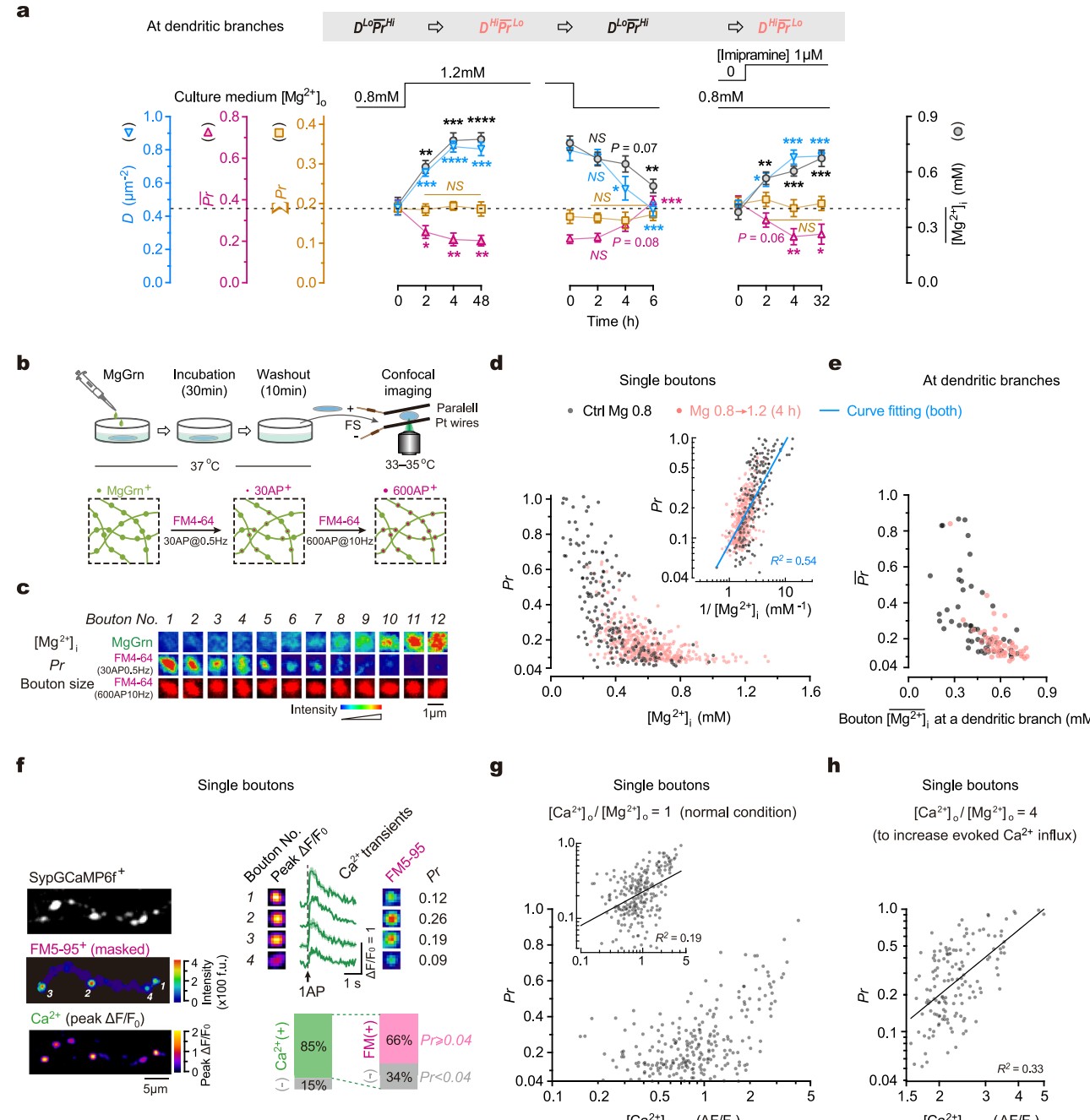

**Fig. 4 | Intracellular $Mg^{2+}$ regulates synaptic configuration by inhibiting release probability. a** Time-courses of changes in synaptic configuration and $\overline{[Mg^{2+}]}_i$ (mM) in neurites after various treatments ($n = 5, 8, 10, 11, 8, 8, 7, 7, 6, 7, 9, 8$ repeats for configuration, $5, 8, 9, 8, 9, 7, 7, 6, 6, 6, 6, 6$ repeats for Mg$^{2+}$). Two-sided unpaired $t$-tests (time 4 h $vs.$ 0 h). For $\Sigma Pr$, $P = 0.4068, 0.7444, 0.7391$; for $D$, $P < 0.0001, = 0.0305, 0.0003$; for $\overline{Pr}$, $P = 0.0084, 0.0799, 0.0044$; for $[Mg^{2+}]_i$, $P = 0.0001, 0.0724, 0.0009$. NS, no significance, *$P < 0.05$, **$P < 0.01$, ***$P < 0.001$, ****$P < 0.0001$. **b** Experimental design for measuring $[Mg^{2+}]_i$ and $Pr$ in the same boutons. **c** FM4-64$^+$ and MgGrn$^+$ signals in the same boutons. **d** Measurement of $Pr$ and $[Mg^{2+}]_i$ (mM) in boutons ($n = 204, 444$ from 3, 4 repeats). Inset, $Pr$ versus $1/[Mg^{2+}]_i$. Double logarithmic fitting, $R^2 = 0.54$, $P < 0.0001$. **e** Measurement of $\overline{Pr}$ and $\overline{[Mg^{2+}]}_i$ (mM) of the boutons at individual branches ($n = 55, 56$ branches from 4, 4 repeats). **f**–**h** Concurrent measurement of $Pr$ and 1AP-evoked Ca$^{2+}$ influx in the same boutons. **f** Left, images to show evoked Ca$^{2+}$ influx (peak $\Delta F/F_0$) and FM5-95$^+$ puncta (masked by GCaMP$^+$ axon morphology) in the same boutons. Top-right, peak $\Delta F/F_0$, Ca$^{2+}$ traces, FM5-95$^+$ puncta, and $Pr$ values in the same boutons. Bottom-right, percentage of boutons undergoing Ca$^{2+}$ influx, referred to as Ca$^{2+}$(+), and vesicle turnover ($Pr \geq 0.04$) ($n = 3$ repeats). $\Delta F/F_0$ puncta and Ca$^{2+}$ traces were averaged from 30 repeats. Band over traces, SEM. **g** Plot of $Pr$ against $[Ca^{2+}]_{evoke}$ (the amplitude of 1AP-evoked Ca$^{2+}$ influx) in boutons ($n = 243$ boutons from 5 repeats, physiological Mg$^{2+}$ condition). Inset, logarithmic plot. Line, double logarithmic fitting, $R^2 = 0.19$, $P < 0.0001$. **h** Similar to (**g**) but using working solution containing $[Ca^{2+}]_o$ 4.8 mM and $[Mg^{2+}]_o$ 1.2 mM (ratio 4:1) ($n = 146$ boutons from 3 repeats). Line, double logarithmic fitting, $R^2 = 0.33$, $P < 0.0001$. Data are mean ± SEM. Source data are provided as a Source Data file.

known to increase $[Mg^{2+}]_i$ by regulating Na$^+$/Mg$^{2+}$ antiporters[66], allowing us to test the effect of increased $[Mg^{2+}]_i$ on synaptic configurations without altering $[Mg^{2+}]_o$. As expected, imipramine induced changes in the synaptic configuration accompanied by an elevation of $[Mg^{2+}]_i$

(Fig. 4a). These findings suggest that $[Mg^{2+}]_i$ plays a crucial role in controlling the synaptic configuration.

To elucidate how $[Mg^{2+}]_i$ regulates the synaptic configuration (that is, the $D \sim \overline{Pr}$ combination), we first examined the relationship

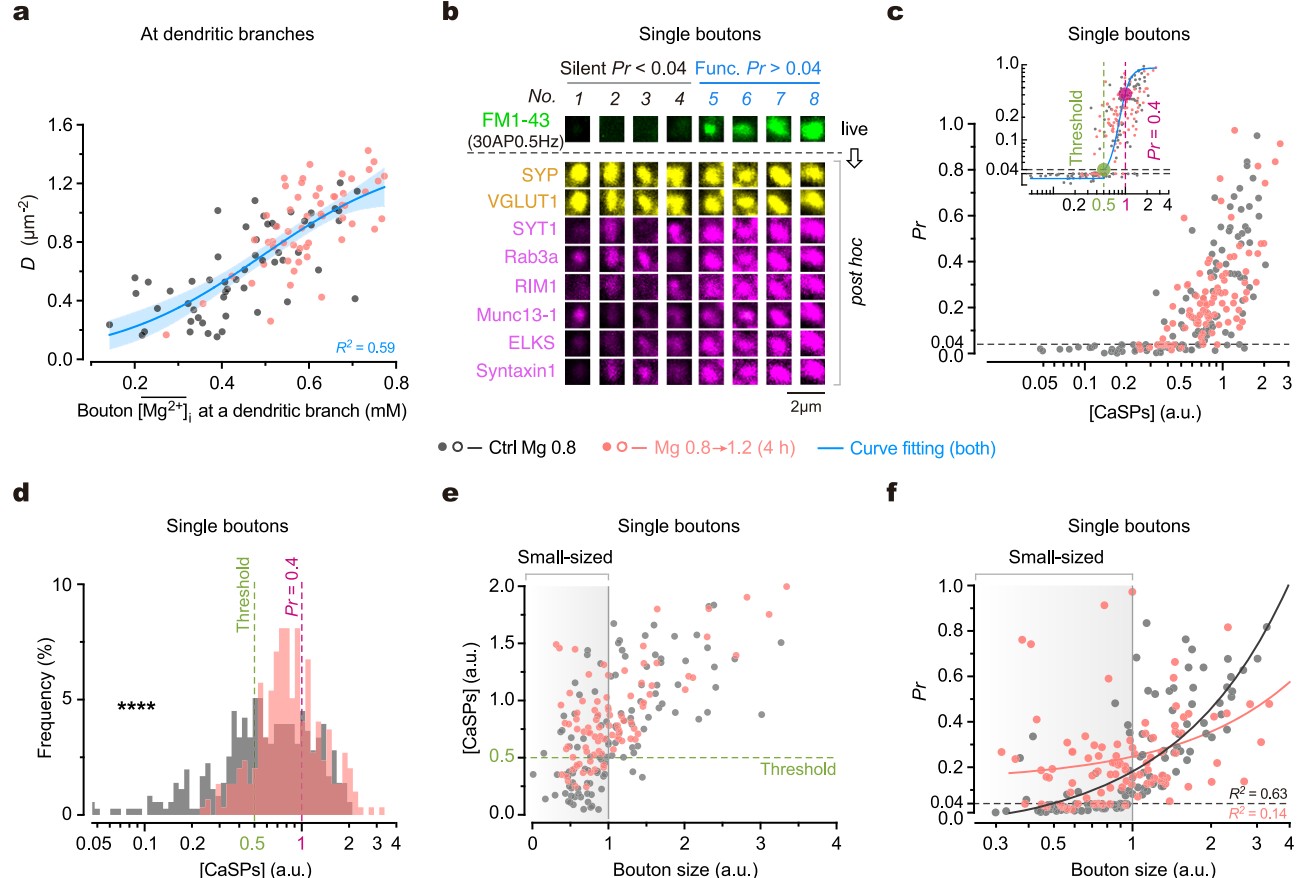

**Fig. 5 | Intracellular Mg²⁺ regulates synaptic configuration by converting small boutons from silent to functional. a** Plot of $D$ (measured under the 30AP@0.5 Hz stimulation) against bouton $\overline{[Mg^{2+}]_i}$ at dendritic branches ($n = 55$, 56 branches from 4, 4 repeats; the same experiment as in Fig. 4e). Logistic growth curve fitting, $R^2 = 0.59$. Blue line, fitted curve. Error band, 95% CI. **b** Confocal images to show FM1-43 (indicative of $Pr$) and *post hoc* immunostaining of various proteins in the same boutons (see also Supplementary Fig. 6). **c** Plot of $Pr$ against [CaSPs] in the same boutons ($n = 121$, 94 boutons from 4, 3 repeats). Inset, double logarithmic plot. Boltzmann-Sigmoidal curve fitting, $R^2 = 0.44$. Dashed lines and large dots, the locations on fitted curve where $Pr = 0.04$ (threshold) and 0.4. **d** Shifted distribution of [CaSPs] in boutons ($n = 345$, 288 boutons from 8, 6 repeats; two-sided Kolmogorov–Smirnov test, ****$P < 0.0001$). Dashed lines, [CaSPs] values when $Pr = 0.04$ (threshold) and 0.4 based on (**c** inset). **e** Plot of [CaSPs] against estimated bouton size (estimated by median-normalized [VGLUT1]) in single boutons ($n = 121$, 94 boutons from 4, 3 repeats). Dashed line, threshold of [CaSPs]. Shade, small-sized boutons (size < 1). **f** Plot of $Pr$ against bouton size ($n = 121$, 94 boutons from 4, 3 repeats). Note the marked increase in $Pr$ of small-sized boutons after a 4-h elevation of $[Mg^{2+}]_o$. Lines, linear regressions ($R^2 = 0.63$, 0.14). Shade, small-sized boutons (size < 1). a.u. in (**c**–**f**), arbitrary unit. Source data are provided as a Source Data file.

between $[Mg^{2+}]_i$ and $Pr$ in the same presynaptic boutons (Fig. 4b; Methods). Under a physiological $Mg^{2+}$ condition ($[Mg^{2+}]_o$ 0.8 mM), $[Mg^{2+}]_i$ exhibited remarkable variations among boutons (Fig. 4c, Supplementary Fig. 4). Strikingly, $Pr$ was inversely correlated with $[Mg^{2+}]_i$ in individual boutons (Fig. 4c, d). When elevating $[Mg^{2+}]_o$ from 0.8 to 1.2 mM for 4 h, mean $[Mg^{2+}]_i$ ($\overline{[Mg^{2+}]_i}$) of boutons increased from 396.34 ± 205.12 to 597.53 ± 175.41 μM (mean ± SD) and the distribution of $[Mg^{2+}]_i$ in boutons was shifted; concurrently, the $Pr$ distribution was significantly shifted, but the inverse relationship remained (Fig. 4d). These results demonstrate a negative correlation between $Pr$ and $[Mg^{2+}]_i$ in single boutons. In addition, we further confirmed at individual dendritic branches that $\overline{Pr}$ and $\overline{[Mg^{2+}]_i}$ of the boutons were negatively associated (Fig. 4e).

We noticed that this observation differs from what is observed in large synapses, such as neuromuscular junctions (NMJs) and the Calyx of Held synapses, where presynaptic $Pr$ is dominantly controlled by single AP-evoked $[Ca^{2+}]_i$ (reviewed in[67]). To compare, we concurrently measured $Pr$ and single AP-evoked $Ca^{2+}$ influx in the same boutons using synaptophysin-fused GCaMP6f (SypGCaMP6f) for detecting presynaptic $Ca^{2+}$ activity[68,69] and FM5-95, a red fluorescence FM dye, for measuring $Pr$ (Fig. 4f, Supplementary Fig. 5). We observed evoked $Ca^{2+}$ events in 85% of SypGCaMP6f⁺ boutons and

66% of these boutons underwent vesicle release (Fig. 4f). Generally, there was a positive association between $Pr$ and the amplitude of evoked $Ca^{2+}$ influx, but this association diminished at the lower range of $Ca^{2+}$ influx (Fig. 4g). This observation raises a possibility that the association between $Pr$ and $[Ca^{2+}]_i$ would be stronger when AP-evoked $[Ca^{2+}]_i$ is sufficiently high. Indeed, when single AP-evoked $[Ca^{2+}]_i$ was remarkably elevated by an acute increase in the ratio of extracellular $[Ca^{2+}]_o/[Mg^{2+}]_o$ from 1 to 4, the $Ca^{2+}$ dependency of $Pr$ became significantly improved (Fig. 4h).

Taken together, $[Mg^{2+}]_i$ exerts a significant negative effect on $Pr$ in small hippocampal synapses.

### Intracellular Mg²⁺ regulates configuration by increasing functional synapse density

Next, we investigated how intracellular $Mg^{2+}$ regulates the other parameter of synaptic configuration, functional synapse density $D$. To explore this, we examined the correlation between $D$ and the bouton $[Mg^{2+}]_i$ of synapses within individual dendritic branches. Our findings revealed a positive association between them, and a 4-h elevation of $[Mg^{2+}]_o$ concurrently increased both $D$ and $\overline{[Mg^{2+}]_i}$ (Fig. 5a). These results suggest that local $[Mg^{2+}]_i$ positively correlates with $D$ in a branch-specific manner. To further understand the underlying

molecular mechanisms, we investigated the expression levels of pre-synaptic $Ca^{2+}$-sensitivity-related proteins (CaSPs)[70,71], which have been shown in our previous study to be involved in the conversion of silent boutons to functional ones through elevated $[Mg^{2+}]_i$[38]. We measured both $Pr$ and the expression level of individual CaSPs within the same boutons (Fig. 5b, Supplementary Fig. 6). Consistently, we observed a positive correlation between $Pr$ and the total expression level of CaSPs ([CaSPs], a normalized value, refer to Methods) in individual boutons (Fig. 5c). Interestingly, there seemed to be an empirical threshold of [CaSPs] for functional boutons, as most boutons remained silent when [CaSPs] was below 0.5 (Fig. 5c, inset). Importantly, increasing $[Mg^{2+}]_i$ by elevating $[Mg^{2+}]_o$ from 0.8 to 1.2 mM for 4 h significantly increased [CaSPs] in silent boutons, converting them into functional ones (Fig. 5c, d). Thus, the upregulation of $D$ by elevated $[Mg^{2+}]_i$ is primarily attributed to the supplementation of CaSPs in originally silent boutons lacking these proteins.

Considering that the abundance of presynaptic proteins is correlated with the structural size of boutons[72,73], where smaller boutons are more likely to lack CaSPs and remain silent, we used the vesicle protein VGLUT1 as a marker of bouton size (Methods) to examine its association with [CaSPs] and $Pr$ in the same boutons. Our results demonstrated a positive association for both [CaSPs] and $Pr$ with bouton size (Fig. 5e, f). Remarkably, increasing $[Mg^{2+}]_i$ substantially augmented the presence of CaSPs in small-sized boutons, resulting in the conversion of a significant portion of them from silent to functional (Fig. 5e, f).

Taken together (Figs. 4, 5), these findings demonstrate that intracellular $Mg^{2+}$ plays a dual role in regulating the synaptic configuration. Firstly, it exerts an ongoing negative effect on vesicle release (to reduce $Pr$). Secondly, over a period of hours, elevated $[Mg^{2+}]_i$ gradually increases the concentration of presynaptic CaSPs, enabling the conversion of a substantial number of small-sized, silent boutons into functional ones (to increase $D$).

## Multiple pathways converge on intracellular $Mg^{2+}$ in regulating synaptic configuration

The results presented above demonstrate that intracellular $Mg^{2+}$ plays a significant role in regulating synaptic configuration. To further understand the influence of other major signaling pathways on synaptic configuration, we investigated the impact of $Ca^{2+}$ signaling, neurotrophic factors (such as BDNF), neuroinflammatory cytokines (such as TNF-α), and cAMP/PKA signaling. For each pathway (excluding BDNF), we first examined whether reducing the background level of signaling affected synaptic configuration. If a positive effect was observed, we then investigated whether moderately upregulating the signaling pathway caused a shift in synaptic configuration in the opposite direction. This approach ensured that each identified signaling pathway had physiological relevance.

We initially examined the influence of basal presynaptic $[Ca^{2+}]_i$ (that is, the presynaptic $[Ca^{2+}]_i$ at resting condition) on synaptic configuration, as it is known to affect $Pr$[74]. Bath application of the NMDAR blocker AP5 (20 μM) or ifenprodil (1 μM) reduced basal $[Ca^{2+}]_i$ by ~20% after 10 min (Supplementary Fig. 7a, b). This extent of basal $[Ca^{2+}]_i$ alteration should not affect $Pr$ significantly[74]. Indeed, the $Pr$ was not significantly changed following 10 min of NMDAR blockade (Supplementary Fig. 7c). However, a chronic suppression of basal $[Ca^{2+}]_i$ for 4 h resulted in changes in synaptic configuration towards $D^{Hi}\overline{Pr}^{Lo}$ (Fig. 6a, b). Conversely, we examined the effect of bath application of glutamate, which could increase basal presynaptic $[Ca^{2+}]_i$ directly through activation of presynaptic glutamate receptors and subsequently influence $Pr$[75]. We found that application of ambient glutamate (5 μM) increased basal presynaptic $[Ca^{2+}]_i$ by ~20% after 10 min (Supplementary Fig. 7a, b) but this moderate increase did not change $Pr$ acutely (Supplementary Fig. 7c). However, a 4-h administration of ambient glutamate at the same concentration significantly shifted the

synaptic configuration towards $D^{Lo}\overline{Pr}^{Hi}$ (Fig. 6a, b). In both perturbations, $\Sigma Pr$ of dendrites remained constant across branches. These findings suggest that chronic alterations in basal presynaptic $[Ca^{2+}]_i$ can bidirectionally modify synaptic configurations.

We next investigated the influence of TNF-α signaling on synaptic configuration. TNF-α signaling is involved in the homeostatic regulation of synaptic transmission[76], making it a potential candidate for regulating synaptic configuration. We found that chronic inhibition of basal TNF-α signaling by administering recombinant soluble TNF receptor 1 (sTNFR1, 10 pg ml⁻¹) shifted the synaptic configuration towards $D^{Hi}\overline{Pr}^{Lo}$ (Fig. 6c). In contrast, a 4-h moderate elevation of TNF-α concentration (100 pg ml⁻¹) shifted the synaptic configuration towards $D^{Lo}\overline{Pr}^{Hi}$ (Fig. 6c). In both perturbations, $\Sigma Pr$ of dendrites remained constant across branches. These results suggest that chronic manipulations of endogenous TNF-α signaling around the basal level can modify the synaptic configuration.

Consistent results were obtained when perturbating cAMP/PKA or BDNF signaling around basal levels. Inhibiting cAMP/PKA or increasing [BDNF] shifted synaptic configuration towards $D^{Hi}\overline{Pr}^{Lo}$, while activating cAMP/PKA shifted the configuration towards $D^{Lo}\overline{Pr}^{Hi}$ (Fig. 6c). Overall, these pharmacological experiments demonstrate that under physiological conditions, the synaptic configuration can be regulated by multiple signaling pathways.

An intriguing observation was the concurrent influence of these pathways on the mean intracellular $Mg^{2+}$ concentration ($[Mg^{2+}]_i$) (Fig. 6a, c, d). This coincidence suggests two possibilities: either these pathways regulate synaptic configuration and $[Mg^{2+}]_i$ in parallel, or their influence on synaptic configuration is mediated by changes in $[Mg^{2+}]_i$. Our data support the latter hypothesis, as the changes of synaptic configuration induced by perturbations of ambient glutamate, TNF-α, or 8-Br-cAMP were counteracted by preventing the decrease in $[Mg^{2+}]_i$ through the addition of $[Mg^{2+}]_o$ (+0.4 mM) or imipramine (1 μM) (Fig. 6c). Lastly, we pooled the data of synaptic configurations ($D \sim \overline{Pr}$ combinations) and their corresponding $[Mg^{2+}]_i$ values from various experimental conditions (from Figs. 4a, 6a, c, d) and found a linear correlation for $D$ and $\overline{Pr}$ with $[Mg^{2+}]_i$ (Fig. 6e). These findings collectively suggest that intracellular $Mg^{2+}$ is a crucial regulator of synaptic configuration under physiological conditions.

## Elevating brain $Mg^{2+}$ levels modifies hippocampal synaptic configuration

The above findings demonstrate a regulatory scheme of $[Mg^{2+}]_i \rightarrow$ configuration → transmission/plasticity of synapses at dendritic branches in cultured hippocampal neurons. However, it remains unclear if $Mg^{2+}$ plays a similar role in intact animals and whether elevating brain $Mg^{2+}$ levels can change synaptic configurations in vivo.

To address these questions, it is crucial to identify a reliable indicator for the synaptic configuration in vivo. In our in vitro research, we employed $D \sim \overline{Pr}$ combinations as simplified representation of diverse synaptic configurations. The configurational transition from $D^{Lo}\overline{Pr}^{Hi}$ to $D^{Hi}\overline{Pr}^{Lo}$ induced by elevated $[Mg^{2+}]_i$ involves two concurrent processes of synaptic weight redistribution: the conversion from silent to functional synapses and the shift from high-$Pr$ to low-$Pr$ synapses (Fig. 1c). Consequently, the distribution of synaptic weights undergoes a transition from a wide and long-tailed shape to a narrower distribution. Therefore, the changes in the shape of synaptic weight distribution can be taken as an in vivo indicator of synaptic configuration.

Measuring synaptic weights directly in the hippocampus of living animals is difficult due to technical limitations. As an alternative, we utilized indirect markers to infer synaptic weights. Synaptic weight strongly correlates with multiple parameters that report the structural size of individual synapses[77-82], including the presynaptic active zone (AZ) area (a robust $Pr$ indicator), the postsynaptic density (PSD) area, and the spine head volume. Thus, both the AZ/PSD area and the spine head volume serve as reliable indicators of synaptic weights. Here, we

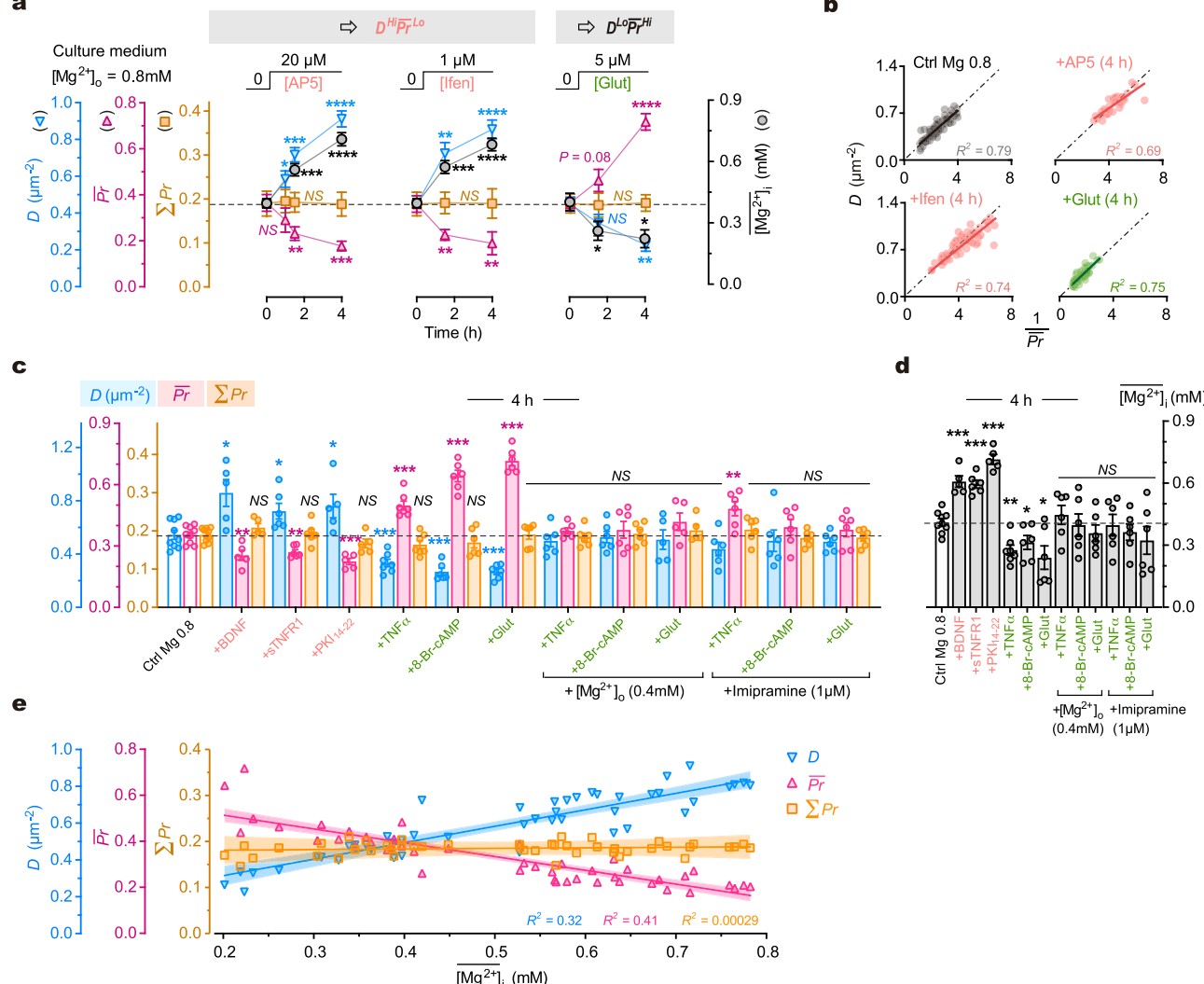

**Fig. 6 | Multiple pathways converge on intracellular Mg²⁺ levels in regulating synaptic configuration. a** Time courses of changes in synaptic configuration and $[Mg^{2+}]_i$ (mM) after various treatments (for timepoints from left to right, $n = 6, 7, 8,$ 8, 6, 5, 6, 5, 7, 6 repeats for configuration measurements, 8, 7, 7, 9, 6, 8, 8, 8, 8 repeats for $[Mg^{2+}]_i$ measurements). Two-sided unpaired $t$-tests (time 4 h *vs.* 0 h). For $\Sigma Pr$, $P = 0.8979, 0.9945, 0.9370$; for **D**, $P = 0.0334, < 0.0001, = 0.0015$; for $\overline{Pr}$, $P = 0.0003, 0.0093, < 0.0001$; for $[Mg^{2+}]_i$, $P < 0.0001, < 0.0001, = 0.0109$. **b** Plots of **D** against $1/\overline{Pr}$ at dendritic branches 4 h after various treatments ($n = 81, 42, 64, 57$ branches from 7, 5, 6, 5 for control, AP5, ifenprodil and glutamate, respectively). Solid lines, linear regressions ($P < 0.0001$ for all). Dash-dot lines, bisectors. **c, d** Measurements of **D**, $\overline{Pr}$, $\Sigma Pr$ and $[Mg^{2+}]_i$ 4 h after various treatments (left to

right, $n = 9, 5, 6, 5, 7, 6, 6, 6, 6, 5, 6, 6, 6$ repeats). Two-sided Mann-Whitney tests (*vs.* control). For **D**, $P = 0.0190, 0.0360, 0.0420, 0.0007, 0.0004, 0.0004, 0.4559,$ $0.8639, 0.6993, 0.1135, 0.3277, 0.4559$; for $\overline{Pr}$, $P = 0.0070, 0.0048, 0.0010, 0.0002,$ $0.0004, 0.0004, 0.6070, 0.8639, 0.2977, 0.0076, 0.3277, 0.5287$; for $[Mg^{2+}]_i$, $P = 0.0010, 0.0004, 0.0010, 0.0021, 0.0360, 0.0256, \underline{0.4559}, 0.7756, 0.2238, >$ $0.9999, 0.3277, 0.3884$. **e** Plot of **D**, $\overline{Pr}$ and $\Sigma Pr$ against $[Mg^{2+}]_i$ at the same time-points after various treatments ($n = 46$ various conditions from **a, c, d** and Fig. 4a; linear regressions, $P < 0.0001, < 0.0001, = 0.79$). Lines, fitted curves. Error bands, 95% CIs. Data are presented as mean ± SEM. *NS*, no significance, *$P < 0.05$, **$P < 0.01$, ***$P < 0.001$, ****$P < 0.0001$. Source data are provided as a Source Data file.

quantified the PSD area and spine head volume to infer synaptic weight distribution in the hippocampus of animals. Notably, considering that silent and low-weight synapses are typically small in structural size (Fig. 5e, f), we expected to observe a subset of putative nonfunctional synapses with extremely small PSD area and spine head volume.

To modulate synaptic configuration in the hippocampus of living brains, we developed a Mg²⁺ compound named magnesium L-threonate (MgT), which effectively elevates brain Mg²⁺ levels through daily oral intake[83]. We aimed to compare the distribution of synaptic weights between animals with or without MgT treatment. As a baseline, we examined the PSD area and spine head volume of randomly selected spiny synapses in the CA1 stratum radiatum (s.r.) of hippocampus in aged rats (28 months of age) (Fig. 7a). We performed 3D reconstruction of synapses using serial scanning electron microscopy images (Fig. 7b, c). Remarkably, both parameters exhibited a

nonuniform and bimodal-like distribution, with synapses predominantly falling into either the large or small size categories, while medium-sized synapses were scarce (Fig. 7d, e, Supplementary Fig. 8a). Importantly, this long-tailed distribution pattern resembled the distribution pattern observed under the $D^{Lo}\overline{Pr}^{Hi}$ configuration in vitro (Figs. 1–3). Subsequently, we investigated the influence of elevated brain Mg²⁺ levels on the distribution of structural size of synapses. Age-matched rats were subjected to daily oral Mg²⁺ supplementation from 12 to 28 months of age by drinking water containing MgT (Fig. 7a). Interestingly, the distribution pattern of both PSD area and spine head volume significantly changed in MgT-treated animals, with a reduction in the number of both small and large synapses, and concurrently, an increase in medium-sized synapses (Fig. 7d, e). These observations suggest a significant modification of synaptic configuration in the hippocampus after elevating brain Mg²⁺ levels.

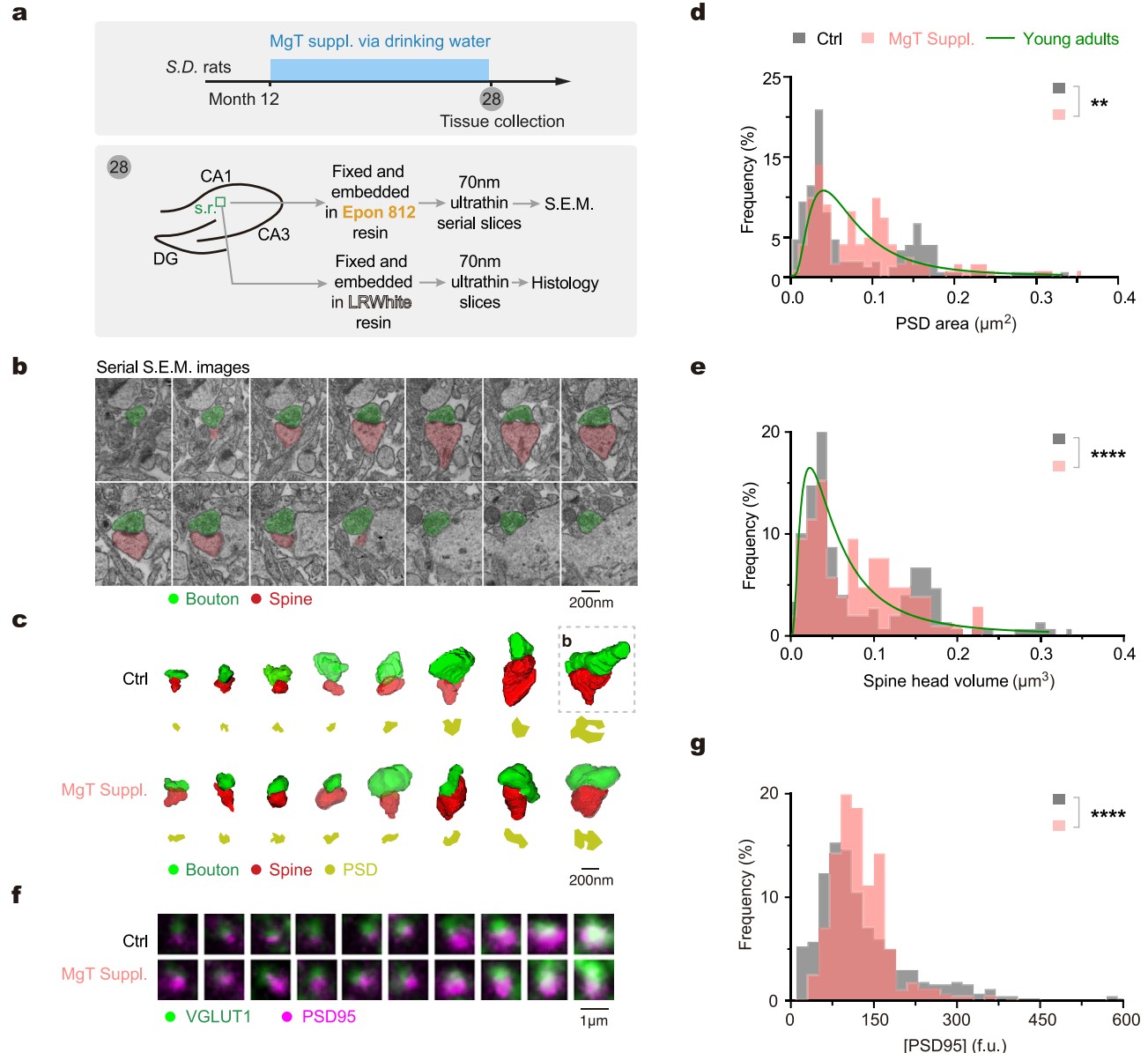

**Fig. 7 | Brain Mg²⁺ supplementation modifies synaptic configuration in the hippocampus. a** Experimental design. Tissue samples were collected from CA1 stratum radiatum (s.r.) of the hippocampus for serial scanning electron microscopy (S.E.M.) and immunohistology. Note these animals were tested for learning and memory by water maze before tissue collection (see Supplementary Fig. 10). **b** An example of a spiny synapse from serial S.E.M. images at a 70-nm thickness (*n* = 7 rats, the same as in **c**–**e**). **c** Examples of 3D-reconstructed synapses from rats of control and MgT-treated. Note the difference in PSD area (yellow) and spine head volume (red) between the two groups. Box, the synapse shown in (**b**). **d**, **e** Distribution of PSD area (**d**) and spine head volume (**e**) (*n* = 165, 187 synapses

from 3, 4 rats for control and MgT groups; two-sided Kolmogorov–Smirnov tests, **P = 0.0064, ****P < 0.0001). Green lines, distributions of PSD area and spine head volume in young adult animals (data from refs. 118,119). **f** Examples of synapses (juxtaposed contacts) labeled by PSD95 and VGLUT1 on 70-nm ultrathin slices. VGLUT1 was used to identify presynaptic contacts. Note the difference in PSD95⁺ puncta between the two groups. **g** Distribution of [PSD95] in single synaptic contacts from control and MgT-treated rats (*n* = 438, 682 synapses from 3, 4 rats; two-sided Kolmogorov–Smirnov test, ****P < 0.0001). Source data are provided as a Source Data file.

To assess functional implications of the above ultrastructural changes, we further measured [PSD95] in individual synapses as a functional biomarker. Immunostaining on 70-nm tissue slices from the same hippocampal region of the same rats as described above (Fig. 7a) allowed us to determine the levels of [PSD95] in synapses, while VGLUT1 co-staining facilitated the identification of presynaptic contacts (Fig. 7f). Consistently, MgT treatment caused a reduced percentage of both low- and high-[PSD95] synapses, along with an increased percentage of medium-[PSD95] synapses. Consequently, these changes narrowed the distribution of [PSD95] to the medium range (Fig. 7g), aligning with the in vitro observations (Supplementary

Fig. 8b–d). These findings further suggest that synaptic configuration in the hippocampus was modified following an elevation of brain Mg²⁺ levels.

The above data provide evidence for configurational changes induced by elevating brain Mg²⁺ levels, primarily based on the postsynaptic sites in dendrites. We further examined the critical presynaptic parameter, [CaSPs], that influences presynaptic functional state and consequently synaptic connectivity as shown above (Fig. 5). We observed an aging-related decline in [CaSPs] in the hippocampus and found that treatment with MgT to elevate brain Mg²⁺ levels prevented this decline (Supplementary Fig. 9). This evidence consistently

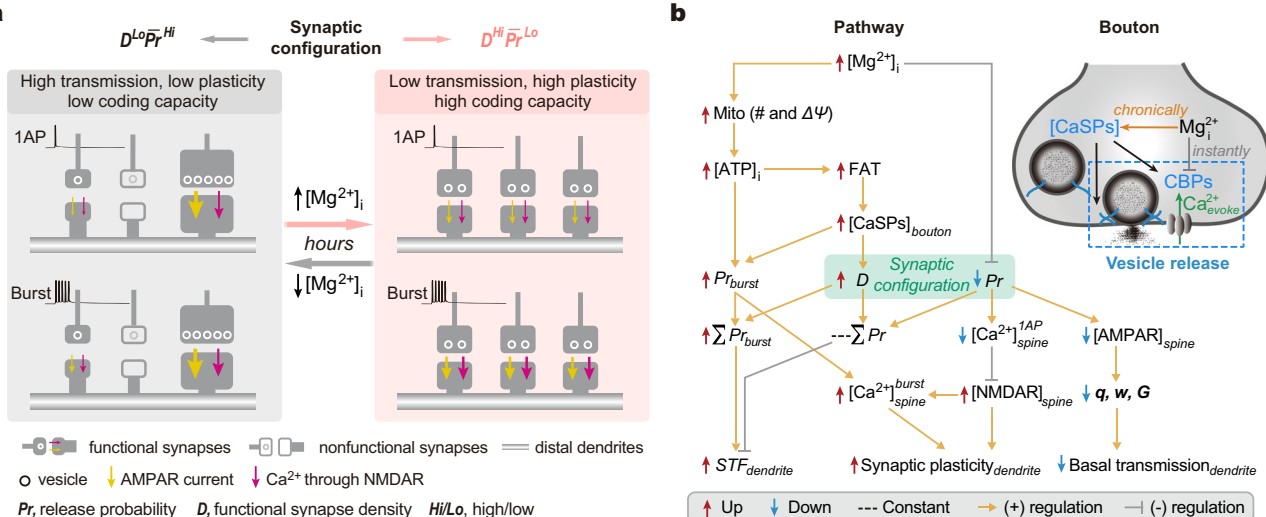

**Fig. 8 | Summary of an intracellular Mg²⁺-mediated pathway regulating synaptic configuration. a** Schematic to show that the synaptic configuration is a biomarker for dendritic branch-specific synaptic computations. Intracellular Mg²⁺ regulates the configuration and optimizes computational properties of nearby synapses along individual branches, including transmission efficiency, plasticity, and coding capacity (see Discussion). **b** A [Mg²⁺]ᵢ-mediated regulatory pathway (left, schematic pathway; top-right, regulatory scheme in a bouton). In brief, [Mg²⁺]ᵢ exerts a dual effect on **D** and **Pr**. On the one hand, [Mg²⁺]ᵢ enhances mitochondrion (Mito) functions by increasing Mito number and membrane potential (ΔΨ)[38]. This leads to an increase in intracellular free ATP levels ([ATP]ᵢ), promoting the antegrade fast axonal transport (FAT) of CaSPs[38]. As a result, presynaptic [CaSPs] increases, converting a remarkable portion of synapses from silent to functional and increasing **D**. On the other hand, it has an ongoing negative effect on **Pr** likely by competing with intracellular Ca²⁺ on presynaptic Ca²⁺-binding proteins (CBPs)

(top-right). Overall, the changes in [Mg²⁺]ᵢ lead to varying synaptic configurations across individual dendritic branches. Notably, at presynaptic sites, the strength of vesicle release during bursting transmission is dominantly determined by both burst-induced facilitation of presynaptic Ca²⁺ influx and [CaSPs], suggesting a regulatory scheme: ↑[Mg²⁺]ᵢ → ↑ [CaSPs] → ↑ **Pr**_burst. In synergy, the increase in [ATP]ᵢ accompanied by elevated [Mg²⁺]ᵢ can well support the high-energy demand of vesicle turnover during bursting transmission. Consequently, ↑[Mg²⁺]ᵢ causes increases in both ΣPr_burst and short-term facilitation (**STF**, =ΣPr_burst/ΣPr) at dendritic branches. At postsynaptic sites, elevated [Mg²⁺]ᵢ induces an increase in [GluN2B], enhancing Ca²⁺ influx via GluN2B*NMDARs in dendritic spines ([Ca²⁺]^burst_spine) during bursting transmission. Hence, both pre- and postsynaptic sites synergize burst-induced synaptic plasticity[83,111]. Overall, the synaptic configuration, crucially regulated by [Mg²⁺]ᵢ, determines branch-specific synaptic computations during information processing.

supports the notion that synaptic configuration in the hippocampus can be modified by an increase in brain Mg²⁺ levels.

In addition, we also examined learning and memory of the same living animals ahead of the above *post hoc* experiments and confirmed that the aged animals with chronic MgT treatment showed better performance in water maze tests as compared to the control group (Supplementary Fig. 10), indicating improved learning and memory.

In summary, the above experiments support the hypothesis that elevating brain Mg²⁺ levels can modify the synaptic configuration in the hippocampus of intact animals, coincident with improved learning and memory.

## Discussion

### Organization principle of synapses along dendrites

In this study, we examined the connectivity of synapses at dendritic branches and revealed that the synaptic configuration is a significant biomarker for branch-specific synaptic computations (for a schematic, Fig. 8a), which are fundamental for information processing in learning and memory. We directly measured and/or indirectly estimated multiple biophysical variables in characterizing properties of synaptic computations (for a full list and description of the variables, see Supplementary Table 1). Here, we summarized the mathematical associations between these variables to clarify the organizational principle behind.

First, influence of synaptic configuration on presynaptic properties. At presynaptic sites, the total presynaptic strength per unit area of dendrites (Σ**Pr**) during basal transmission remains constant, and functional synapse density is inversely correlated with $\overline{Pr}$, as per

formulae (1) and (2).

$$D \propto \frac{1}{\overline{Pr}} \tag{1}$$

$$\Sigma Pr = D \cdot \overline{Pr} = \text{Constant} \tag{2}$$

Second, influence of synaptic configuration on postsynaptic AMPAR-related properties. At postsynaptic sites, the quantal size, as approximated by GluA2*AMPAR or PSD95 expression levels, positively correlates with **Pr** in individual synapses. The total quantal size per unit area of dendrites (Σ**q**), as approximated by GluA2 or PSD95 density, remains constant. Notably, the $D \sim \overline{q}$ relationship mirrors the $D \sim \overline{Pr}$ combination, as per formulae (3) and (4).

$$D \propto \frac{1}{\overline{q}} \tag{3}$$

$$\Sigma q = D \cdot \overline{q} = \text{Constant} \tag{4}$$

Third, influence of synaptic configuration on postsynaptic NMDAR-related properties. At postsynaptic sites, the NMDAR quantal size (**q**_NMDAR), as approximated by the expression level of GluN2B*NMDARs, negatively correlates with **Pr** in individual synapses, while the total **q**_NMDAR per unit area of dendrites (Σ**q**_NMDAR), as represented by the total amount of receptors, positively correlates with **D**, as per formula (5). Consequently, Σ**q**_NMDAR is not constant but influenced by the synaptic configuration, as per

formulae (6) and (7).

$$D \propto \overline{q_{NMDAR}} \qquad (5)$$

$$\Sigma q_{NMDAR} = D \cdot \overline{q_{NMDAR}} \neq \text{Constant} \qquad (6)$$

$$\Sigma q_{NMDAR} \propto \frac{1}{\overline{Pr}} \text{ or } D \qquad (7)$$

Fourth, influence of synaptic configuration on AMPAR-mediated synaptic transmission. With above arrangements, the total transmission strength of excitatory synapses per unit area of dendrites during basal transmission (also named the basal gain, $G$) branch-specifically depends on the synaptic configuration. The $D^{Lo}\overline{Pr}^{Hi}$ configuration is more efficient in basal synaptic transmission than the $D^{Hi}\overline{Pr}^{Lo}$ configuration, as per formulae (8) and (9). During bursting transmission, the total strength of excitatory transmission (also named the bursting gain, $G_{burst}$) also depends on the synaptic configuration, but with larger variations, as per formula (10).

$$G = \Sigma w = \Sigma(Pr \cdot q) \neq \text{Constant} \qquad (8)$$

$$G \propto \overline{Pr} \text{ or } \frac{1}{D} \qquad (9)$$

$$G_{burst} = \Sigma w_{burst} = \Sigma(Pr_{burst} \cdot q) \neq \text{Constant} \qquad (10)$$

Fifth, influence of synaptic configuration on postsynaptic NMDAR-mediated $Ca^{2+}$ signals. During basal transmission, the total spine $Ca^{2+}$ influx per unit area of dendrites ($\Sigma w_{Ca}$) remains constant, independent of the synaptic configuration, as per formula (11). However, during bursting transmission, the total postsynaptic $Ca^{2+}$ influx upon a burst is not constant, depending on the synaptic configuration, as per formulae (12) and (13).

$$\Sigma w_{Ca} = \Sigma(Pr \cdot q_{NMDAR}) \neq \text{Constant} \qquad (11)$$

$$\Sigma w_{Ca_{burst}} = \Sigma(Pr_{burst} \cdot q_{NMDAR}) \neq \text{Constant} \qquad (12)$$

$$\Sigma w_{Ca_{burst}} \propto \frac{1}{\overline{Pr}} \text{ or } D \qquad (13)$$

In summary, the synaptic configuration influences the total strength of excitatory transmission and the postsynaptic NMDAR-mediated $Ca^{2+}$ signals. The $D^{Lo}\overline{Pr}^{Hi}$ configuration has high basal transmission efficiency, while the $D^{Hi}\overline{Pr}^{Lo}$ configuration is more effective in generating high postsynaptic NMDAR-mediated $Ca^{2+}$ influx, contributing to the induction of synaptic plasticity.

Notably, a key aspect of this organizational principle is the inverse relationship between synapse density and strength. Similar inverse relationship has been observed in different types of synapses across species[84–87]. Such a relationship may hold significant implications for cognitive functions, because underlying various cognitive processes, when a single synapse or a cluster of synapses is strengthened (or enlarged), its nearby synapses within the same dendritic branch may be weakened (or eliminated), leading to changes in synaptic connectivity[15,29,88–101]. Thus, the inverse density-strength relationship likely represents a fundamental and universal way of synaptic organization that is crucial for maintaining brain functions (see also Supplementary Notes).

## Implications of synaptic configuration to information turnover

Governed by this organizational principle, the synaptic configuration is likely a significant biomarker for branch-specific information turnover (that is, information storage versus erasure) underlying learning and memory.

Firstly, the synaptic configuration of an individual branch may determine its functional state favorable for either learning or memory. Dendritic branches of $D^{Lo}\overline{Pr}^{Hi}$ configuration are more efficient in transmitting excitatory currents in basal transmission, suggestive of a higher efficiency in information transfer; meanwhile, their less amount of postsynaptic GluN2B*NMDARs reduces the ability of coincidence detection, thus protecting the current synaptic connectivity against perturbations induced by new learning. These features could synergistically contribute to holding the stored information, which is pertinent to memory. By contrast, dendritic branches of $D^{Hi}\overline{Pr}^{Lo}$ configuration present higher burst-induced postsynaptic $Ca^{2+}$ signal ($\Sigma w_{Ca_{burst}}$) through larger number of GluN2B*NMDARs. Thereby, they are more sensitive in detecting coincidence and inducing synaptic plasticity to modify the current synaptic connectivity, favorable for encoding new information from learning. Therefore, varying configurations define the ability of nearby synapses in processing information during learning and memory in a branch-specific manner.

Secondly, the synaptic configuration per se may indicate the information coding capacity of an individual branch in its present state (such as before or after learning). It is widely accepted that synapses with high contrast in strength may crucially contribute to information representation. Single synapses, as the fundamental coding units, have varying capacities for information encoding. According to Shannon's information theory[102], synapses with normalized $Pr$ values have higher information entropy than those with polarized $Pr$ values. Our calculations of information entropy for diverse configurations within individual branches showed that the $D^{Hi}\overline{Pr}^{Lo}$ configuration has higher density of information entropy than the $D^{Lo}\overline{Pr}^{Hi}$ configuration, ranging from 0.77 to 0.10 bits $\mu m^{-2}$ (Supplementary Fig. 11). Therefore, the $D^{Hi}\overline{Pr}^{Lo}$ configuration conceptually has greater coding capacity (or less stored information) than the $D^{Lo}\overline{Pr}^{Hi}$ configuration, making it more favorable for encoding new information from learning.

Thirdly, alterations in synaptic configuration may represent information turnover. Learning induces branch-specific encoding of new information, such that it is reasonable to expect a configurational transition from $D^{Hi}\overline{Pr}^{Lo}$ to $D^{Lo}\overline{Pr}^{Hi}$ in relevant dendritic branches after learning. Indeed, theoretical and experimental studies have demonstrated at the cellular level such changes in the configuration of synaptic weight distribution following learning processes[3]. For example, after motor task learning, synaptic weights redistribute across the dendritic tree of cerebellar Purkinje cells, resulting in both an increased number of silent synapses and polarized synaptic weights in the remaining functional synapses[13]. Such a configurational transition is also evident by 3D reconstructions through electron microscopy, revealing strengthened synapses accompanied by weakened nearby synapses along the same dendrites of Purkinje cells after learning[12]. This configurational change is considered crucial for encoding task-specific input-output associations after motor learning[3,13], supporting our speculation.

Taken together, we propose that the synaptic configuration is a biomarker indicating the present state of information capacity of an individual dendritic branch, while dynamic alterations to configuration contribute to branch-specific information turnover. Future studies in behaving animals are required to obtain direct evidence at the single-synapse resolution regarding the role of synaptic configuration in information turnover during learning and memory.

## Intracellular Mg²⁺ crucially regulates synaptic configuration

In this study, intracellular Mg²⁺ is identified as a key regulator of synaptic configuration, revealing a Mg²⁺-mediated regulatory pathway under physiological conditions. Here, we summarized and illustrated this pathway (Fig. 8b) and briefly described it below.

Within individual boutons, intracellular Mg²⁺ levels ($[Mg^{2+}]_i$) enhance functions of mitochondria by increasing their density and membrane potential ($\Delta\Psi$), thereby raising intracellular free ATP concentration ($[ATP]_i$)[38]. The boosted energy supply accelerates antegrade fast axonal transport (FAT) of Ca²⁺-sensitivity-related proteins (CaSPs), leading to an increase in presynaptic [CaSPs] to facilitate the activation of silent boutons (and ultimately an increase in $D$). Concurrently, $[Mg^{2+}]_i$ exerts a pronounced inhibitory effect on $Pr$. The dual effect of $[Mg^{2+}]_i$ confers varying configurations for individual dendritic branches, thus branch-specifically modulating transmission efficiency, plasticity, and coding capacity of the synapses. Notably, factors other than intracellular Mg²⁺ could also regulate the synaptic configuration, as exemplified by the muscle-selective overexpression of fasciclin II at NMJs[87], the activation of presynaptic p35/p25/Cdk5 pathway in the hippocampal CA3 recurrent synapses[103], and the activation of postsynaptic CaMKII signaling[100]. Future studies should aim to explore and identify additional endogenous configuration-regulating factors in the brain.

In this Mg²⁺-mediated regulatory pathway, an interesting finding is the substantial negative effect of intracellular Mg²⁺ on $Pr$ in hippocampal synapses. The exact molecular mechanisms underlying this negative effect remain unclear, but several possibilities warrant further investigation. For example, physiological concentrations of $[Mg^{2+}]_i$ (that is, 1 mM) can potentially influence the conformation of the synaptotagmin-1–membrane complex via its C2 domains[104,105], which may negatively affect vesicle fusion processes. Another possibility is the peak concentration of presynaptic Ca²⁺ influx in the immediate proximity to sensor proteins might be low in a large portion of synapses; consequently, the $[Ca^{2+}]_i$ elicited by a single AP might not be sufficient to trigger vesicle fusion, making the negative effect of $[Mg^{2+}]_i$ remarkable for these synapses. Additionally, endogenous Ca²⁺ buffers have been shown to modulate $Pr$ at central synapses[106], and in our case, increased $[ATP]_i$ accompanied by high $[Mg^{2+}]_i$[38] could buffer intracellular Ca²⁺ influx upon a single AP invasion, thereby reducing the sensitivity of Ca²⁺ in triggering vesicle fusion. These possibilities, among others, require further investigation in future studies.

## Is intracellular Mg²⁺ a "memory molecule"?

We showed that intracellular Mg²⁺ crucially regulates the synaptic configuration at dendritic branches. This regulatory effect suggests that intracellular Mg²⁺ may serve as a molecular constituent of information turnover at dendritic branches, given the implications of synaptic configuration to information turnover as discussed above. First, the present $[Mg^{2+}]_i$ levels make the synapses within an individual branch optimal for either learning or memory. Low $[Mg^{2+}]_i$ promotes the $D^{Lo}\overline{Pr}^{Hi}$ configuration with the features of "high basal synaptic transmission efficiency, low plasticity and coding capacity", favoring holding information that is critical for memory, while high $[Mg^{2+}]_i$ promotes the $D^{Hi}\overline{Pr}^{Lo}$ configuration exhibiting "low basal transmission efficiency, high plasticity and coding capacity", favorable for encoding information during learning. Second, reversible fluctuation of $[Mg^{2+}]_i$ between low and high levels causes the configurational transition, which may contribute to the dynamic processes of information storing and erasing at a dendritic branch as discussed above. Therefore, intracellular Mg²⁺ may act as a modulator of branch-specific information turnover, behaving like a "memory molecule". Notably, it takes tens-of-minutes to hours for $[Mg^{2+}]_i$ to mediate an equilibrium configurational transition. Interestingly, this timescale coincides with that of short-term or daily memory turnover, raising a speculation that intracellular Mg²⁺ might be a molecular substrate of short-term or daily memory. However, direct evidence at the branch level is required to test this hypothesis in future in vivo studies.

## Translational implications

Lastly, we discussed translational implications of synaptic configuration. Aging and neurodegeneration perpetuate pronounced changes in synaptic connectivity across different brain regions in both humans and non-human primates[33–37], hallmarked by reduced synapse density and increased synaptic weight (or structural size) in the remaining synapses, consistent with our observations on hippocampal synapses in aging rodents. These long-lasting connectivity changes, resembling the $D^{Lo}\overline{Pr}^{Hi}$ configuration, impair the dynamic balance between $D^{Lo}\overline{Pr}^{Hi}$ and $D^{Hi}\overline{Pr}^{Lo}$ configurations, perhaps compromising the ability of information encoding relevant to learning. Such a scenario may be more detrimental in brain regions, like the hippocampus and prefrontal cortex, that require dynamic memory turnover to maintain persistent ability of learning and working/short-term memory.

To protect learning and memory against aging, reversing the synaptic configuration towards the $D^{Hi}\overline{Pr}^{Lo}$ mode might be a plausible approach. In support, our in vivo experiments demonstrate that the restoration of hippocampal synaptic configuration in aged animals, towards that in young animals, coincides with their improved learning and memory. Thus, targeting synaptic configuration by manipulating intracellular Mg²⁺ concentration may have a therapeutic potential for treating cognitive impairments caused by aging and neurodegenerative diseases in humans (see Supplementary Notes). Indeed, clinical studies have reported that brain Mg²⁺ supplementation by oral intake of MgT (also known as L-TAMS) improves cognitive functions of individuals with mild cognitive impairment (MCI)[107], patients with mild-to-moderate Alzheimer's disease[108,109], and attention-deficit/hyperactivity disorder (ADHD)[110].

Overall, our findings reveal that the synaptic configuration is a biomarker indicating branch-specific synaptic computations and underscore an important role of intracellular Mg²⁺, as a crucial regulator of configuration, in maintaining brain health and rejuvenating aging brains (see also Supplementary Notes). More importantly, besides intracellular Mg²⁺, these findings further suggest that any synaptic configuration-regulating factors would be potential therapeutic targets for preventing, mitigating, or perhaps reversing brain aging and neurodegeneration.

## Methods

### Animals

Neonatal *Sprague-Dawley* (*S.D.*) rats of both sexes were obtained from Vital River Laboratory (Beijing, China) or Anhui Medical School Animal Facility (Hefei, China). Adult male rats were also acquired from Vital River Laboratory and housed in temperature- and humidity-controlled rooms under a 12-h light-dark cycle. The rats were randomly assigned to either the control group or the magnesium L-threonate (MgT) supplement group. As previously described in our study[83], MgT (604 mg kg⁻¹ d⁻¹, equivalent to 50 mg kg⁻¹ d⁻¹ of elemental Mg²⁺) was administered via *ad libitum* water intake starting at 12 months of age. The average daily water intake was measured to be ∼ 30 ml. At 26 months of age, both control and MgT-treated animals were subjected to behavior tests for learning and memory using water maze (see 'Water maze'). Then, at 28 months of age, these animals were anesthetized with intraperitoneal injection of chloral hydrate (400 mg kg⁻¹) and subsequently sacrificed for histology and electron microscopy imaging. For some experiments, separate groups of young adult rats (6 months of age), aged rats (24 months of age), and aged rats (24 months of age) supplemented with MgT (starting from 16 months of age) were sacrificed for histological comparisons. The animal protocols adhered to the guidelines and were approved by the Institutional Committees on Animal Care of Tsinghua University and USTC.

## Culture of hippocampal neurons

High-density primary cultures of hippocampal CA3–CA1 pyramidal neurons obtained from neonatal *S.D.* rats (< 24 h) were utilized in this study, following the protocols described in our previous studies[38,111]. In brief, the dissociated cells were plated and cultured on 1# coverslips (8 mm × 8 mm) in a water jacket incubator (ThermoFisher) maintained at 37 °C, 5% $CO_2$, and saturated humidity for a minimum of 14 days in vitro (d.i.v.) before experimentation. The culture medium contained a physiological concentration of 0.8 or 1.2 mM $[Mg^{2+}]_o$ (for rationales, see Supplementary Notes) and a constant level of 1.2 mM $[Ca^{2+}]_o$. For experiments, cultures aged 18.69 ± 1.92 (mean ± SD) d.i.v. were utilized. In the standard working solution for imaging, both $[Ca^{2+}]_o$ and $[Mg^{2+}]_o$ were set at 1.2 mM to achieve a $[Ca^{2+}]_o/[Mg^{2+}]_o$ ratio of one. However, in certain experiments, the $[Ca^{2+}]_o/[Mg^{2+}]_o$ ratio was acutely increased to four by elevating $[Ca^{2+}]_o$ to 4.8 mM while maintaining $[Mg^{2+}]_o$ at 1.2 mM. This adjustment aimed to enhance the $Ca^{2+}$ influx in triggering vesicle release during action potential (AP) propagation by mitigating the inhibitory effect of extracellular $Mg^{2+}$ on $Ca^{2+}$ channels. All pre-treatments were administered within the home incubator prior to transfer to the working solution for imaging. Each timepoint/condition was repeated by individual coverslips derived from a minimum of three individual batches of cultures. For all culture experiments, an individual coverslip was taken as a "biological repeat".

## Pharmacological treatments

To investigate the impact of various signaling pathways on the synaptic configuration, specific drugs were introduced into the culture medium and incubated at 37 °C, 5% $CO_2$, and saturated humidity for a duration ranging from 10 min to 4 h prior to imaging. The following drugs and concentrations were utilized (Supplementary Table 2): AP5 (20 μM), ifenprodil (1 μM), glutamate (5 μM), imipramine (1 μM), TNF-α (100 pg ml⁻¹), sTNFR1 (10 pg ml⁻¹), 8-Br-cAMP (10 μM), $PKI_{14-22}$ (0.25 μM), and BDNF (10 ng ml⁻¹). To counteract the effects of TNF-α, 8-Br-cAMP, or glutamate on synaptic configuration, concurrent administration of $[Mg^{2+}]_o$ at an extra concentration of 0.4 mM (to a final 1.2 mM in concentration) or imipramine at a concentration of 1 μM was implemented alongside each drug.

## Imaging systems

Imaging experiments were conducted in a perfusion chamber (RC-27NE2; Warner) integrated into a custom-made mini-incubator, which maintained control over $CO_2$ levels (5%), temperature (33–35 °C), and humidity (saturated). The working solution was continuously perfused through the chamber using a peristaltic pump (Gilson) and pre-warmed to 33–35 °C using a line heater (SC-20 and TC344C; Sutter).

Three different imaging systems were employed in this study: a regular confocal system (FV300 or FV1000; Olympus), a spinning disk confocal system (Yokogawa), and an ultrafast imaging system (custom-made). The regular confocal system was utilized for imaging $[Mg^{2+}]_i$, FM dyes, and immunofluorescence of histology. A stack of fluorescence images was captured at a resolution of 1024 × 1024 pixels, with a pixel size of 0.05754 μm and a z-step of 0.5 μm. The spinning disk confocal system was used for imaging $Ca^{2+}$ activity at a resolution of 512 × 512 pixels, with a pixel size of 0.1151 μm and a frame rate of 36 Hz. In certain experiments, this system also captured a z-stack of FM images (0.5 μm z-step) at the same resolutions, albeit in a slower mode. The ultrafast imaging system was employed to investigate the involvement and percentage of GluN2B*NMDARs mediated $Ca^{2+}$ influx in dendritic spines. Images were acquired at a resolution of 512 × 512 pixels, with a pixel size of 0.1151 μm and a frame rate of 2000 Hz.

For most experiments, a 60 × water-immersion objective lens (UplanSApo 60 × W N.A. 1.20; Olympus) was used. The z-stack images were subsequently projected at maximum intensity for *post hoc* analysis using ImageJ (NIH). Time-lapse images were registered and automatically analyzed using CaImAn[112], an automated pipeline specifically designed for $Ca^{2+}$ activity analysis (see 'Calcium imaging and analysis').

## Measurement of intracellular magnesium concentration

To measure $[Mg^{2+}]_i$ in boutons or neurites, we utilized the fluorescence of Magnesium Green-acetoxymethyl (AM) ester (MgGrn; 5 μM) (Supplementary Table 2). Prior to imaging, the cells were incubated with MgGrn for 30 min and then transferred to medium absent of MgGrn for 15 min (to wash out extracellular dye) under the aforementioned culture conditions. A stack of MgGrn images was acquired using the imaging systems described earlier and subsequently projected at maximum intensity for *post hoc* analysis. Importantly, MgGrn imaging was performed before any electrical stimulations were delivered to avoid potential confounds introduced by evoked $Ca^{2+}$ responses.

Following MgGrn imaging, the release probability (***Pr***) of the same boutons was measured by staining with FM4-64 or FM5-95, which was elicited by a 30AP@0.5 Hz stimulation. For details on FM staining, please refer to the section 'Detection of vesicle turnover'. To compare $[Mg^{2+}]_i$ among boutons with diverse volumes (sizes), the intensity of MgGrn fluorescence was normalized by the bouton volume. Bouton volume was estimated by quantifying the total fluorescence of FM4-64 labeled by a 600AP@10 Hz stimulation, known as the 'maximum stimulation' protocol[113]. This protocol induces vesicle release for all the releasable vesicles in individual boutons[113]. The measurement of FM staining with the 600AP stimulation was performed after ***Pr*** measurement. The contours of individual boutons were determined using the 600AP⁺ puncta. For the measurement of neurite $[Mg^{2+}]_i$, the MgGrn fluorescence was normalized by the volume of individual neurite segments[38].

To calibrate the absolute value of $[Mg^{2+}]_i$ in boutons, subsequent to loading MgGrn into the cytoplasm, neurons were incubated in a Tyrode solution containing 10 μM A-23187 (a divalent ionophore; Invitrogen), 0.05–1.6 mM $Mg^{2+}$, and 0 mM $Ca^{2+}$ for 15 min to equilibrate intracellular and extracellular $Mg^{2+}$ concentrations. Following confocal imaging of MgGrn, boutons were stained with FM4-64 elicited by the 600AP stimulation to estimate bouton volume. Normalizing MgGrn fluorescence in individual boutons by bouton volume, as mentioned above, the average normalized MgGrn fluorescence of boutons in the area of interest (AOI) was used to generate the calibration curve. This curve was fitted by the Hill equation, allowing us to calculate the absolute concentration of $[Mg^{2+}]_i$.

## Detection of vesicle turnover

To visualize vesicle turnover in single boutons, we employed styryl dyes including FM1-43 (10 μM), FM4-64 (20 μM), or FM5-95 (20 μM) (Supplementary Table 2), following the previously described protocols[38,85,111]. In brief, the labeling procedure for vesicle turnover consisted of three steps: loading, washout, and unloading (refer to Supplementary Fig. 1a). Working solutions were pre-warmed and perfused into the chamber through a peristaltic pump. During loading, FM molecules were taken up by released vesicles that were elicited by different stimulation patterns such as 30AP@0.5 Hz, 6 × 5AP@100 Hz (or other bursting patterns as shown in Supplementary Fig. 1b), or 600AP@10 Hz. APs were evoked by a field stimulus (1 ms, 50 mA per pulse) generated by a stimulus isolator (A385; WPI) and delivered via parallel platinum electrodes placed at an 8 mm distance (RC-27NE2; Warner). After loading, nonspecific FM labeling was washed out using a solution containing low $[Ca^{2+}]_o$ (0.4 mM) and the quencher ADVASEP-7 (300 μM) (Supplementary Table 2). Importantly, the washout process did not affect the endocytosed FM in vesicles[38,85,111].

To exclude artifacts introduced by unhealthy boutons, an unloading procedure was performed to test whether the dyes could be exocytosed after delivering a 480AP@2 Hz stimulation. Images were captured (as described in the 'Imaging systems' section) after washout

($F_1$) and unloading ($F_2$), and the difference ($F_2-F_1$) was used to quantify the number of released vesicles in single boutons.

During the loading and unloading procedures, kynurenic acid (200 μM) or a combination of NBQX (10 μM) and AP5 (50 μM) (Supplementary Table 2) was applied to block synaptic transmission[111]. The $Pr$ or $Pr_{burst}$ was measured using quantal analysis as previously described[111]. Briefly, the fluorescence of a quantal release ($F_Q$) was measured by FM staining elicited by a single AP. The number of released vesicles per bouton ($N_v$) after 30 APs was calculated by dividing $F_Q$ by its fluorescence, and subsequently, $Pr$ was determined by $N_v/30$. To measure $D$, the number of FM$^+$ puncta per unit area of distal dendritic branches (shaft diameter in mean ± SD was 0.62 ± 0.21 μm) was quantified. Dendrites were identified by *post hoc* labeling of microtubule-associated protein 2 (MAP2, a dendritic skeleton protein, see 'Histology'), and their diameters were measured from corresponding differential interference contrast (DIC) images.

### Calcium imaging and analysis

Synaptic Ca$^{2+}$ imaging was conducted using a spinning disk confocal system (Yokogawa) equipped with a high-speed CCD camera (Andor). When imaging Ca$^{2+}$ in boutons, kynurenic acid (200 μM) or a combination of NBQX (10 μM) and AP-5 (50 μM) was added to the working solution to exclude the involvement of NMDARs and prevent recurrent activity. For Ca$^{2+}$ imaging in spines, NBQX (10 μM) was administered to exclude the participation of AMPARs. In some experiments, Ro25-6981 (1 μM) was applied to selectively block GluN2B*NMDARs.

The evoked change in fluorescence ($\Delta F/F_0$) in single boutons or spines was quantified by averaging the responses from 30 sweeps (when stimulated by 30AP@0.5 Hz) or 6 sweeps (when stimulated by 6 × 5AP@100 Hz). To determine the total amount of 1AP-evoked Ca$^{2+}$ influx in a single spine (referred to as $w_{Ca}$), the area under the $\Delta F/F_0$ curve during the evoked Ca$^{2+}$ event was calculated and averaged from 30 repeats. For quantifying $w_{Caburst}$, the area under the $\Delta F/F_0$ curve during a burst-evoked Ca$^{2+}$ event was calculated and averaged from 6 repeats, and the value was further divided by 5 to normalize it to the number of APs in the burst.

To simultaneously image spine Ca$^{2+}$ activity and presynaptic vesicle release, Ca$^{2+}$ activity in spines was measured in the presence of FM5-95 (20 μM) and NBQX (10 μM). This allowed for recording the sweeps of evoked Ca$^{2+}$ activity while visualizing AP-induced vesicle turnover (refer to 'Detection of vesicle turnover') in the same synapses. We compared the strength of vesicle release ($Pr$ or $Pr_{burst}$) with and without NMDAR blockade using AP5 (50 μM) and found no significant difference between the two conditions, indicating that asynchronous release did not significantly affect our measurements of release probability.

Time-lapse images were aligned using the Image J (NIH) plugin 'stabilizer' and then analyzed using an automated pipeline CaImAn[112], with a signal-to-noise ratio (SNR) threshold set at > 1.5. The threshold of 1.5 × SNR was determined empirically based on a consensus between automated extraction of Ca$^{2+}$ events and manual proofreading.

### Visualization of GluN2B in spines

To visualize and quantify the presence of GluN2B in individual dendritic spines, two types of fluorescent proteins were transfected into cultured neurons: CaMKIIα-mKate2 (serving as a structural marker and internal control for expression level) and CaMKIIα-GFP-GluN2B (GFP-fused GluN2B subunits)[61]. To quantify the relative amount of transfected GluN2B in each spine ([GluN2B]$_{Sp}$), the fluorescence intensity of GFP ($F_{GFP}$) was divided by the fluorescence intensity of mKate2 ($F_{mKate2}$) within each individual spine. This normalization process considers the variations in spine volume and gene expression levels among different spines. The resulting ratio, $F_{GFP}/F_{mKate2}$, represents the relative abundance of exogenous GluN2B in each individual spine.

### Plasmids and transfection

To transduce the plasmids encoding CaMKIIα-Synaptophysin-GCaMP6f, CaMKIIα-GCaMP6f, CaMKIIα-mKate2, and CaMKIIα-GFP-GluN2B into cultured neurons, the Ca$^{2+}$ phosphate protocol was employed[114] (for plasmids and reagents, see Supplementary Table 2). The process involved mixing the plasmids and CaCl$_2$ in Hank's balanced solution to achieve a final DNA concentration of 20 ng μl$^{-1}$ immediately before transfection. The DNA–Ca$^{2+}$-phosphate solution, with homogenous precipitates formed, was then applied to the cultured neurons on coverslips. The coverslips with the transfected neurons were incubated for 1 h under conditions of 5% CO$_2$, 37 °C, and saturated humidity. Subsequently, the precipitates were dissolved by transferring the coverslips to an incubator with 10% CO$_2$ and 37 °C for 15–20 min. Finally, the coverslips were returned to the original incubator with 5% CO$_2$, 37 °C, and saturated humidity to allow for gene expression. The transfected neurons, with a transfection rate ranging from 18% to 59%, were used for imaging purposes after maturation, which corresponds to > 1 week after the transfection. As a validation, it was observed that the transfection process itself did not have any effect on vesicle release probability (Supplementary Fig. 5b).

### Histology

**Cultured neurons.** A strategy of retrograde immunofluorescence was employed for multiple rounds of labeling presynaptic proteins in a synaptic network (for a schematic, see Supplementary Fig. 6a), employing the method of 'Single-synapse analysis using FM1-43 and immunofluorescence imaging array' (SAFIA) as previously described[38]. After imaging FM1-43 in the AOI with dimensions of 58.9 μm × 58.9 μm (see 'Detection of vesicle turnover'), the culture coverslips were incubated in normal Tyrode solution with [Ca$^{2+}$]$_o$ at 1.2 mM for 20 min. Then, they were fixed in a solution of 1% paraformaldehyde (E.M.S.), 0.01% glutaraldehyde (Alfa Aesar), and 4% sucrose in 1 × PBS (pH 7.4) for 1 h at room temperature (RT). The fixatives were gently washed out several times using Tris buffer (pH 7.6), which consisted of Tris Base 25 mM, Tris-HCl 25 mM, and NaCl 150 mM (filtered through a 0.22 μm film before use).

Next, the coverslips were permeabilized and blocked in a freshly prepared blocking solution (in Tris buffer) containing 1% BSA, 0.1% saponin, and 300 μM ADVASEP-7 for 1.5 h at RT. This was followed by several rounds of immunostaining. In each round, two to three primary antibodies (1:200–1:500) are incubated for 2–4 h at RT, followed by incubation with fluorophore-conjugated secondary antibodies (1:200–1:500) for 2 h at RT (for antibodies, see Supplementary Table 2). Confocal fluorescent images were then captured in the AOI, with light-field DIC images used for recognizing landmarks. After each round of staining, the antibodies were eluted by a stripping buffer (0.2 M NaOH and 0.015% SDS in deionized water) for 20 min at RT, repeated twice. The coverslips were rinsed in Tris buffer for over 1 h at RT to wash out SDS. Confocal imaging validated that < 1% fluorescent signal of presynaptic proteins remained in individual boutons after elution as compared to before (Supplementary Fig. 6b). Extreme care was taken to avoid cell detachment during this process. The coverslips were then subjected to the next round of immunostaining using 2–3 primary antibodies, with one antibody used as an internal quality control and landmark for image registration in data analyses. Typically, a maximum of 4–5 rounds of immunostaining could be performed on a culture coverslip (Supplementary Fig. 6c).

Specifically, for immunostaining the extracellular epitope of GluA2*AMPARs, the neurons were incubated for 1 h at RT in the blocking solution with anti-GluA2 antibody (dilution, 3 μg ml$^{-1}$) but with no addition of any detergents to keep the membrane intact. This procedure was performed after measuring $Pr$ in the same synapses using FM1-43 (see 'Detection of vesicle turnover').

For analyzing images using the SAFIA approach in cultured neurons, the acquired images were initially coarsely aligned using 'Rigid

Body Registration'[115] and then further refined using 'UnwarpJ'[116] in Image J (NIH) to correct any physical distortions present. To normalize the fluorescence intensity of VGLUT1 and each individual CaSP (referred to as [CaSP$_i$]) within individual boutons, they were initially divided by their respective median values in the bouton population. The size of each bouton was estimated based on the normalized quantity of VGLUT1 fluorescence (referred to as [VGLUT1]), as there is a well-established correlation between [VGLUT1] and the total number of vesicles in individual boutons[38]. To quantify the collective impact of all CaSPs on vesicle release within a bouton, the geometric mean of the quantity of all CaSPs (referred to as [CaSPs]) was calculated using the following formula

$$[CaSPs] = \frac{\prod_i [CaSP_i]}{[VGLUT1]} \tag{14}$$

where CaSP$_i$ represents the $i$th CaSP, $\prod$ is the product operator.

**Tissue slices.** For immunostaining on ultrathin tissue slices of the hippocampus, procedures were taken as previously described[38]. In detail, the rats were anesthetized and transcardially perfused with $1 \times$ PBS and fixative solution (1% paraformaldehyde and 0.01% glutaraldehyde in $1 \times$ PBS), subsequently. The CA1 stratum radiatum (s.r.) region of hippocampus was then sectioned at 100 μm thickness using a vibratome (VT1000S; Leica). These sections were further trimmed into < 1 mm$^2$ square blocks for either immunostaining or electron microscopy imaging (see 'Electron microscopy-based 3D reconstruction'). The tissue blocks for immunostaining were treated with 1% tannic acid (Sigma) for 1 h, followed by treatment with 1% uranyl acetate in maleate acid buffer (pH 6.0) for 1 h. Subsequently, the blocks were dehydrated sequentially in 50%, 70%, 80%, 95%, 100%, and 100% ethanol containing 1% paraphenylenediamine (PPD, Sigma) for 15 min each. The tissue blocks were then infiltrated with a mixture of ethanol and LR White resin in various ratios, followed by pure resin, with each infiltration step lasting for 1 h. Finally, the blocks were placed in capsules filled with LR White resin and polymerized in a 55 °C oven for 48 h (for related reagents, see Supplementary Table 2). Ultrathin slices of 70 nm thickness were cut and mounted on coverslips for immunostaining, using primary antibodies for synaptic proteins at 1:300, incubated at 4 °C overnight and secondary antibodies at 1:500, incubated at RT for 2 h (for antibodies, see Supplementary Table 2).

**Electron microscopy-based 3D reconstruction**
Tissue blocks were dissected from the CA1 s.r. region of aged rats at 28 months of age, as described in the 'Histology' section. The dissected blocks were then post-fixed in a solution containing 1% osmium tetroxide and 1.5% potassium ferricyanide in Maleate buffer (pH 6.0) for 1 h. Subsequently, the blocks were stained with 3% uranyl acetate for 1 h. After staining, the blocks underwent a series of dehydration steps using sequential ethanol concentrations (50%, 70%, 80%, 95%, and 100%) for 15 min each. Following dehydration, the blocks were treated with 100% propylene oxide twice for 10 min each. These procedures were performed while keeping the blocks on ice. The next step involved the infiltration of the blocks with a mixture of 50% propylene oxide and 50% epoxy resin for 60 min at RT, followed by pure epoxy resin for 24 h at 4 °C. The blocks were then transferred to embedment molds filled with pure epoxy resin and placed in a 60 °C oven for 24 h to allow polymerization (for related reagents, see Supplementary Table 2).

Serial ultrathin slices with a thickness of 70 nm were cut using a diamond knife on a microtome (Leica) and mounted on silicon chips to enhance electro-conductivity. The slices were subsequently stained with 3% uranyl acetate and 0.4% (w/v) lead citrate in sequential steps. Images of the slices were captured using a scanning electron microscopy (Supra55, Zeiss) with specific imaging parameters (10 keV, $8000 \times 8000$ pixels, 2 nm pixel$^{-1}$, 15 μs pixel$^{-1}$). To correct any distortions, the serial electron microscopy images were registered using UnwarpJ in ImageJ[116]. The registered images were then utilized for the reconstruction of synapses. Individual synapses were randomly selected and reconstructed in 3D using the TrakEM2 plugin in Image J[117].

**Water maze**
A circular metal tank of 150 cm in diameter and 50 cm in depth was filled with 22 °C water, 30 cm in water depth. In the training session, an acrylic platform (15 cm in diameter) was set in the center of a quadrant, immersed in the pool, with the upper surface 2 cm below the water. Four spatial cues distinct in shape and color were placed evenly around the tank. The training session was split into two 4 days, 5 trials per day with an 1 h inter-trial interval. The positions into the water were randomly selected. In each training trial, an individual rat was allowed for a maximum 90 s staying in the water. During this period, if the rat found the platform, it was allowed for standing on the platform for 30 s and then dried by towels and returned to the home cage prewarmed by a 37 °C heater; otherwise, it was guided to the platform, standing for 30 s. The latency (90 s at maximum) to reach the platform was recorded to generate the learning curve. After the training session, each rat was tested in the water maze for 90 s in the absence of platform 3 days after the last training trial, and the percentage of time spent in the quadrants was analyzed. The data were recorded and analyzed automatedly by a behavioral system (EthoVision XT, Noldus, Netherlands).

**Information entropy**
Information entropy of individual presynaptic boutons was calculated using the Shannon's formula[102], as follows.

$$H(Pr) = -[Pr \cdot \log_2(Pr) + (1 - Pr) \cdot \log_2(1 - Pr)] \tag{15}$$

For each dendritic branch, the density of information entropy of presynaptic boutons was calculated by the total entropy of the synapses per unit area of dendrites.

**Statistics**
The normality of the data was tested using the Shapiro-Wilk test, and based on the results, either nonparametric or parametric statistical tests were chosen. For comparing two groups, paired or unpaired $t$-tests were used, depending on the nature of the data and the study design. When multiple comparisons were made, one-way or two-way ANOVA was performed, followed by *post hoc* Bonferroni's test to determine specific group differences. The Kolmogorov–Smirnov test was employed to compare distributions between groups. All statistical tests were two-sided, meaning that both the possibility of a positive and a negative effect was considered. The significance level was set at $P < 0.05$. GraphPad Prism software was used for conducting all the statistical analyses.

**Reporting summary**
Further information on research design is available in the Nature Portfolio Reporting Summary linked to this article.

## Data availability
All data are available in the main text or the supplementary materials. Source data are provided with this paper.

## Code availability
The codes used in this study are open-source and accessible online. github.com/flatironinstitute/CaImAn.

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

## Acknowledgements

We express our gratitude to Drs. T. Südhof (Stanford) and J. Feldman (UCLA) for their valuable input and critical review of the manuscript before submission. We would also like to acknowledge Drs. H. Han and X. Chen (CASIA) for their support with electron microscopy, Drs. L. Qi, C. Xu (USTC) for providing plasmids, Dr. F. Xu (SIAT) for support with the ultrafast imaging system, and Drs. P.-M. Lau (SIAT) and R. Zhao (Shenzhen Bay Laboratory) for their scientific inputs. This work was supported by the THU-CASIA Technical Cooperation Project (THU 20142001129) and Univ. SIAT Tenure-Track Startup (to H.Z.), National Natural Science Foundation of China (31630030) and Chinese Academy of Sciences (XDB32030200) (to G.-Q.B.), and THU Initiative Scientific Research Program (THU 20131080156) (to G.L.).

## Author contributions

H.Z. and G.L. conceived the project and wrote the manuscript. H.Z., G.-Q.B. and G.L. designed the experiments and interpreted the data. H.Z. performed the experiments and data analyses. All authors edited and approved the final manuscript.

## Competing interests

G.L. is a cofounder of NeuroCentria, a company dedicated to developing drugs for brain diseases. H.Z. and G.-Q.B. declare no competing interests.
