## [Peer Review File · Nature Communications]

Intracellular magnesium optimizes transmission efficiency and plasticity of hippocampal synapses by reconfiguring their connectivityREVIEWER COMMENTS

Reviewer #1 (Remarks to the Author):

The manuscript, "Intracellular magnesium optimizes transmission efficiency and plasticity of hippocampal synapses by reconfiguring their connectivity," by Zhou et al is centered on deciphering how Mg²⁺ levels impact synaptic transmission properties within dendritic branches and their potential role in information processing. To interrogate this question, authors utilize an in vitro primary hippocampal cell culture model of CA1 and CA3 neurons at DIV 14-28. To measure synaptic properties along defined regions on the dendritic branch they use an all-optical approach to measure synaptic strength, Ca²⁺ influx in response to varying [Mg²⁺] and pharmacological drugs. The authors then demonstrate that increasing [Mg²⁺] in 28 month old rats through daily oral uptake of a Mg²⁺ compound leads to changes in PSD95 levels and synapse volumes in the CA1 region. Finally, using their measured parameters, the authors formalize a series of equations to describe how synaptic connectivity on the dendritic branch impacts information transfer in the brain. Based on these findings, the authors propose that intracellular Mg²⁺ levels are key regulators of synaptic connectivity and is a molecular substrate of memory. Furthermore, they propose that initial synaptic strength and changes in synaptic strength along the dendritic branch are key determinants of learning and memory.

Overall, the studies are largely confirmatory of previous findings and concepts in the field, many which have been published by the Liu group. The experiments appear to replicate earlier findings by the Liu group which demonstrated: 1) that increased [Mg²⁺] in cultured neurons changes synaptic strength (Slutsky et al 2004 PMID: 15572114). 2) increase in higher Mg²⁺ levels increase synaptic protein levels known to control synaptic strength (Zhou and Liu 2015 (PMID 26184109) and 3) A magnesium compound, MgT, increased animal memory and density of synapses (Slutsky et al 2010 PMID: 20152124). In addition, the concept of synapses with low synaptic strength being sites of plasticity for learning and memory has been established by this group and many others. However, new to this study compared to old studies by the Liu group is the measure of synaptic strength along the dendritic branch and a mathematical formalization of how changes in synaptic strength impact information transfer at the dendritic branch. At a technical level, the experiments are solid. However, this reviewer has comments for the authors to consider.

- 1). The authors rely on cell-culture model of DIV14-28. It is unclear how development of the culture model impacts their findings. There will be clear differences between DIV 14 and 28 neurons and this is never accounted for. In addition, this is an artificial culture model that unlikely replicates the native circuit in vivo. Thus, it is extremely difficult to extrapolate how a culture of glutamatergic neurons mimics what is happening in the in vivo circuit and whether proposed mechanisms described here are in fact utilized in the native circuit.

2) Many of the parameters are defined by responses to 0.5 Hz and 30 AP. However, for this reviewer it is difficult to understand why these parameters to measure D and Pr are chosen. These parameters differ depending on the number of APs and frequency. Typically, basal synaptic transmission is defined as 0.1 Hz. Furthermore, how are potential branch point failures accounted for.

3) The idea of Mg²⁺ concentrations having negligible impact at NMJ and the Calyx and being due to difference in the synapses in this study are not accurately represented. The experiments in this manuscript are incubating cultures with different [Mg²⁺] for 4hrs at minimum. While experiments in the NMJ and Calyx were completely different and not incubated for 4hrs. Therefore, the authors findings of long-term incubation are describing a change in homeostatic plasticity, while the Calyx and NMJ work is not looking at long term chronic changes.

4) Given that the authors have already defined many of the molecules impacted by changes in [Mg²⁺] it would have been ideal to perturb one of these molecules or a molecule that is affected by aging and analyze the impact on connectivity by increasing Mg²⁺ levels.

Reviewer #2 (Remarks to the Author):

In this manuscript, Zhuo et al. elucidated the possible mechanism of how intracellular magnesium modulates synaptic transmission efficiency and the implication of potential beneficial effects of supplement magnesium to improve learning and memory. The authors suggested that “high transmission efficiency” synapses have low plasticity and coding capacity, which favors memory, but “low transmission efficiency” synapse has high plasticity and coding capacity, which favors learning. Using FM dye imaging, the authors showed that different dendritic branches have a relatively stable per unit area presynaptic weight distribution, termed synaptic configuration by the authors, which remains constant. They show that adjusting Magnesium concentration reduced the release probability of synaptic vesicle release but also shifted the “silence synapses” into functional synapses. They further combined FM imaging with post hoc immunostaining. They showed that postsynaptic proteins such as GluA1 and PSD95 expression levels positively correlate with synaptic weight, and their expression level remains constant per unit area of dendrites under different release conditions. Combined with FM imaging and dendritic spine calcium imaging, they show that per unit dendrite Calcium influx (mediated by NMDA R) in individual dendritic branches remains constant under basal conditions but low release probability dendrites favor calcium influx during burst transmission. Using magnesium imaging, they showed that intracellular magnesium concentration is negatively correlated with synaptic release probability. Finally, using EM studies, they show that magnesium supplements in aged rodents increase functional synapses. Overall, this is a fascinating study, following their previous studies (e.g., Slutsky I et al. 2010 Neuron), suggesting magnesium might be a therapeutic target for enhancing learning and memory. The paper's

conclusions are supported by the data presented and will be interesting to the community. Nevertheless, I have the following comments.

1. The data presented in the paper is very intense, and the writing of the article is tough to understand, except for those experts with strong computational backgrounds. Despite years of experience in electrophysiology and synaptic physiology, I had to read the paper many times to understand it. I must admit that I still don't understand certain calculations. I recommend the authors consider revising the paper to make it more readable for a wider audience. It might help increase the impact of the study.

2. The second question I have is that the data presented suggested that there is a "homeostasis" of presynaptic release and postsynaptic protein distributions. Although the authors showed that some signaling pathways are involved, it is still a mystery to me how the proteins are redistributed under different conditions, e.g., high intracellular magnesium.

3. The central nervous system is exceedingly diverse, with hundreds of types of neurons and trillions of synapses, some of which need high fidelity, for example, the auditory and visual systems. How will the general level changes of magnesium in the brain, i.e., by supplement magnesium, affect other types of synapses than the hippocampal ones?

4. Release probability is the central question of this paper; I wasn't entirely clear, maybe I missed reading it, how the release probability at a single synaptic bouton was calculated using FM imaging. It also appears to me that only evoked releases are considered. Are there any components of spontaneous releases at these synapses? Does the spontaneous release unload the loaded FM dyes? Moreover, how to combat and correct photobleaching during the imaging process?

5. I wasn't entirely convinced by the term "silence synapse" used in the study. I think the authors defined "silence synapse" for those loaded puncta that show a release probability of <0.04 , considered a silence synapse. This definition is different from the classical "silence synapses" proposed previously. (e.g., Liao et al. 1995, Nature; Isaac et al. Neuron 1995). I understand that the authors also showed a correlation of postsynaptic protein expression level in correlation with the release probability. I wonder if some other endocytic process than the retrievals of synaptic vesicles from the plasma membrane during the loading process would also be labeled, and those probably would have a very low possibility of being re-released.

6. For establishing presynaptic release and postsynaptic Calcium signals, they used GCaMP6f; I was wondering if they could also observe presynaptic Ca^{2+} dynamic changes in these experiments (since the videos have been acquired).

7. The authors used many immunostainings for postsynaptic proteins. The source of antibodies should be provided. In addition, Figure 5b showed eight post hoc immune staining for various postsynaptic proteins after FM unloading after stimulation. I wonder how this was done. I wasn't clear about this after reading the method section.

8. A lot of the data shown are cropped single-button images. It would be helpful if the author could provide some level of unprocessed images as supplemental data.

Reviewer #3 (Remarks to the Author):

Zhoe et al. investigated the role of intracellular magnesium in optimizing synaptic transmission efficiency and plasticity in the hippocampus, which is important for learning and memory processes. By examining the connectivity of individual synapses within dendritic arbors, they demonstrated an organizational principle that directs nearby synapses to generate specific computational features using two modes favoring “memory” or “learning”, and that increasing intracellular magnesium levels enhances synaptic transmission and plasticity, leading to improved learning and memory. The study was based on classical physiological approaches, and similar experiments were repeated. Also, the gap between the micro-level behavior of synapses and the macro-level behavior of learning is not yet bridged. As a result, the overall impression was one of blurred focus and a sense that the discussion was not theoretically robust. The (intentional?) addition of the aging perspective (which seems an ulterior motive aimed at making a good impression) has also contributed to the impression that the discussion is scattered. This is due partly to the fact that the landing point of this study is unclear, despite the complexity of the experiments. Apart from these general impressions, the physiological validity of the detailed experimental conditions remained highly questionable, as follows.

(i) The distribution per branch fluctuates in Fig. 1, but under the high Mg^{2+} condition, the plots move to positions that are not present under the low Mg^{2+} condition, can this be considered physiological? Under actual in vivo conditions, are the Mg^{2+} concentrations used in Figs. 1-6 feasible?

(ii) What is the concentration of the high Mg^{2+} condition referred to (the low Mg^{2+} condition is described in the text)? It is not clear under what circumstances such high concentrations could occur.

(iii) Is it an appropriate experiment to cause a uniform increase in concentration in all spines in Fig. 3 and elsewhere?

(iv) The authors consistently refers to intracellular Mg^{2+} , but to what extent can the extracellular application of Mg^{2+} change intracellular Mg^{2+} ? Quantification is needed. (The study focuses on the intracellular Mg^{2+} concentration, but it does not examine the absorption of Mg^{2+} into cells or the measurement of intracellular Mg^{2+} concentrations.)

(v) Is there any knowledge that intracellular Mg^{2+} concentration changes with ageing, or are recovery experiments alone sufficient?

(vi) To confirm whether Mg^{2+} is actually involved in aging, it is necessary to examine what happens when Mg^{2+} is restricted in young mice.

We express our sincere gratitude to the three reviewers for their meticulous evaluation and constructive feedback on our manuscript.

The reviewers raised concerns regarding the model systems of cultured hippocampal neurons and the consistency of conclusions between *in vitro* and *in vivo* systems from a translational perspective. Meanwhile, reviewer 3 felt there appears a "gap" between micro-level and macro-level behaviors in this manuscript. As this study is part of a series investigating the positive impact of Mg^{2+} on brain health and aging, we would like to provide a concise overview of our serial studies, which explore the role of Mg^{2+} ions in synaptic, neuronal, circuitry, and cognitive functions over the years. Our investigations span from *in vitro* experiments to *in vivo* studies involving animals and humans, from the micro-level of proteins and single synapses to the macro-level behaviors and cognitive functions. We believe that this overview would in general address the related concerns by reviewers.

Initially, we observed that elevating extracellular Mg^{2+} concentration enhanced long-term potentiation (LTP) of synapses in cultured hippocampal neurons, leading to increased expression of GluN2B-containing NMDARs (PMID: 15572114). Building upon this discovery, we hypothesized that raising brain Mg^{2+} levels could improve synaptic plasticity in the hippocampus, thereby enhancing cognitive functions, especially learning and memory, in intact animals. To achieve this, we developed Magnesium L-Threonate (MgT), a compound that effectively increased Mg^{2+} bioavailability in the cerebrospinal fluid (CSF) when orally consumed (PMID: 20152124). Elevating Mg^{2+} in the rodent brain's CSF demonstrated enhanced synaptic plasticity and cognitive functions in both young and aging animals (PMID: 20152124), validating our *in vitro* hypotheses. Concurrently, we observed beneficial effects in treating cognitive declines in Alzheimer's disease model mice (PMID: 25213836) and depression model mice (PMID: 22016520).

Encouraged by these animal studies, we expanded our research to translational studies. The first double-blind placebo-controlled clinical study demonstrated that MgT supplementation improves cognitive functions in mild cognitive impairment (MCI) patients (PMID: 26519439). Currently, three ongoing FDA-approved phase 2b/3 clinical trials are investigating MgT's role in treating cognitive disorders in humans, including Alzheimer's disease (DOI: 10.1016/j.jalz.2017.06.1865), Attention Deficit Hyperactivity Disorder (ADHD) (PMID: 32162987), and depression/anxiety.

Despite these promising clinical studies, the mechanism underlying the powerful impact of Mg^{2+} on human brain functions remains elusive. Initially, we believed that the primary effect of extracellular Mg^{2+} targets NMDARs to influence plasticity based on electrophysiological and molecular evidence (PMID: 15572114), demonstrating its extracellular modulatory effect. However, we later discovered that the beneficial effects extend beyond modulating synaptic plasticity. Subsequently, our findings revealed that intracellular Mg^{2+} plays an even more crucial

role in regulating the density of functional presynaptic boutons (PMID: 26184109), offering a new perspective on Mg^{2+} 's role in promoting brain health.

Intriguingly, in the compound MgT, threonate (T) itself synergizes with Mg^{2+} , elevating intracellular Mg^{2+} levels and increasing the density of presynaptic boutons in cultured hippocampal neurons (PMID: 27178134). This insight contributes to understanding the pharmacological effects of MgT in elevating brain Mg^{2+} levels and enhancing animal cognitive functions. Despite focusing on single synapses in these mechanistic studies, it remains unclear how intracellular Mg^{2+} governs multiple synapses along individual dendritic branches, imparting different transmission efficiency, plasticity, and coding capacity. Given the fundamental role of dendritic branches in processing information, addressing this question could illuminate how nearby synapses are regulated to achieve specific computational features at individual dendritic branches and identify endogenous factors controlling such synaptic organization.

In our current study, we demonstrate that altering intracellular Mg^{2+} levels reconfigures connectivity at individual branches, modulating their computational features to favor learning and memory. These configurational changes, coinciding with improved learning and memory in aged animals, advance our understanding of Mg^{2+} 's beneficial role in promoting brain health. Importantly, our findings suggest that beyond Mg^{2+} , any endogenous factors that can induce synaptic configuration changes may play crucial roles in maintaining brain health and treating aging-related cognitive decline. This concept may hold significance for exploring new compounds in future studies aimed at combating brain aging and neurodegeneration.

The point-by-point responses to reviewers are below.

Reviewer #1 (Remarks to the Author):

The manuscript, “Intracellular magnesium optimizes transmission efficiency and plasticity of hippocampal synapses by reconfiguring their connectivity,” by Zhou et al is centered on deciphering how Mg^{2+} levels impact synaptic transmission properties within dendritic branches and their potential role in information processing. To interrogate this question, authors utilize an *in vitro* primary hippocampal cell culture model of CA1 and CA3 neurons at DIV 14-28. To measure synaptic properties along defined regions on the dendritic branch they use an all-optical approach to measure synaptic strength, Ca^{2+} influx in response to varying $[Mg^{2+}]$ and pharmacological drugs. The authors then demonstrate that increasing $[Mg^{2+}]$ in 28 month old rats through daily oral uptake of a Mg^{2+} compound leads to changes in PSD95 levels and synapse volumes in the CA1 region. Finally, using their measured parameters, the authors formalize a series of equations to describe how synaptic connectivity on the dendritic branch impacts information transfer in the brain. Based on these findings, the authors propose that intracellular Mg^{2+} levels are key regulators of synaptic connectivity and is a molecular substrate of memory. Furthermore, they propose that initial synaptic strength and changes in synaptic strength along the dendritic branch are key determinants of learning and memory.

Overall, the studies are largely confirmatory of previous findings and concepts in the field, many which have been published by the Liu group. The experiments appear to replicate earlier findings by the Liu group which demonstrated: 1) that increased $[Mg^{2+}]$ in cultured neurons changes synaptic strength (Slutsky et al 2004 PMID: 15572114). 2) increase in higher Mg^{2+} levels increase synaptic protein levels known to control synaptic strength (Zhou and Liu 2015 (PMID 26184109) and 3) A magnesium compound, MgT, increased animal memory and density of synapses (Slutsky et al 2010 PMID: 20152124). In addition, the concept of synapses with low synaptic strength being sites of plasticity for learning and memory has been established by this group and many others. However, new to this study compared to old studies by the Liu group is the measure of synaptic strength along the dendritic branch and a mathematical formalization of how changes in synaptic strength impact information transfer at the dendritic branch. At a technical level, the experiments are solid. However, this reviewer has comments for the authors to consider.

1). The authors rely on cell-culture model of DIV14-28. It is unclear how development of the culture model impacts their findings. There will be clear differences between DIV 14 and 28 neurons and this is never accounted for.

Thank you for pointing out this issue. In our initial submission, we described in methods as 14–28 days *in vitro* (DIV) cultures, which was not an accurate methodological description. To make it more accurate, we revisited our data and calculated the mean age and standard deviation (SD) of the cultures we used. The cultures were 18.69 ± 1.92 DIV ($n = 487$ biological repeats), and the distribution of their ages is as follows. Most of the data were collected from 17–20 DIV cultures. We revised the inaccurate description in Methods, and now it reads “ 18.69 ± 1.92 days *in vitro*”.

1) (...) In addition, this is an artificial culture model that unlikely replicates the native circuit in vivo. Thus, it is extremely difficult to extrapolate how a culture of glutamatergic neurons mimics what is happening in the in vivo circuit and whether proposed mechanisms described here are in fact utilized in the native circuit.

We totally agree that the discrepancy between *in vitro* and *in vivo* systems would affect the significance of scientific findings for translational research. Primary culture of hippocampal neurons is a conventional *in vitro* model system for studying synaptic transmission and plasticity, and the value of this system has been well proved over decades. In our studies over decades, we employed this simplified system to generate primary hypotheses, and then tested them in both animals and humans.

We have briefly reviewed the journey of our serial studies on the beneficial role of Mg^{2+} for synaptic plasticity and brain health at the beginning of this “response to reviewers” document. These studies suggest that at least in our specific case, such a simplified model system works well in bridging *in vitro* and *in vivo* findings. Notably, in these studies, we have shown some evidence at the *in vivo* circuit level, such as the CA3–CA1 Schaeffer-collateral glutamatergic circuits (PMID: 20152124), to confirm the enhancement of hippocampal synaptic transmission and plasticity by elevating Mg^{2+} levels. Moreover, at the behavior level, the results of clinical trials are also consistent with the findings from the culture system (for details, see the reply to **Reviewer-3-Point-v**). Nevertheless, we agree that the implications of this study to the behavior of native circuits still need to be further addressed in the *in vivo* systems.

We knew and really appreciated that you have already read most of the papers we have published, still we hope this brief review would further address your concerns on the translational significance of the current study.

2) Many of the parameters are defined by responses to 0.5 Hz and 30 AP. However, for this reviewer it is difficult to understand why these parameters to measure D and Pr are chosen. These parameters different depending on the number of APs and frequency. Typically, basal synaptic transmission is defined as 0.1 Hz. Furthermore, how are potential branch point failures accounted for.

Thank you for bringing up this concern. As you pointed out, low-frequency electrical stimulations are typically used for examining basal synaptic transmission. Conventionally, 0.1 Hz or even lower frequency such as 0.033 Hz is employed in single-cell or axonal fiber stimulations for electrophysiological recordings in long-time experiments (for example, tens of minutes to several hours) on acute brain slices. Such low-frequency stimulations are necessary for avoiding synaptic depression introduced by repeated stimulations. In contrast, our experimental system employed field electrical stimulations (FS) for the measurement of vesicle release probability (Pr). FS directly and evenly delivers currents to all axons within the parallel electrical field to trigger vesicle release in individual presynaptic boutons.

The choice of our stimulation protocol, "30 APs at 0.5 Hz", is grounded in a careful balance of three major factors: number of action potentials (APs), frequency of APs, and the time of FM presence in extracellular solution. First, given our imaging system's capability to detect the fluorescent content of a single vesicle, we selected a 30-AP stimulation for Pr measurement, ensuring the detection of at least 1 released vesicle per 30 APs in a single bouton (*i.e.*, minimum detectable Pr is 1/30). Second, the frequency was set at 0.5 Hz, validated in our previous studies to avoid inducing changes in synaptic plasticity during 30 APs stimulation. Moreover, NMDARs and AMPARs were concurrently blocked with AP5 and NBQX to further minimize potential plasticity changes. Third, to limit nonspecific FM dye staining, we optimized the FM presence duration to 1 min (equivalent to the 30 APs at 0.5 Hz duration), which did not introduce significant nonspecific staining of FM dyes. This protocol is a result of balancing these three factors (PMID: 15572114).

In addition, we have also validated that an FS with 50-mA current can sufficiently depolarize all axonal arbors within the parallel electrical field (PMID: 15572114), minimizing the likelihood of "branch point failures" of axons contributing to the measurement of presynaptic vesicle release.

3) The idea of Mg^{2+} concentrations having negligible impact at NMJ and the Calyx and being due to difference in the synapses in this study are not accurately represented. The experiments in this manuscript are incubating cultures with different $[Mg^{2+}]$ for 4hrs at minimum. While experiments in the NMJ and Calyx were completely different and not incubated for 4hrs. Therefore, the authors findings of long-term incubation are describing a change in homeostatic plasticity, while the Calyx and NMJ work is not looking at long term chronic changes.

Upon further review of the literature, we realized that the studies akin to our experimental design, specifically focusing on the impact of chronically elevated intracellular Mg^{2+} levels on transmitter release of individual release sites at Calyces and NMJs, are lacking. Therefore, the notion "intracellular Mg^{2+} has negligible effect on vesicle release in Calyces and NMJs" may not be accurate, as you rightly pointed out.

To address this, we have removed the inaccurate description in both the Results and Discussion sections.

4) Given that the authors have already defined many of the molecules impacted by changes in $[Mg^{2+}]$ it would have been ideal to perturb one of these molecules or a molecule that is affected by aging and analyze the impact on connectivity by increasing Mg^{2+} levels.

We presume you are referring to downstream molecules affected by intracellular Mg^{2+} levels, specifically Ca^{2+} -sensitivity-related proteins (CaSPs), including Rab3a, RIM1, Munc13-1, Synaptotagmin (SYT), ELKS, and Syntaxin-1, whose protein levels in presynaptic boutons fluctuate with changes in intracellular Mg^{2+} .

We want to clarify that our study demonstrates a significant reduction in the protein levels of many CaSPs during aging (new **Fig. S9**). These proteins collectively influence the sensitivity of Ca^{2+} -triggered transmitter release during vesicle turnover and, consequently, synaptic connectivity. Knockdown (KD) or knockout (KO) of one or several of these proteins significantly alters or impairs synaptic connectivity in the brain, and in extreme cases, can be lethal for animals after birth (for a review, PMID: 15630409). Considering the massive impact of genetic manipulations on synaptic connectivity, we speculate that the elevation of Mg^{2+} levels may not be able to counteract the negative effect of these genetic manipulations on synaptic connectivity. Therefore, we did not conduct such experiments given these concerns.

Alternatively, we performed the following experiment to examine the role of elevating Mg^{2+} levels for restoring CaSP protein levels during aging in animals. Immunohistology on 70-nm ultrathin brain slices from the CA1 stratum radiatum (s.r.) region of the hippocampus (HP/CA1/s.r.) was conducted to measure changes in these molecules during aging. A significant decline in CaSP protein levels, ranging from $-19.01 \pm 15.84\%$ to $-25.79 \pm 16.07\%$ for individual ones, was observed when comparing aged animals (24 months of age) with young adults (6 months of age) (new **Fig. S9**), indicating a natural decline in CaSP protein levels during aging. Therefore, aging-associated decline in synaptic functionality is likely attributed to reduced levels of these proteins.

Importantly, when we elevated brain Mg^{2+} levels through oral MgT supplementation in aged animals, the reduction of CaSP protein levels was significantly mitigated in the HP/CA1/s.r. (new **Fig. S9**). These observations suggest that increasing brain Mg^{2+} levels can prevent aging-related changes in synaptic connectivity. Indeed, our electron microscopic data directly demonstrate that elevating brain Mg^{2+} levels rejuvenates synaptic connectivity in aged rats to that of young adult rats (**Fig. 7**). Intriguingly, both animal and human studies have documented a decline in brain Mg^{2+} levels during aging (see the response to **Reviewer-3-Point-vi**). Collectively, these findings emphasize the association between brain Mg^{2+} levels, changes in synaptic connectivity, and protein levels of presynaptic CaSPs during aging. This association may inspire new strategies for anti-brain aging and anti-neurodegeneration. In the revised manuscript, these results have been incorporated into new **Fig. S9**, as shown below.

In response to your insightful suggestion, although we didn't conduct experiments to perturb one or several molecules affected by aging, we believe that our current data still provide insight into the involvement of brain Mg^{2+} levels in alterations of synaptic connectivity during aging. We agree that such an intriguing experiment you proposed should be considered in the future.

New Fig. S9 | Brain Mg²⁺ supplementation mitigates the decline of CaSPs in the hippocampus during aging.

a, Immunostaining of CaSPs on 70-nm ultrathin slices from the CA1 stratum radiatum (s.r.) region of the hippocampus. From left to right, representative confocal images from young adult rats (6 months of age), aged rats (24 months of age), and aged rats (24 months of age) supplemented with MgT for 8 months (starting from 16 months of age). **b**, Quantification of the protein levels of individual CaSPs (fluorescent intensity of individual rats normalized to the mean of young adults) ($n = 8, 10, 11$ rats, respectively).

Reviewer #2 (Remarks to the Author):

In this manuscript, Zhuo et al. elucidated the possible mechanism of how intracellular magnesium modulates synaptic transmission efficiency and the implication of potential beneficial effects of supplement magnesium to improve learning and memory. The authors suggested that “high transmission efficiency” synapses have low plasticity and coding capacity, which favors memory, but “low transmission efficiency” synapse has high plasticity and coding capacity, which favors learning. Using FM dye imaging, the authors showed that different dendritic branches have a relatively stable per unit area presynaptic weight distribution, termed synaptic configuration by the authors, which remains constant. They show that adjusting Magnesium concentration reduced the release probability of synaptic vesicle release but also shifted the “silence synapses” into functional synapses. They further combined FM imaging with post hoc immunostaining. They showed that postsynaptic proteins such as GluA1 and PSD95 expression levels positively correlate with synaptic weight, and their expression level remains constant per unit area of dendrites under different release conditions. Combined with FM imaging and dendritic spine calcium imaging, they show that per unit dendrite Calcium influx (mediated by NMDA R) in individual dendritic branches remains constant under basal conditions but low release probability dendrites favor calcium influx during burst transmission. Using magnesium imaging, they showed that intracellular magnesium concentration is negatively correlated with synaptic release probability. Finally, using EM studies, they show that magnesium supplements in aged rodents increase functional synapses. Overall, this is a fascinating study, following their previous studies (e.g., Slutsky I et al. 2010 Neuron), suggesting magnesium might be a therapeutic target for enhancing learning and memory. The paper's conclusions are supported by the data presented and will be interesting to the community. Nevertheless, I have the following comments.

1. The data presented in the paper is very intense, and the writing of the article is tough to understand, except for those experts with strong computational backgrounds. Despite years of experience in electrophysiology and synaptic physiology, I had to read the paper many times to understand it. I must admit that I still don't understand certain calculations. I recommend the authors consider revising the paper to make it more readable for a wider audience. It might help increase the impact of the study.

We really appreciate your careful review of our manuscript and your concern on its readability.

In the preparation of this manuscript, we faced two key challenges in making the content accessible to a general audience. First, to ensure the robustness of our conclusions, we incorporated intensive experimental results as a foundation for our major findings. Second, in addressing the organizational principle of nearby synapses along a dendritic branch, we utilized multiple calculations to precisely describe the biophysical phenomena. Given the inherent complexity of the problem, conveying this concept in a straightforward manner proved challenging. Despite this,

we dedicated substantial effort to enhance readability, undertaking multiple rounds of revisions based on suggestions from synapse physiologists and systems biologists before formal submission. Despite this, we agree that the materials provided in the manuscript are intense. In response to your concern, we decided to include the summary schematics from the original **Fig. S9** into the new main **Fig. 8**, aiming to facilitate a clearer understanding for readers.

New Fig. 8 | Summary of a $[Mg^{2+}]_i$ -mediated pathway for regulating the synaptic configuration

2. The second question I have is that the data presented suggested that there is a “homeostasis” of presynaptic release and postsynaptic protein distributions. Although the authors showed that some signaling pathways are involved, it is still a mystery to me how the proteins are redistributed under different conditions, e.g., high intracellular magnesium.

There exists a homeostasis of presynaptic vesicle release (Pr) and postsynaptic AMPAR distribution along individual dendritic branches.

On the presynaptic side, the elevation of intracellular Mg^{2+} levels leads to an increased protein level of Ca^{2+} -sensitivity-related proteins (CaSPs) across all boutons, converting silent boutons into functional ones. Simultaneously, intracellular Mg^{2+} downregulates Pr . This combination of upregulating low- Pr synapses and downregulating high- Pr synapses represents a crucial mechanism contributing to the homeostatic redistribution of Pr . The schematic pathway, as depicted in the old **Fig. S9** (now new **Fig. 8b** in the revised manuscript), illustrates this process (PMID: 26184109).

At postsynaptic sites, elevating Mg^{2+} levels induces a homeostatic redistribution of GluA2-containing AMPAR and PSD95 among postsynaptic spines. While direct experiments addressing these redistributions are still needed, previous studies have provided insights. As silent presynaptic

boutons become functional and release transmitters, Ca^{2+} influx through NMDARs at postsynaptic spines activates the CaMKII pathway, recruiting non-synaptic AMPARs from neighboring membrane or cytoplasm to the postsynaptic density (for reviews, PMID: 36056211, 33160201). An essential pathway involves the recruitment of AMPARs from neighboring synapses to functional synapses (PMID: 33503435, 34949992, 33160201). Consequently, there would be a redistribution of postsynaptic receptors while maintaining a constant total amount. We speculate such a mechanism may underlie the pre- and postsynaptic homeostatic coordination. Further exploration of molecular mechanisms is still needed in future studies.

3. The central nervous system is exceedingly diverse, with hundreds of types of neurons and trillions of synapses, some of which need high fidelity, for example, the auditory and visual systems. How will the general level changes of magnesium in the brain, i.e., by supplement magnesium, affect other types of synapses than the hippocampal ones?

Thank you for presenting this insightful and crucial question.

Firstly, the impact of brain Mg^{2+} on synaptic plasticity is contingent on the type of synapse. In our current and prior studies, we demonstrated a significant enhancement in the plasticity of CA3–CA1 hippocampal synapses with elevated brain Mg^{2+} levels (PMID: 20152124). Collaborative research using a monocular deprivation paradigm revealed that increasing brain Mg^{2+} levels in adult mice enhances excitatory synapse plasticity in L2/3 of the primary visual cortex (V1), resembling levels observed in juveniles (PMID: 26282667).

Secondly, the influence of Mg^{2+} on synaptic plasticity is region-specific within the brain. Elevated brain Mg^{2+} levels effectively enhance synaptic plasticity in certain regions, such as the hippocampus (PMID: 20152124), infralimbic prefrontal cortex (PMID: 22016520), and V1 (PMID: 26282667). However, it does not affect synaptic plasticity in the basolateral amygdala (BLA) (PMID: 22016520).

Our working hypothesis posits that brain Mg^{2+} primarily modulates highly plastic synapses in specific brain regions. Yet, further investigations are needed to discern the differential effects of brain Mg^{2+} levels in various regions during different cognitive processes.

Beyond its implications for basic sciences, we recognize the critical importance of this question in translational research. Human studies have revealed that MgT treatment improves cognitive functions and emotional regulation without affecting sensory perception and motor control (PMID: 26519439, 32162987; DOI: 10.1016/j.jalz.2017.06.1865).

4. Release probability is the central question of this paper; I wasn't entirely clear, maybe I missed reading it, how the release probability at a single synaptic bouton was calculated using FM imaging.

Our laboratory developed a protocol to precisely measure the Pr of individual presynaptic boutons at the quantal resolution (Slutsky et al., 2004; PMID: 15572114). Briefly, in this protocol, we follow a two-step process to measure and calculate Pr of a single bouton using FM imaging. First, we perform FM labeling of released vesicles by delivering 30 action potentials (APs) at 0.5 Hz via field stimulation to elicit vesicle release. The inner membrane of exocytosed vesicles is exposed to extracellular fluid, incorporating FM dye via endocytosis. Consequently, each released vesicle presumably contains an equal amount of FM molecules. We then image individual boutons and acquire the fluorescence of FM dye in each bouton (ΔF).

Second, we calibrate the FM fluorescence of a single vesicle, *i.e.*, the quantal size. To achieve this, we deliver only 1 AP to induce endocytosis of FM dye. Conceptually, the majority of releasable boutons secrete only 1 vesicle, but a few may release 2 or more vesicles. By measuring the fluorescence of individual boutons, we obtain a distribution of fluorescence, which can be curve-fitted by a multi-Gaussian distribution (PMID: 10448213, 15572114). The peaks of this distribution exhibit a constant interval among each other. This constant interval is the quantal size represented by FM fluorescence (F_Q), indicating the amount of FM contained in a single released vesicle (PMID: 10448213, 15572114). With F_Q , we can convert FM fluorescence in a single bouton to the number of released vesicles. Thus, Pr of individual boutons is calculated by $(\Delta F/F_Q)/30$ (PMID: 15572114).

In our manuscript, we included this protocol in 'Detection of vesicle turnover' of Methods. It reads:

“Detection of vesicle turnover

*To visualize vesicle turnover in single boutons, we employed styryl dyes including FM1-43 (10 μ M), FM4-64 (20 μ M), or FM5-95 (20 μ M), following the previously described protocols. The labeling procedure for vesicle turnover consisted of three steps: loading, washout, and unloading (refer to **Fig. S1a**). Working solutions were pre-warmed and perfused into the chamber through a peristaltic pump. During loading, FM molecules were taken up by released vesicles that were elicited by different stimulation patterns such as 30AP@0.5Hz, 6 \times 5AP@100Hz (or other bursting patterns as shown in **Fig. S1b**), or 600AP@10Hz (referred to as the 'maximum stimulation' protocol). Action potentials were evoked by a field stimulus (1 ms, 50 mA per pulse) generated by a stimulus isolator (A385; WPI) and delivered via parallel platinum electrodes placed at an 8-mm distance (RC-27NE2; Warner). After loading, nonspecific FM labeling was washed out using a solution containing low $[Ca^{2+}]_o$ (0.4 mM) and the 'quencher' ADVASEP-7 (300 μ M). Importantly, the washout process did not affect the endocytosed FM in vesicles.*

To exclude artifacts introduced by unhealthy boutons, an unloading procedure was

performed to test whether the dyes could be exocytosed after delivering a 480AP@2Hz stimulation. Images were captured (as described in the 'Imaging systems' section) after washout (F_1) and unloading (F_2), and the difference ($F_2 - F_1$) was used to quantify the number of released vesicles in single boutons.

*During the loading and unloading procedures, kynurenic acid (200 μ M) or a combination of NBQX (10 μ M) and AP5 (50 μ M) was applied to block synaptic transmission. The **Pr** or **Pr_{burst}** was measured using quantal analysis as previously described. Briefly, the fluorescence of a quantal release (F_Q) was measured by FM staining elicited by a single AP. The number of released vesicles per bouton (N_v) after 30 APs was calculated by dividing F_Q by its fluorescence, and subsequently, **Pr** was determined by $N_v/30$.”*

4. (...) It also appears to me that only evoked releases are considered. Are there any components of spontaneous releases at these synapses? Does the spontaneous release unload the loaded FM dyes? Moreover, how to combat and correct photobleaching during the imaging process?

Primarily, cultured neurons exhibit spontaneous network activity. However, during the FM dye staining session, we effectively suppress this network activity using NBQX (an AMPAR blocker) and AP-5 (an NMDAR antagonist). This blockade results in an exceptionally low frequency of spontaneous action potentials. Consequently, FM loading predominantly occurs upon evoked action potentials induced by field stimulations, making the amount of FM loaded in the absence of field stimulation negligible.

To mitigate photobleaching, we constrained the laser power of the confocal microscopy to a minimal level, yielding a less than 2% reduction in fluorescence intensity between two successive and identical imaging sessions. Additionally, we maintained consistent imaging parameters throughout the 'quantal size measurement' experiment, ensuring that the measured quantal size is equally affected by photobleaching. This approach serves to correct any potential errors introduced by photobleaching.

5. I wasn't entirely convinced by the term “silence synapse” used in the study. I think the authors defined “silence synapse” for those loaded puncta that show a release probability of <0.04 , considered a silence synapse. This definition is different from the classical “silence synapses” proposed previously. (e.g., Liao et al. 1995, Nature; Isaac et al. Neuron 1995). I understand that the authors also showed a correlation of postsynaptic protein expression level in correlation with the release probability.

Thank you for highlighting a potential source of confusion in nomenclature that may arise for readers.

As you rightly pointed out, the classic definition of "silent synapses" is traditionally based on the absence of postsynaptic AMPAR, leading to failure of synaptic transmission (PMID: 7760933, 7646894). Meanwhile, synapses can also be considered silent due to failure of releasing transmitters upon action potentials, termed "presynaptic silence" (for a review, PMID: 21908849).

In our previous studies, we observed a significant portion of presynaptic boutons that cannot faithfully release transmitters during basal transmission (PMID: 15572114), primarily attributed to a lack of presynaptic Ca^{2+} -sensitivity-related proteins (CaSPs) (PMID: 26184109). In the current study, we found that low-AMPA postsynaptic sites correspond to low-*Pr* boutons, demonstrating a positive association between the amount of postsynaptic AMPAR and the release probability (*Pr*) at individual synapses (**Fig. 2**), which is consistent with previous findings showing a strong presynaptic and postsynaptic association (PMID: 9221783, 17596435, 22683683, 17237775, 21555073).

Our study predominantly focuses on the functional/nonfunctional (silent) state of presynaptic boutons, and we consistently use the term "silent boutons" throughout most of the manuscript. In response to your comment, we have removed the term "silent synapses" from the revised manuscript to avoid any potential confusion in nomenclature, ensuring clarity for readers.

5. (...) I wonder if some other endocytic process than the retrievals of synaptic vesicles from the plasma membrane during the loading process would also be labeled, and those probably would have a very low possibility of being re-released.

In our experiments, we implemented several measures to minimize non-specific staining of FM dyes, including non-vesicular endocytic processes. Firstly, we restricted the time of FM presence in the extracellular fluid to about 1 min during the induction of vesicle release through action potentials (APs), reducing the extent to non-specific FM staining on membrane (PMID: 15572114). Secondly, we observed that puncta-like staining of FM dyes without APs is extremely low after a 1-min exposure to FM dyes, indicating a very low level of non-vesicular endocytosis. Thirdly, following the AP-induced FM staining procedure, we consistently performed an FM de-staining procedure by delivering 480 APs at 2 Hz, which can fully turnover all the releasable vesicles (PMID: 10448213). This step allowed us to examine whether individual FM^+ puncta can undergo exocytosis upon APs again (PMID: 15572114). Any puncta-like FM staining that remains after de-staining is excluded by image subtraction, further excluding the FM staining by AP-independent, non-vesicular endocytic processes. In conclusion, non-specific staining of FM dyes in our experiments is either negligible or, at the very least, not a primary concern.

6. For establishing presynaptic release and postsynaptic Calcium signals, they used GCaMP6f; I was wondering if they could also observe presynaptic Ca^{2+} dynamic changes in these experiments (since the videos have been acquired).

Yes, we visualized and quantified presynaptic Ca^{2+} dynamic changes upon APs using a vector transducing synaptophysin-fused GCaMP6f (SypGCaMP6f) (old **Fig. S4**, but now new **Fig. S5**), expressed on presynaptic vesicles to report presynaptic Ca^{2+} activity (old **Fig. S4a**, but now new **Fig. S5a**).

Following the elevation of $[\text{Mg}^{2+}]_i$ by changing extracellular Mg^{2+} levels, we observed no changes in the amplitude of presynaptic Ca^{2+} influx upon a single AP or a 5-AP burst (old **Fig. S4d–i**, but now new **Fig. S5d–i**). This suggests that the decrease in average *Pr* of boutons after elevating Mg^{2+} levels may not be attributed to a reduction in evoked Ca^{2+} influx. Technically, despite the validity of GCaMP6f in reporting amplitude changes of Ca^{2+} events, it is important to note that our visualization of the presynaptic Ca^{2+} dynamics is limited due to the slow chemical kinetics of GCaMP6f- Ca^{2+} reaction. The biophysical process of presynaptic Ca^{2+} dynamics is in milliseconds, comparing that of GCaMP6f- Ca^{2+} reaction in hundreds of milliseconds (PMID: 23868258, 18817727).

Responding to your question, we have decided to relocate panels in the old **Fig. S4j–l** to the main **Fig. 4** (as new **Fig. 4f–h**) to enhance clarity and facilitate a direct comparison between the effects of intracellular Mg^{2+} and Ca^{2+} on *Pr* for the readers.

The old **Fig. S4** and new **Fig. 4** are as follows.

Old Fig. S4 (but now as new Fig. S5) | Measurements of *Pr* and evoked presynaptic Ca²⁺ influx in single boutons

a, Experimental design. Left, Schematic to show FM5-95 labeling in the boutons transfected by CaMKII α -Synaptophysin-GCaMP6f (SypGCaMP6f). Right, Experimental procedures for measuring evoked presynaptic Ca²⁺ influx ([Ca²⁺]_{evoked}) and vesicle turnover (*Pr*) in the same synapses. In loading session, 30AP@0.5Hz or 6 trains of 5AP@100Hz is delivered via field stimulation (FS) to measure *Pr* or *Pr*_{burst}. *F*₁, fluorescence of FM dye loaded in boutons. *F*₂, residual fluorescence after FM dye unloading.

b, *Pr* distribution showed no difference in transfected (SypGCaMP6⁺) and non-transfected (SypGCaMP6⁻) boutons (*n* = 406, 232 from 5 repeats). Inset, discrete data points in violin plots, where black and magenta lines indicate median and quartiles. Kolmogorov-Smirnov test, *P* = 0.86.

- c**, Left, average traces of Ca^{2+} influx of boutons (visualized by SypGCaMP6f) evoked by various input patterns ($n = 302, 387$ boutons from 5, 5 repeats). Traces were averaged from 30 sweeps of the boutons. Right, relationship between $[\text{Ca}^{2+}]_{\text{evoked}}$ and AP number. Solid lines, linear regressions. Dashed line, extension of the black line. The frequency of APs in all bursts was 100 Hz.
- d**, Left, representative images of 1AP-evoked peak $\Delta\text{F}/\text{F}_0$ in the same boutons with various $[\text{Ca}^{2+}]_o/[\text{Mg}^{2+}]_o$ ratios in working solution (WS). Right, stacked 30 sweeps of evoked Ca^{2+} influx (thin lines) and their average traces (thick lines).
- e**, Cumulative distributions of $[\text{Ca}^{2+}]_{\text{evoked}}$ of boutons under conditions of $[\text{Ca}^{2+}]_o/[\text{Mg}^{2+}]_o$ 1 and 4 ($n = 217, 253$ boutons from 4, 4 repeats).
- f**, Plot of average $[\text{Ca}^{2+}]_{\text{evoked}}$ against $[\text{Ca}^{2+}]_o/[\text{Mg}^{2+}]_o$ ($n = 217, 253$ boutons from 4, 4 repeats).
- g–i**, The same boutons as in (**d–f**), but with the input of 5AP@100Hz bursts. In (**g**), stacked 6 sweeps and their average traces were shown.

(the old **Fig. S4j–l** panels are moved to the main **Fig. 4** as new **Fig. 4f–h**)

- j**, Left, images to show 1AP-evoked Ca^{2+} influx (peak $\Delta\text{F}/\text{F}_0$) and FM5-95 fluorescence (masked by bouton morphology, indicative of **Pr**) in the same boutons. Upper right, Peak $\Delta\text{F}/\text{F}_0$, Ca^{2+} influx FM5-95 puncta, and **Pr** in single boutons. Lower right, Fractions of boutons with detectable Ca^{2+} influx, referred to as $\text{Ca}^{2+}(+)$, and vesicle turnover ($Pr \geq 0.04$) ($n = 3$ repeats). $\Delta\text{F}/\text{F}_0$ puncta and Ca^{2+} traces were averaged from 30 repeats. Envelope, SEM.
- k**, Plot of **Pr** against $[\text{Ca}^{2+}]_{\text{evoked}}$ (the amplitude of 1AP-evoked Ca^{2+} influx) in boutons ($n = 243$ boutons from 5 repeats, physiological Mg^{2+} condition). Inset, logarithmic plot. Line, double logarithmic fitting, $R^2 = 0.19$, $P < 0.0001$.
- l**, Similar to (**k**) but using working solution of $[\text{Ca}^{2+}]_o$ 4.8 mM and $[\text{Mg}^{2+}]_o$ 1.2 mM (ratio 4:1) to immediately increase $[\text{Ca}^{2+}]_{\text{evoked}}$ ($n = 146$ boutons from 3 repeats). Line, double logarithmic fitting, $R^2 = 0.33$, $P < 0.0001$.
- Data are presented as mean \pm SEM. One way ANOVA followed by *post hoc* Bonferroni's tests (**c**, **f**, **i**). Kolmogorov-Smirnov tests (**c**, **e**, **h**). Significance: *NS*, no significance.

New Fig. 4 | Intracellular Mg^{2+} regulates synaptic configurations by inhibiting release probability of individual synapses

a, Time courses of changes in synaptic configuration and $[Mg^{2+}]_i$ (f.u.) in neurites after various treatments ($n = 5-9$ repeats per point). Elevating $[Mg^{2+}]_o$ under normal condition (Left), lowering $[Mg^{2+}]_o$ under elevated Mg^{2+} condition (middle), and administering imipramine ($1\mu M$) under normal condition (right). f.u., fluorescence unit. **b**, Experimental design for measuring $[Mg^{2+}]_i$ and Pr in the same boutons. **c**, Representative boutons of similar size to show an inverse trend between Pr (FM4-64) and $[Mg^{2+}]_i$ (MgGrn). **d**, Measurement of Pr and $[Mg^{2+}]_i$ (mM) in boutons ($n = 204, 444$ from 3, 4 repeats). Inset, plot of Pr against $1/[Mg^{2+}]_i$. Double logarithmic fitting, $R^2 = 0.54$, $P < 0.0001$. **e**, Measurement of \bar{Pr} and $[Mg^{2+}]_i$ (mM) of the boutons at individual dendritic branches ($n = 58, 53$ branches from 4, 4 repeats). **f-h**, Concurrent measurement of Pr and single AP-evoked Ca^{2+} influx in the same boutons using synaptophysin-

fused GCaMP6f (SypGCaMP6f). **f**, Left, images to show 1AP-evoked Ca^{2+} influx (peak $\Delta\text{F}/\text{F}_0$) and FM5-95 fluorescence (masked by bouton morphology, indicative of **Pr**) in the same boutons. Upper right, Peak $\Delta\text{F}/\text{F}_0$, Ca^{2+} influx FM5-95 puncta, and **Pr** in single boutons. Lower right, Fractions of boutons with detectable Ca^{2+} influx, referred to as $\text{Ca}^{2+}(+)$, and vesicle turnover ($\text{Pr} \geq 0.04$) ($n = 3$ repeats). $\Delta\text{F}/\text{F}_0$ puncta and Ca^{2+} traces were averaged from 30 repeats. Envelope, SEM. **g**, Plot of **Pr** against $[\text{Ca}^{2+}]_{\text{evoked}}$ (the amplitude of 1AP-evoked Ca^{2+} influx) in boutons ($n = 243$ boutons from 5 repeats, physiological Mg^{2+} condition). Inset, logarithmic plot. Line, double logarithmic fitting, $R^2 = 0.19$, $P < 0.0001$. **h**, Similar to (**g**) but using the working solution of $[\text{Ca}^{2+}]_o$ 4.8 mM and $[\text{Mg}^{2+}]_o$ 1.2 mM (ratio 4:1) to immediately increase $[\text{Ca}^{2+}]_{\text{evoked}}$ ($n = 146$ boutons from 3 repeats). Line, double logarithmic fitting, $R^2 = 0.33$, $P < 0.0001$. Data are mean \pm SEM. Two-sided unpaired *t* tests (vs. time 0) for (**a**). Significance: NS, no significance, * $P < 0.05$, ** $P < 0.01$, *** $P < 0.001$.

7. The authors used many immunostainings for postsynaptic proteins. The source of antibodies should be provided.

In our initial submission, we adhered to the editorial policy by including the source of antibodies in both the Methods section (refer to the Reagent table) and the Reporting Summary.

Reagent table (Methods)

Rabbit polyclonal anti-ERC1b/2 (ELKS)	Synaptic Systems	Cat#143003
Mouse monoclonal anti-GluA2 (clone 6C4)	Invitrogen	Cat#32-0300
Guinea pig polyclonal anti-MAP2	Synaptic Systems	Cat#188 004
Mouse monoclonal anti-Munc13-1 (clone 266B1)	Synaptic Systems	Cat#126 111
Rabbit polyclonal anti-Munc13-1	Synaptic Systems	Cat#126103
Mouse monoclonal anti-PSD95 (clone 7E3-1B8)	Millipore	Cat#CP35
Mouse monoclonal anti-Rab3a (clone 42.2)	Synaptic Systems	Cat#107111
Rabbit polyclonal anti-Rab3a	Synaptic Systems	Cat#107102
Rabbit polyclonal anti-RIM1	Synaptic Systems	Cat#140003
Mouse monoclonal anti-Synaptophysin (clone SY38)	Millipore	Cat#MAB5258
Guinea pig polyclonal anti-Synaptophysin	Synaptic Systems	Cat#101004
Mouse monoclonal anti-Synaptotagmin1 (clone 41.1)	Synaptic Systems	Cat#105011
Mouse monoclonal anti-Syntaxin1 (clone 78.2)	Synaptic Systems	Cat#110011
Guinea pig polyclonal anti-VGLUT1	Millipore	Cat#AB5905
CF488A Goat Anti-Guinea pig IgG (H+L)	Biotium	Cat#20017
CF488A Goat Anti-Mouse IgG (H+L)	Biotium	Cat#20018
CF488A Goat Anti-Rabbit IgG (H+L)	Biotium	Cat#20019
CF555 Goat Anti-Guinea pig IgG (H+L)	Biotium	Cat#20036
CF555 Goat Anti-Mouse IgG (H+L)	Biotium	Cat#20231
CF555 Goat Anti-Rabbit IgG (H+L)	Biotium	Cat#20232
CF640R Goat Anti-Guinea pig IgG (H+L)	Biotium	Cat#20085
CF640R Goat Anti-Mouse IgG (H+L)	Biotium	Cat#20175
CF640R Goat Anti-Rabbit IgG (H+L)	Biotium	Cat#20176

Reporting Summary:

Antibodies

Antibodies used

Rabbit polyclonal anti-ERC1b/2 (ELKS), Synaptic Systems, Cat#143003
 Mouse monoclonal anti-GluA2 (clone 6C4), Invitrogen, Cat#32-0300
 Guinea pig polyclonal anti-MAP2, Synaptic Systems, Cat#188 004
 Mouse monoclonal anti-Munc13-1 (clone 266B1), Synaptic Systems, Cat#126 111
 Rabbit polyclonal anti-Munc13-1, Synaptic Systems, Cat#126103
 Mouse monoclonal anti-PSD95 (clone 7E3-1B8), Millipore, Cat#CP35
 Mouse monoclonal anti-Rab3a (clone 42.2), Synaptic Systems, Cat#107111
 Rabbit polyclonal anti-Rab3a, Synaptic Systems, Cat#107102
 Rabbit polyclonal anti-RIM1, Synaptic Systems, Cat#140003
 Mouse monoclonal anti-Synaptophysin (clone SY38), Millipore, Cat#MAB5258
 Guinea pig polyclonal anti-Synaptophysin, Synaptic Systems, Cat#101004
 Mouse monoclonal anti-Synaptotagmin1 (clone 41.1), Synaptic Systems, Cat#105011
 Mouse monoclonal anti-Syntaxin1 (clone 78.2), Synaptic Systems, Cat#110011
 Guinea pig polyclonal anti-VGLUT1, Millipore, Cat#AB5905

7. (...) In addition, Fig. 5b showed eight post hoc immune staining for various postsynaptic proteins after FM unloading after stimulation. I wonder how this was done. I wasn't clear about this after reading the method section.

This experiment unfolded in two main stages. First, in living cultures, we labeled the released vesicles of presynaptic boutons with an FM dye induced by action potentials for *Pr* measurement. Subsequently, after capturing FM signals, we fixed the cultures and conducted *post hoc* immunostaining to label multiple presynaptic proteins in the same boutons. To achieve this, we employed rounds of staining and elution cycles. In each round of immunostaining, 2–3 primary antibodies for corresponding presynaptic proteins, along with fluorophore-conjugated secondary antibodies, were successively incubated and visualized through confocal imaging. Following this, cultures were delicately transferred to a solution containing SDS detergent to fully elute all antibodies from epitopes. The incubation time was optimized to ensure that the residual fluorescent signal in boutons was less than 1% of the initial signal (see representative images of VGLUT1 immunoreactivity before and after SDS elution; new **Fig. S6b**). Additionally, we validated that the immunoreactivity of epitopes was only minimally compromised even after four rounds of staining/eluting procedures (see the comparison of VGLUT1 immunofluorescence at rounds 1 and 5; new **Fig. S6c**). We iterated through these staining/eluting cycles for several rounds to label all eight presynaptic proteins. Importantly, between adjacent rounds, a common antibody served as a landmark for the registration of proteins in individual boutons.

The procedures were included in Histology section of Methods in the manuscript as follows.

*“**Cultured neurons.** A strategy of retrograde immunofluorescence was employed for multiple rounds of labeling presynaptic proteins in a synaptic network (for a schematic, see **Fig. S6a**), using the method*

of 'Single-synapse analysis using FM1-43 and immunofluorescence imaging array' (SAFIA) as previously described³⁸. After imaging FM1-43 in the areas of interest (AOIs) with dimensions of 58.9 μm \times 58.9 μm (see 'Detection of vesicle turnover'), the culture coverslips were incubated in normal Tyrode solution with $[\text{Ca}^{2+}]_o$ at 1.2 mM for 20 min. Then, they were fixed in a solution of 1% paraformaldehyde (E.M.S.), 0.01% glutaraldehyde (Alfa Aesar), and 4% sucrose in 1 \times PBS (pH 7.4) for 1 h at room temperature (RT). The fixatives were gently washed out several times using Tris buffer (pH 7.6), which consisted of Tris Base 25 mM, Tris-HCl 25 mM, and NaCl 150 mM (filtered through a 0.22- μm film before use).

Next, the coverslips were permeabilized and blocked in a freshly prepared blocking solution (in Tris buffer) containing 1% BSA, 0.1% saponin, and 300 μM ADVASEP-7 for 1.5 h at RT. This was followed by several rounds of immunostaining. In each round, two to three primary antibodies (1:200–1:500) are incubated for 2–4 h at RT, followed by incubation with fluorophore-conjugated secondary antibodies (1:200–1:500) for 2 h at RT. Confocal fluorescent images were then captured in the AOIs, with light-field DIC images used for recognizing landmarks. After each round of staining, the antibodies were eluted by a stripping buffer (0.2 M NaOH and 0.015% SDS in deionized water) for 20 min at RT, repeated twice. The coverslips were rinsed in Tris buffer for over 1 h at RT to wash out SDS. Extreme care was taken to avoid cell detachment during this process. Confocal imaging validated that less than 1% fluorescent signal of presynaptic proteins remained in individual boutons after elution as compared to before (**Fig. S6b**). The coverslips were then subjected to the next round of immunostaining using 2–3 primary antibodies, with one antibody used as an internal quality control and landmark for image registration in post hoc analysis. Typically, a maximum of 4–5 rounds of immunostaining could be performed on a culture coverslip (**Fig. S6c**)."

Nevertheless, as you pointed out, to make the experimental procedures more straightforward, we have drawn a schematic to illustrate the staining procedures (new **Fig. S6a**), which would help broader audience to quickly learn how the experiment is performed.

New **Fig. S6a–c** panels are shown after **Point-8**.

8. A lot of the data shown are cropped single-button images. It would be helpful if the author could provide some level of unprocessed images as supplemental data.

As you suggested, we have provided some unprocessed representative images in supplemental information to show FM1-43 staining and *post hoc* immunostaining of 8 presynaptic proteins in

the same presynaptic boutons under physiological ($[Mg^{2+}]_o$ 0.8 mM) and elevated Mg^{2+} ($[Mg^{2+}]_o$ 1.2 mM) conditions (new Fig. S6b, c; related to Fig. 5b–f), as follows.

New Fig. S6 | Co-labeling of released vesicles and presynaptic proteins.

a, Schematic to show experimental procedures (for details see Methods). Ab, antibody. **b**, Comparison of VGLUT1⁺ immunofluorescence before and after SDS-mediated antibody elution. **c**, Comparison of VGLUT1⁺ immunofluorescence in round 1, 3, and 5 of staining/eluting cycles. Notably, VGLUT1 immunoreactivity is similar after 2 and 4 rounds of elution (Round 3, 5 vs. Round 1). **d**, **e**, Representative confocal images from physiological (**d**) and elevated Mg^{2+} (**e**) conditions to show labeling of FM1-43 and immunofluorescence of multiple presynaptic proteins in the same region.

Reviewer #3 (Remarks to the Author):

Zhoe et al. investigated the role of intracellular magnesium in optimizing synaptic transmission efficiency and plasticity in the hippocampus, which is important for learning and memory processes. By examining the connectivity of individual synapses within dendritic arbors, they demonstrated an organizational principle that directs nearby synapses to generate specific computational features using two modes favoring “memory” or “learning”, and that increasing intracellular magnesium levels enhances synaptic transmission and plasticity, leading to improved learning and memory. The study was based on classical physiological approaches, and similar experiments were repeated. Also, the gap between the micro-level behavior of synapses and the macro-level behavior of learning is not yet bridged. As a result, the overall impression was one of blurred focus and a sense that the discussion was not theoretically robust. The (intentional?) addition of the aging perspective (which seems an ulterior motive aimed at making a good impression) has also contributed to the impression that the discussion is scattered. This is due partly to the fact that the landing point of this study is unclear, despite the complexity of the experiments. Apart from these general impressions, the physiological validity of the detailed experimental conditions remained highly questionable, as follows.

In addressing the concerns you raised regarding the presentation of our findings, we would like to provide some background on the current study and explain the organization of the manuscript.

Over the past two decades, our laboratory has been dedicated to exploring the beneficial effects of Mg^{2+} in maintaining brain health and mitigating cognitive decline associated with aging and neurodegenerative diseases. To contextualize our current work, we have briefly summarized our previous research at the beginning of this "response to reviewers" document. Notably, what might seem like a gap between micro-level synaptic behavior and macro-level animal behavior in the current manuscript has been addressed at different levels in our earlier studies. For instance, at the synapse level, we demonstrated enhanced plasticity and density of cultured hippocampal synapses (PMID: 15572114, 26184109). At the circuitry level, we showed enhanced plasticity of CA3–CA1 Schaffer collateral projection synapses by elevating Mg^{2+} levels in acute brain slices (PMID: 20152124). At the levels of body metabolism and behavior, we validated the high bioavailability of Magnesium L-Threonate (MgT) compound for increasing brain Mg^{2+} levels and illustrated the reversible effect of brain Mg^{2+} levels on cognitive functions in adult animals (PMID: 20152124). Additionally, we reported the positive impact of MgT supplementation on cognitive functions in aged animals (PMID: 20152124) and Alzheimer’s disease model animals (PMID: 25213836). These studies collectively span various levels, including protein, synapse, circuitry, and behavior, addressing micro to macro aspects.

In the current manuscript, the landing point is to tackle a longstanding question in the field: how nearby synapses at individual dendritic branches are organized to generate distinct synaptic

computations, essentially regulating the "transfer function" of synapses at a dendritic branch. This question is crucial as dendritic branches are considered the basic computational unit for information processing underlying cognitive functions. Our findings reveal that intracellular Mg^{2+} serves as an endogenous factor in organizing nearby synapses from different presynaptic neurons, influencing the configuration of synaptic connectivity at individual dendritic branches. This, in turn, determines the "transfer function" of each dendritic branch. We introduced a general principle of synaptic organization at dendritic branches, proposing that nearby synapses are consistently organized along an individual branch to maintain a constant total presynaptic strength (the first part of the Discussion). It's important to note that the concept of *configuration* is more generalized, with the regulatory effect of intracellular Mg^{2+} serving as a significant example. As different configurations impart distinct features of synaptic computations to an individual branch, the transition between configurations becomes crucial for branch-specific synaptic computations during information processing for learning and memory. Significantly, our principle hints at the possibility of other essential endogenous factors, beyond intracellular Mg^{2+} , regulating synaptic configuration. Such factors could be promising candidates for anti-brain aging and anti-neurodegeneration strategies, providing a novel avenue for drug exploration. Overall, we believe that this study offers precise and comprehensive mechanisms, serving as a cornerstone in our series of studies on the beneficial effects of brain Mg^{2+} in maintaining brain health.

We hope that these explanations could provide clarity on the significance of our study and its implications for translational research.

(i) The distribution per branch fluctuates in Fig. 1, but under the high Mg^{2+} condition, the plots move to positions that are not present under the low Mg^{2+} condition, can this be considered physiological?

Following the elevation of extracellular Mg^{2+} concentration ($[Mg^{2+}]_o$), the synaptic connectivity configuration tends to diverge from that under normal $[Mg^{2+}]_o$ conditions, yet there remains a significant overlap between the two conditions (**Fig. 1b**). Analyzing the percentages of dendritic branches within this overlap range (*i.e.*, \overline{Pr} is between 0.23–0.51), we observed that 75.76% (50 out of 66 branches) and 53.73% (36 out of 67 branches) of dendritic branches under normal and elevated Mg^{2+} conditions, respectively, fall within this range. This suggests that the distribution of Pr undergoes continuous changes among dendritic branches after Mg^{2+} levels are elevated, rather than exhibiting a complete separation under the two conditions.

(i) (...) Under actual in vivo conditions, are the Mg^{2+} concentrations used in Figs. 1-6 feasible?

Mg^{2+} stands as the second most abundant intracellular mineral after K^+ and is present in substantial amounts in the cerebrospinal fluid (CSF) of both rodents (around 0.8 mM) and humans (around

1.0–1.2 mM in healthy individuals) (for a review, PMID: 29920011). The concentrations of 0.8–1.2 mM used in the current study are supported by multiple lines of evidence. Under *in vivo* conditions, $[Mg^{2+}]_o$ in the CSF of animal brains can increase by 21% above control (*i.e.*, from ~1 mM to 1.2 mM) 5.5 hours after intravenous injection of $MgCl_2$ or $MgSO_4$ (PMID: 5877425). Similarly, $[Mg^{2+}]_o$ in the CSF of human brains can be raised from 0.95 ± 0.11 to 1.13 ± 0.19 mM by intravenous injection of $MgSO_4$ (PMID: 9339404). In our studies, we demonstrated in living rats that oral MgT treatment can elevate $[Mg^{2+}]_o$ in the CSF by 15% (~0.2 mM) through water consumption (PMID: 20152124). Other studies in living mice, using advanced techniques to measure brain interstitial $[Mg^{2+}]_o$, reported that during the transition from wakefulness to sleep, $[Mg^{2+}]_o$ quickly increases by ~0.13 mM from a baseline of ~0.7 mM; conversely, during the transition from sleep to wakefulness, $[Mg^{2+}]_o$ decreases by ~0.11 mM from a baseline of ~1 mM (PMID: 27126038). Importantly, they demonstrated variations in $[Mg^{2+}]_o$ among individual animals, ranging from ~0.5–1.2 mM (PMID: 27126038), indicating that $[Mg^{2+}]_o$ can vary by up to twofold in mouse brains. Additionally, during the transition from wakefulness to isoflurane anesthesia in mice, brain $[Mg^{2+}]_o$ can increase by ~0.44 mM (ranging from ~0.5–1.5 mM in different mice), illustrating a notable brain state-dependent change in $[Mg^{2+}]_o$ (PMID: 27126038).

Therefore, the concentrations of $[Mg^{2+}]_o$ employed in our *in vitro* model system, 0.8–1.2 mM, fall within the physiologically relevant range observed under *in vivo* conditions.

(ii) What is the concentration of the high Mg^{2+} condition referred to (the low Mg^{2+} condition is described in the text)?

In our study, we denoted 1.2 mM $[Mg^{2+}]_o$ as the "elevated Mg^{2+} condition", while 0.8 mM $[Mg^{2+}]_o$ as the "physiological Mg^{2+} condition". The initial mention of these conditions is found in the Results section and is presented as follows.

“Here, we evaluated the effect of changing $[Mg^{2+}]_o$ on synaptic configurations. By elevating $[Mg^{2+}]_o$ in the culture medium from 0.8 mM (corresponding to a typical 'physiological Mg^{2+} condition' in the rodent brain) to 1.2 mM (corresponding to an 'elevated Mg^{2+} condition' within physiological range) for 4 h, we observed ...”

(ii) (...) It is not clear under what circumstances such high concentrations could occur.

As mentioned above (the second paragraph of the reply to **Question (i)**), the changes of $[Mg^{2+}]_o$ between 0.8–1.2 mM can occur under *in vivo* physiological conditions in rodent brains. In this regard, the "high" concentration is actually within physiological ranges *in vivo*.

(iii) Is it an appropriate experiment to cause a uniform increase in concentration in all spines in Fig. 3 and elsewhere?

As stated above, a uniform increase in brain $[Mg^{2+}]_o$ can occur in various physiological scenarios, including MgT supplementation, transitions between wakefulness and sleep, and transitions between wakefulness and isoflurane anesthesia. Therefore, an increase in $[Mg^{2+}]_o$ from 0.8 to 1.2 mM is considered a realistic change under physiological conditions *in vivo*. Functionally, this uniform elevation of $[Mg^{2+}]_o$ can gradually increase intracellular Mg^{2+} concentration ($[Mg^{2+}]_i$) in synapses over several hours, globally modulating their transmission and plasticity, as supported by the findings of the current study and our prior research (PMID: 15572114, 20152124).

(iv) The authors consistently refers to intracellular Mg^{2+} , but to what extent can the extracellular application of Mg^{2+} change intracellular Mg^{2+} ? Quantification is needed. (The study focuses on the intracellular Mg^{2+} concentration, but it does not examine the absorption of Mg^{2+} into cells or the measurement of intracellular Mg^{2+} concentrations.)

In our study, we utilized a modified chemical fluorescent Mg^{2+} probe, Magnesium Green (MgGrn) acetoxymethyl ester (AM), to visualize and quantify intracellular Mg^{2+} concentration ($[Mg^{2+}]_i$). The hydrophobic AM tail of MgGrn allows its penetration through intact neuronal membranes. Once inside the cytoplasm, intracellular esterases hydrolyze the AM groups, converting MgGrn into a highly fluorescent form, which is retained within the cytoplasm and indicates $[Mg^{2+}]_i$. The fluorescence images obtained through slow-scanning confocal microscopy quasi-linearly reflect $[Mg^{2+}]_i$ levels and chronic changes in $[Mg^{2+}]_i$. Throughout the manuscript, we used the intensity of intracellular MgGrn fluorescence to report $[Mg^{2+}]_i$ and quantify its changes resulting from various experimental conditions.

While we initially considered relative changes in $[Mg^{2+}]_i$ indexed by MgGrn fluorescence sufficient for addressing the points in our study, we have conducted a further analysis of the calibration experiment that had been performed alongside the intracellular Mg^{2+} imaging experiments. In the revised manuscript, we included the calibration curve in the supplemental figures (new **Fig. S4**) and convert the fluorescence of intracellular MgGrn into absolute concentrations throughout figures. For example, we showed that the 4-hour elevation of extracellular Mg^{2+} concentration from 0.8 to 1.2 mM led to a 50.76% increase in presynaptic bouton $[Mg^{2+}]_i$, from $396.34 \pm 205.12 \mu M$ to $597.53 \pm 175.41 \mu M$ (mean \pm SD) (**Fig. 4d**). Average bouton $[Mg^{2+}]_i$ at individual dendritic branches increased from $447.19 \pm 138.67 \mu M$ to $606.10 \pm 96.70 \mu M$ (**Fig. 4e**).

New **Fig. 4d, e**, **Fig. 6** and the calibration curve in new **Fig. S4** are shown below.

New Fig. S4 | Calibration of intracellular Mg^{2+} concentrations

a, Representative confocal images to show MgGrn fluorescence signals with various $[Mg^{2+}]_i$ in boutons. Ionophore (A-23187) was used to equilibrate various concentrations of $[Mg^{2+}]_i$. Following the collection of confocal images of MgGrn, FM4-64 labeling elicited by 600 APs at 10 Hz field simulation was utilized to visualize the boutons and normalize bouton volume (see Methods). **b**, Calibration curve ($n = 6$ biological repeats for each concentration) that was fitted by the Hill equation ($R^2 = 0.99$, $K_d = 0.91$ mM). Note the quasi-linear relationship between MgGrn fluorescence and real $[Mg^{2+}]_i$ within the range of 50–1200 f.u. Envelope, 95% CI.

New Fig. 4b–e (Measurement of absolute $[Mg^{2+}]_i$ in boutons)

b, Experimental design for measuring $[Mg^{2+}]_i$ and Pr in the same boutons. **c**, Representative boutons of similar size to show an inverse trend between Pr (FM4-64) and $[Mg^{2+}]_i$ (MgGrn). **d**, Measurement of Pr and $[Mg^{2+}]_i$ (mM) in boutons ($n = 204, 444$ from 3, 4 repeats). Inset, plot of Pr against $1/[Mg^{2+}]_i$. Double logarithmic fitting, $R^2 = 0.54$, $P < 0.0001$. **e**, Measurement of \overline{Pr} and $\overline{[Mg^{2+}]_i}$ (mM) of the boutons at individual dendritic branches ($n = 58, 53$ branches from 4, 4 repeats).

New Fig. 6 | Multiple pathways converge on intracellular Mg^{2+} concentration in regulating synaptic configurations

a, Time courses of changes in synaptic configuration and $[Mg^{2+}]_i$ after various treatments ($n = 5-9$ repeats per timepoint). **b**, Plots of D against $1/\overline{Pr}$ at dendritic branches 4 h after various treatments ($n = 81, 42, 64, 57$ branches from 7, 5, 6, 5 for control, AP5, ifenprodil and glutamate, respectively). Solid lines, linear regressions ($P < 0.0001$ for all). Dash-dot lines, bisectors. **c**, **d**, Measurements of D , \overline{Pr} , ΣPr and $[Mg^{2+}]_i$ 4 h after various treatments ($n = 5-11$ repeats). **e**, Plot of D , \overline{Pr} and ΣPr against $[Mg^{2+}]_i$ at the same timepoints after various treatments. Data are from (a, c and d) and Fig. 4a. Linear regressions, $P < 0.0001$ for $D-[Mg^{2+}]_i$ and $\overline{Pr}-[Mg^{2+}]_i$; $P = 0.79$ for $\Sigma Pr-[Mg^{2+}]_i$. Shadows, 95% CIs. Data are presented as mean \pm SEM. Two-tailed unpaired t tests (vs. time 0) (a) and Mann-Whitney tests (vs. control) (c, d). Significance: NS, no significance; * $P < 0.05$, ** $P < 0.01$, *** $P < 0.001$.

(v) Is there any knowledge that intracellular Mg^{2+} concentration changes with ageing, or are recovery experiments alone sufficient?

Aging poses a significant risk for Mg^{2+} deficit, as highlighted in various reviews (PMID: 29920011, 33573164, 20388094, 9595547, 8155490, 8155489, 20228001, 9513930, 17172010). Clinical studies reveal a substantial decrease in brain cerebrospinal fluid (CSF) Mg^{2+} concentration during aging and neurodegenerative diseases in humans (PMID: 1772577). Notably, elemental Mg^{2+} levels are markedly reduced in the brains of Alzheimer's disease patients (PMID: 11008926, 16131728). As regard to intracellular Mg^{2+} levels, clinical studies employed the phosphorus magnetic resonance spectrum (^{31}P MRS), a method for measuring intracellular ionized Mg^{2+} concentrations *in vivo*, demonstrate a significant decrease in body $[Mg^{2+}]_i$ during aging (PMID: 11759266, 31827445). These findings suggest that the decline in $[Mg^{2+}]_i$ serves as a hallmark of aging and neurodegeneration, emphasizing the crucial role of Mg^{2+} in protecting brain health. Indeed, both animal and human studies underscore the effectiveness of brain Mg^{2+} supplementation in addressing cognitive deficits associated with aging and neurodegenerative disorders.

In animal studies, brain Mg^{2+} supplementation exhibits a protective effect against aging-dependent cognitive declines (PMID: 30905744, 29920011). Our research demonstrates that cognitive impairments in aged animals (PMID: 20152124) and Alzheimer's disease model animals (PMID: 25213836) can be significantly ameliorated through brain Mg^{2+} supplementation. Additionally, brain Mg^{2+} supplementation shows promise in treating other neurodegenerative diseases. Independent studies report that MgT treatment effectively alleviates motor deficits and dopamine neuron loss in a mouse model of Parkinson's disease (PMID: 31806980).

Translational research assesses the efficacy of MgT (also known as L-threonic acid magnesium salt, L-TAMS) treatment in ameliorating cognitive deficits related to aging and neurological disorders. In our initial double-blind, placebo-controlled clinical study, MgT supplementation is shown to significantly reverse age-dependent cognitive impairment (PMID: 26519439). Consistent results are reproduced in other double-blind, placebo-controlled clinical studies conducted by independent groups (PMID: 36558392). Moreover, a clinical trial by Stanford University researchers demonstrates that MgT treatment effectively alleviates cognitive decline in Alzheimer's disease patients (DOI: 10.1093/geroni/igx004.661). Another open-label pilot study at Massachusetts General Hospital reports that MgT treatment improves cognitive functions in ADHD patients (PMID: 32162987).

Recently, the World Health Organization reached a consensus that dietary Mg^{2+} intake is lower than recommended in a majority of the world's population, especially in the aging demographic (<https://www.who.int/publications/i/item/9789241563550>; see also clinical trials, PMID: 12949381, 9513928). Therefore, based on the compelling evidence, elevating brain Mg^{2+} levels in

the elderly emerges as a promising strategy to minimize, or even prevent, aging-dependent cognitive deficits.

(vi) To confirm whether Mg^{2+} is actually involved in aging, it is necessary to examine what happens when Mg^{2+} is restricted in young mice.

We assume your inquiry pertains to the significance of Mg^{2+} in maintaining cognitive functions during brain aging.

Over the past decades, numerous animal and clinical studies have extensively documented progressive deficits in body Mg^{2+} levels during aging, likely stemming from insufficient intake and disorders in Mg^{2+} metabolism (for reviews, PMID: 33573164, 20388094, 9595547, 8155490, 8155489, 20228001, 9513930). However, the underlying mechanisms still require in-depth exploration. Mg^{2+} deficiency emerges as a high-risk factor for brain aging and neurodegeneration, crucial for sustaining brain health in both young and aged animals.

Firstly, Mg^{2+} sufficiency proves pivotal for maintaining brain health in young adults. On one hand, a 30–35% reduction in dietary Mg^{2+} causes a 40% decrease in $[Mg^{2+}]_i$ in the brains of young adult animals (PMID: 9296515), leading to significant impairments in cognitive functions, especially hippocampus-dependent learning and memory (examples include PMID: 15748878, 30500564, 33675126). Moreover, dietary Mg^{2+} deficiency induces systemic low-grade neuroinflammation in young adults, a hallmark of aging and neurodegenerative diseases (for a review, PMID: 36613667). On the other hand, an early study reported that chronic feeding of a high- Mg^{2+} diet (2% elemental Mg^{2+} in the diet) increases brain Mg^{2+} levels and improves learning behaviors in young rats (PMID: 6097334). Consistently, our studies have demonstrated that when young animals consume a normal- Mg^{2+} diet, supplementation of brain Mg^{2+} through oral intake of MgT in drinking water further enhances their learning and memory (PMID: 20152124).

Secondly, Mg^{2+} supplementation reverses cognitive declines in aging and neurodegeneration. Early studies have reported an improvement in cognitive functions in aged animals through a high dosage of Mg^{2+} in the diet (PMID: 6097334). Our previous studies show restored learning and memory in aged rats by elevating brain Mg^{2+} levels through MgT treatment (PMID: 20152124). Additionally, we demonstrate that cognitive declines can be effectively ameliorated by MgT treatment in Alzheimer's disease model mice (APP/PS1 transgenic mice) (PMID: 25213836). Consistently, an independent study indicates that MgT treatment can reduce neuroinflammation and alleviate cognitive decline in APP/PS1 transgenic mice (PMID: 26549801).

Overall, converging evidence suggests a crucial role of Mg^{2+} in maintaining brain health in young adults and during brain aging.

REVIEWERS' COMMENTS

Reviewer #1 (Remarks to the Author):

In this revision, by Zhou et al, have addressed some of the critiques in the prior review, previous concerns from the original critique still remain. The major concern that these studies are largely confirmatory of previous findings and concepts in the field still remain (Slutsky et al 2004 PMID: 15572114, , Slutsky et al 2010 PMID: 20152124, Zhou and Liu 2015 (PMID 26184109).

In particular, the authors have not addressed how culture development impacts their findings. While the authors now state that the cultures were $\sim 19 \pm 2$ days DIV they do not analyze how results from young cultures 14 DIV vs 28 DIV potentially differ. This was asked for in the original critique. Furthermore, the concern how a group of cultured neurons mimics the mechanisms utilized in an native intact in vivo circuit still remain.

In addition, while the new data in Fig S9 showing that Mg²⁺ supplementation restores levels of synaptic proteins this still confirms previous findings, specifically Zhou and Liu 2015 (PMID 26184109). The question about how a specific molecule implicated in these processes influences connectivity in response to Mg²⁺ supplementation remains. Simple mutations that alter Pr would allow testing on potential mechanisms of Mg²⁺ supplementation.

Reviewer #2 (Remarks to the Author):

The authors carefully addressed my concerns over the last version of the manuscript. I now support the publication of this paper.

Reviewer #3 (Remarks to the Author):

My initial comments consisted partly of conceptual criticism. The authors have responded courteously to these remarks, and at this stage I am largely convinced. Nevertheless, the necessity of the reply letter for the understanding of the manuscript implies that the exposition within the manuscript is at present inadequate. It is my contention that the manuscript would be substantially improved by an increase in

detail within the introductory section or the initial part of the discussion, thereby facilitating a comprehensive understanding of the background and aims of the study for readers not specialised in the field, such as myself, upon a single reading.

Compared to the other reviewers, my contributions were more limited in terms of specific feedback. Regarding comment (vi), it is unfortunate that the authors have sidestepped this issue despite the simplicity of the proposed experiments; however, the remaining concerns have been skilfully addressed and I have no outstanding complaints. Overall, however, the authors' courteous responses are impressive. Should there be a consensus among the other reviewers in favour of acceptance, I am prepared to concur.

REVIEWERS' COMMENTS

Reviewer #1 (Remarks to the Author):

In this revision, by Zhou et al, have addressed some of the critiques in the prior review, previous concerns from the original critique still remain. The major concern that these studies are largely confirmatory of previous findings and concepts in the field still remain (Slutsky et al 2004 PMID: 15572114, , Slutsky et al 2010 PMID: 20152124, Zhou and Liu 2015 (PMID 26184109).

Thank you for your interest in our study and insightful critiques in the review processes. We really appreciate that you have read through our previous studies. However, again, we would like to underscore that both the key findings and the concept in the current manuscript are novel and distinct from the three studies mentioned, as well as the classic concepts in the field.

In brief, the concept of the current study is that the configuration of synaptic connectivity at individual dendritic branches determines branch-specific properties of synaptic computations to optimize information processing, meanwhile the configuration itself is a biomarker indicating the current status of information coding capacity of individual branches. The key findings under this concept include: (1) the synaptic configuration can be regulated by intracellular Mg^{2+} ; (2) many signaling pathways concurrently alter intracellular Mg^{2+} levels when they regulate the synaptic configuration, suggesting that intracellular Mg^{2+} is likely a second messenger for the configurational transition. Of note, this concept and related experimental demonstrations have never been proposed or shown in our previous studies. As most of our points have been addressed in the prior responses (see the "Response to Referees" for Revision 1), here we'd like to focus on comparing the conceptual difference between the current manuscript and the three articles (PMID: 15572114, 20152124, 26184109).

In the Slutsky et al., 2004 paper (PMID: 15572114), the major finding is that reduction of background Ca^{2+} influx leads to upregulation of NMDARs, resulting in an enhancement of synaptic plasticity. The concept of this study is regarding how synaptic plasticity is regulated by Ca^{2+} activity. Among many endogenous factors, the extracellular Mg^{2+} shows a role in regulating synaptic plasticity by reducing background Ca^{2+} influx, providing a piece of proof-of-concept evidence. Following this paper, given the crucial role of synaptic plasticity for learning and memory in intact animals, we asked whether elevating brain Mg^{2+} levels improves learning and memory *in vivo*. This question was addressed in the Slutsky et al., 2010 paper (PMID: 20152124). Noteworthy that this paper is conceptually a translational research, focusing on the beneficial effect of brain Mg^{2+} supplementation on cognitive functions in living animals. As for the Zhou et al., 2015 paper (PMID: 26184109), it demonstrates that the density of functional boutons always adapts to the local energy supply, while intracellular Mg^{2+} is an endogenous factor regulating mitochondrial functions and local energy status; thus the concept of this paper is about how the

dynamic spatial allocation of energy supply along dendrites impacts local synaptic functionality. Obviously, there is no conceptual overlap between the three papers and the current one, albeit the involvement of Mg^{2+} in some proof-of-concept experiments of these studies.

Overall, we argue that the current study has significant conceptual advancements, rather than being a confirmatory follow-up study.

In particular, the authors have not addressed how culture development impacts their findings. While the authors now state that the cultures were $\sim 19 \pm 2$ days DIV they do not analyze how results from young cultures 14 DIV vs 28 DIV potentially differ. This was asked for in the original critique.

We put your initial comment and our response in the last round of review below. Our interpretation of your comment is that you are concerned about the potential impact of culture development on our major findings. We therefore revisited the cultures used throughout the current study and found that most data were collected on cultures of 17–20 days *in vitro* (d.i.v.). Within such a narrow time window, the impact of culture development should be minor (if any). More importantly, the 14 and 28 d.i.v. cultures used for data collection only accounts for 0.6% of all cultures (see the distribution curve). Therefore, the culture developmental impact is minimized in the current study. We believed that this analysis had addressed your primary concern in full.

Your initial comment:

“1). The authors rely on cell-culture model of DIV14-28. It is unclear how development of the culture model impacts their findings. There will be clear differences between DIV 14 and 28 neurons and this is never accounted for.”

Our response:

“Thank you for pointing out this issue. In our initial submission, we described in methods as 14–28 days in vitro (DIV) cultures, which was not an accurate methodological description. To make it more accurate, we revisited our data and calculated the mean age and standard deviation (SD) of the cultures we used. The cultures were 18.69 ± 1.92 DIV ($n = 487$ biological repeats), and the distribution of their ages is as follows. Most of the data were collected from 17–20 DIV cultures. We revised the inaccurate description in Methods, and now it reads “ 18.69 ± 1.92 days in vitro”. ”

Nevertheless, we fully agree with you that there may be differences between the two conditions from certain aspects. Here, to further address your concern, we compared the *Pr* distribution (the major readout in our study) in 14, 18, and 28 d.i.v. cultures, but found no significant difference ($n = 1338, 1576, 1262$ boutons from 3, 3, 3 repeats. Two-sided Kolmogorov-Smirnov tests. 14 vs. 28 d.i.v., $P = 0.3854$; 14 vs. 18 d.i.v., $P = 0.8562$; 18 vs. 28 d.i.v., $P = 0.7081$) (see the following cumulative distributions).

In summary, we would like to highlight three facts, (1) only 0.6% cultures used in this study were 14 and 28 d.i.v.; (2) most cultures used in this study were 17–20 d.i.v. (a narrow time window); and (3) no significant difference in *Pr* distribution (the major readout in our study) between 14–28 d.i.v. cultures. These facts allowed us to conclude that the culture development does not influence our findings.

Furthermore, the concern how a group of cultured neurons mimics the mechanisms utilized in a native intact *in vivo* circuit still remain.

We fully agree that concerns always remain when questioning the translation potential from *in vitro* synaptic network to *in vivo* circuits. In fact, that is the major limitation using such a simplified system to make conceptual advancements. However, as we have replied in the last "Response to Referees" file, we have several lines of evidence, including both *in vitro* and *in vivo* data (including clinical trials), to support the notion made in this manuscript. All these lines of evidence are consistent with the *in vitro* findings. Nevertheless, we admit that direct high-resolution observations at the dendritic branch level in the hippocampal circuits of intact animals have not

been made at the moment. However, such experiments are basically beyond the state-of-the-art for now, as one should measure *Pr*, Ca^{2+} activity, as well as intracellular Mg^{2+} levels in the same synapses of individual dendritic branches in the hippocampus of living brains. In response to your concern, our manuscript includes the following remarks in the Discussion to remind readers of this limitation.

“Future studies in behaving animals are required to obtain direct evidence at the single-synapse resolution regarding the role of synaptic configuration in information turnover during learning and memory.”

In addition, while the new data in Fig S9 showing that Mg^{2+} supplementation restores levels of synaptic proteins this still confirms previous findings, specifically Zhou and Liu 2015 (PMID 26184109). The question about how a specific molecule implicated in these processes influences connectivity in response to Mg^{2+} supplementation remains. Simple mutations that alter *Pr* would allow testing on potential mechanisms of Mg^{2+} supplementation.

We put your initial comment and our response in the last "Response to Referees" below (italic). Although we have addressed our points in the previous response, we would like to emphasize here again that knockdown (KD) or knockout (KO) of one or several of these proteins significantly alters or impairs synaptic connectivity in the brain, and in extreme cases, can be lethal for animals after birth (for a review, PMID: 15630409). Such kind of mutations are far from the real physiological conditions, limiting their significance in mechanistic investigation of the current study. Therefore, we did not conduct such mutation experiments in the current study given these concerns.

More importantly, the key concept of this paper is the synaptic configuration of dendrites and its impact on branch-specific synaptic computations during information processing, taking the Mg^{2+} supplementation as a proof-of-concept experimental condition. In this context, we thought the current understanding in molecular mechanisms underlying intracellular Mg^{2+} 's effect on the synaptic configuration is sufficient (for the molecular pathway, see Fig. 8). More details regarding the molecular mechanisms should be addressed in a number of following studies, typically not all in one single study.

Your initial comment:

“4) Given that the authors have already defined many of the molecules impacted by changes in $[\text{Mg}^{2+}]$ it would have been ideal to perturb one of these molecules or a molecule that is affected by aging and analyze the impact on connectivity by increasing Mg^{2+} levels. ”

Our response:

We presume you are referring to downstream molecules affected by intracellular Mg^{2+} levels, specifically Ca^{2+} -sensitivity-related proteins (CaSPs), including Rab3a, RIM1, Munc13-1, Synaptotagmin (SYT), ELKS, and Syntaxin-1, whose protein levels in presynaptic boutons fluctuate with changes in intracellular Mg^{2+} .

*We want to clarify that our study demonstrates a significant reduction in the protein levels of many CaSPs during aging (new **Fig. S9**). These proteins collectively influence the sensitivity of Ca^{2+} -triggered transmitter release during vesicle turnover and, consequently, synaptic connectivity. Knockdown (KD) or knockout (KO) of one or several of these proteins significantly alters or impairs synaptic connectivity in the brain, and in extreme cases, can be lethal for animals after birth (for a review, PMID: 15630409). Considering the massive impact of genetic manipulations on synaptic connectivity, we speculate that the elevation of Mg^{2+} levels may not be able to counteract the negative effect of these genetic manipulations on synaptic connectivity. Therefore, we did not conduct such experiments given these concerns.*

*Alternatively, we performed the following experiment to examine the role of elevating Mg^{2+} levels for restoring CaSP protein levels during aging in animals. Immunohistology on 70-nm ultrathin brain slices from the CA1 stratum radiatum (s.r.) region of the hippocampus (HP/CA1/s.r.) was conducted to measure changes in these molecules during aging. A significant decline in CaSP protein levels, ranging from $-19.01 \pm 15.84\%$ to $-25.79 \pm 16.07\%$ for individual ones, was observed when comparing aged animals (24 months of age) with young adults (6 months of age) (new **Fig. S9**), indicating a natural decline in CaSP protein levels during aging. Therefore, aging-associated decline in synaptic functionality is likely attributed to reduced levels of these proteins.*

*Importantly, when we elevated brain Mg^{2+} levels through oral MgT supplementation in aged animals, the reduction of CaSP protein levels was significantly mitigated in the HP/CA1/s.r. (new **Fig. S9**). These observations suggest that increasing brain Mg^{2+} levels can prevent aging-related changes in synaptic connectivity. Indeed, our electron microscopic data directly demonstrate that elevating brain Mg^{2+} levels rejuvenates synaptic connectivity in aged rats to that of young adult rats (**Fig. 7**). Intriguingly, both animal and human studies have documented a decline in brain Mg^{2+} levels during aging (see the response to **Reviewer-3-Point-vi**). Collectively, these findings emphasize the association between brain Mg^{2+} levels, changes in synaptic connectivity, and protein levels of presynaptic CaSPs during aging. This association may inspire new strategies for anti-brain aging and anti-neurodegeneration. In the revised manuscript, these results have been incorporated into new **Fig. S9**, as shown below.*

In response to your insightful suggestion, although we didn't conduct experiments to perturb one or several molecules affected by aging, we believe that our current data still provide insight into the involvement of brain Mg^{2+} levels in alterations of synaptic connectivity during aging. We agree that such an intriguing experiment you proposed should be considered in the future.

Reviewer #2 (Remarks to the Author):

The authors carefully addressed my concerns over the last version of the manuscript. I now support the publication of this paper.

Thank you for supporting the publication of our manuscript. We really appreciate your time and valuable comments during the review processes.

Reviewer #3 (Remarks to the Author):

My initial comments consisted partly of conceptual criticism. The authors have responded courteously to these remarks, and at this stage I am largely convinced. Nevertheless, the necessity of the reply letter for the understanding of the manuscript implies that the exposition within the manuscript is at present inadequate. It is my contention that the manuscript would be substantially improved by an increase in detail within the introductory section or the initial part of the discussion, thereby facilitating a comprehensive understanding of the background and aims of the study for readers not specialised in the field, such as myself, upon a single reading.

We thank you for the valuable suggestion. We agree that providing more details for the background of Mg^{2+} 's role in brain health would be helpful for a broader audience. However, due to the length limitation and the logic flow of the current manuscript, it is difficult to put all the information in either introduction or discussion. In response to your suggestion, we have summarized related materials, and alternatively, prepared a "Supplementary Notes" to be published as online materials together with the Article. This would facilitate a more comprehensive understanding of the scientific background to the audience who are interested in such background, without deconstructing the current flow of this manuscript. In addition, we have also included a "Supplementary Table 1" to list and describe the biophysical variables throughout the manuscript to help a broader audience to quickly understand the complex computations in this manuscript.

Compared to the other reviewers, my contributions were more limited in terms of specific feedback. Regarding comment (vi), it is unfortunate that the authors have sidestepped this issue despite the simplicity of the proposed experiments; however, the remaining concerns have been skilfully addressed and I have no outstanding complaints. Overall, however, the authors' courteous responses are impressive. Should there be a consensus among the other reviewers in favour of acceptance, I am prepared to concur.

We really appreciate your kind words and the compliment and thank you for supporting the publication of our manuscript.

As regard to "the reply to comment (vi)", we had listed several previous studies showing cognitive impairments and systemic neuroinflammation after reducing Mg^{2+} supplementation in healthy, young animals, suggesting that restricting Mg^{2+} supply in young animals may promote aging-like declines. We hope you could find that the evidence actually addresses your question. We put the related paragraph in the last "Response to Referees" document as follows.

“Comment (vi): To confirm whether Mg^{2+} is actually involved in aging, it is necessary to examine what happens when Mg^{2+} is restricted in young mice.

Mg^{2+} sufficiency proves pivotal for maintaining brain health in young adults. On one hand, a 30–35% reduction in dietary Mg^{2+} causes a 40% decrease in $[Mg^{2+}]_i$ in the brains of young adult animals (PMID: 9296515), leading to significant impairments in cognitive functions, especially hippocampus-dependent learning and memory (examples include PMID: 15748878, 30500564, 33675126). Moreover, dietary Mg^{2+} deficiency induces systemic low-grade neuroinflammation in young adults, a hallmark of aging and neurodegenerative diseases (for a review, PMID: 36613667). ”